# Online Episodic Convex Reinforcement Learning

**Bianca Marin Moreno** [*1 2 3] **Khaled Eldowa** [*4 5] **Pierre Gaillard** [1] **Margaux Brégère** [2 6] **Nadia Oudjane** [2 3]

## Abstract

We study online learning in episodic finite-horizon Markov decision processes (MDPs) with convex objective functions, known as the concave utility reinforcement learning (CURL) problem. This setting generalizes RL from linear to convex losses on the state-action distribution induced by the agent's policy. The non-linearity of CURL invalidates classical Bellman equations and requires new algorithmic approaches. We introduce the first algorithm achieving near-optimal regret bounds for online CURL without any prior knowledge on the transition function. To achieve this, we use an online mirror descent algorithm with varying constraint sets and a carefully designed exploration bonus. We then address for the first time a bandit version of CURL, where the only feedback is the value of the objective function on the state-action distribution induced by the agent's policy. We achieve a sub-linear regret bound for this more challenging problem by adapting techniques from bandit convex optimization to the MDP setting.

## 1. Introduction

Reinforcement learning (RL) studies the problem where an agent interacts with an environment over time, adhering to a probabilistic policy that maps states to actions and aiming to minimize the cumulative expected losses. The environment's dynamics are represented by a Markov decision process (MDP), assumed here to be episodic, with episodes of length $N$, a finite state space $\mathcal{X}$, a finite action space $\mathcal{A}$, and a sequence of probability transition kernels $p := (p_n)_{n\in[N]}$, such that for each $(x, a) \in \mathcal{X} \times \mathcal{A}$, $p_n(\cdot|x, a) \in \Delta_{\mathcal{X}}$, the simplex over the state space. Formally, the RL problem involves finding a policy $\pi$ that, under a transition kernel $p$, induces a state-action distribution sequence $\mu^{\pi,p} \in (\Delta_{\mathcal{X}\times\mathcal{A}})^N$ minimizing the inner product with a loss vector $\ell := (\ell_n)_{n\in[N]}$, with $\ell_n \in \mathbb{R}^{\mathcal{X}\times\mathcal{A}}$, i.e.: $\min_{\pi\in(\Delta_{\mathcal{A}})^{\mathcal{X}\times N}} \langle \ell, \mu^{\pi,p}\rangle$. A large body of literature is devoted to solving the RL problem efficiently and with theoretical guarantees in many challenging environments (Bertsekas, 2019; Sutton & Barto, 2018).

However, numerous practical problems entail more intricate objectives, such as those encountered within the Concave Utility Reinforcement Learning (CURL) framework (Hazan et al., 2019; Zahavy et al., 2021) (also known as convex RL). The CURL problem consists in minimizing a convex function (or maximizing a concave function) on the state-action distributions induced by an agent's policy:

$$\min_{\pi\in(\Delta_{\mathcal{A}})^{\mathcal{X}\times N}} F(\mu^{\pi,p}). \tag{1}$$

In addition to RL, other examples of machine learning problems that can be written as CURL are pure exploration (Hazan et al., 2019; Mutti et al., 2021; 2022b), where $F(\mu^{\pi,p}) = \langle \mu^{\pi,p}, \log(\mu^{\pi,p})\rangle$; imitation learning (Ghasemipour et al., 2020; Lavington et al., 2022) and apprenticeship learning (Zahavy et al., 2019; Abbeel & Ng, 2004), where $F(\mu^{\pi,p}) = D_g(\mu^{\pi,p}, \mu^*)$, with $D_g$ representing a Bregman divergence induced by a function $g$ and $\mu^*$ being a behavior to be imitated; certain instances of mean-field control (Bensoussan et al., 2013), where $F(\mu^{\pi,p}) = \langle \ell(\mu^{\pi,p}), \mu^{\pi,p}\rangle$; mean-field games with potential rewards (Lavigne & Pfeiffer, 2023); risk-averse RL (Garcıa & Fernández, 2015; Pan et al., 2019; Greenberg et al., 2022), among others. The non-linearity of CURL alters the additive structure inherent in standard RL, invalidating the classical Bellman equations. Consequently, dynamic programming approaches become infeasible, necessitating the development of novel methodologies.

A natural extension of CURL is the online scenario, wherein a sequence of policies $(\pi^t)_{t\in[T]}$ is computed over $T$ episodes, aimed at minimizing a cumulative loss $L_T := \sum_{t=1}^T F^t(\mu^{\pi^t,p})$, where the objective $F^t$ can change arbitrarily (known as the adversarial scenario (Even-Dar et al., 2009)), and the MDP probability kernel $p$ is unknown. Most existing approaches to CURL fail to address the challenges

[*]Equal contribution [1]Univ. Grenoble Alpes, Inria, CNRS, Grenoble INP, LJK, 38000 Grenoble, France. [2]EDF Lab, 7 bd Gaspard Monge, 91120 Palaiseau, France [3]FiME (Laboratoire de Finance des Marchés de l'Energie - Dauphine, CREST, EDF R&D) [4]Università degli Studi di Milano, Milan, Italy [5]Politecnico di Milano, Milan, Italy [6]Sorbonne Université LPSM, Paris, France. Correspondence to: Bianca Marin Moreno <bianca.marin-moreno@inria.fr>.

*Proceedings of the $42^{nd}$ International Conference on Machine Learning*, Vancouver, Canada. PMLR 267, 2025. 

of the online setting (adversarial losses and unknown dynamics). The few methods that attempt to tackle this problem rely on strong assumptions about the probability transition kernel (Moreno et al., 2024), which can be overly restrictive in real-world scenarios. To overcome this, we need an approach capable of optimizing the objective function while simultaneously learning the environment, effectively balancing the exploration-exploitation dilemma.

**Motivations.** The CURL framework presented in this paper is motivated by a variety of application scenarios, some of which we outline below. *Energy grid optimization*: To balance the energy production with the consumption, an energy provider may want to control the average consumption of electrical appliances (electric vehicles, water heaters, etc) to better match a target consumption. The task involves daily control, with the target consumption varying daily due to fluctuations in energy production. To protect user privacy, the energy provider has limited access to individual trajectories, but receives the average consumption of the whole population at the end of each day. The loss is usually quadratic on the state-action distribution. This problem can be framed as our CURL formulation. See (Coffman et al., 2023; Moreno et al., 2025). *Mean-field games (MFG) with potential reward*: As shown by (Barakat et al., 2023), a MFG with potential reward can be framed as a CURL problem. Therefore, any sequential decision problem with a large population of anonymous agents with symmetric interests and potential rewards, such as epidemic spreading, crowd motion control, etc, can be cast as CURL.

**Contribution 1.** In the full-information feedback setting, where the objective function $F^t$ is fully revealed to the learner at the end of episode $t$, we propose the first method achieving sub-linear regret for online CURL with adversarial losses and unknown transition kernels, without relying on additional model assumptions. Our algorithm uses an Online Mirror Descent (OMD) variant incorporating well-designed exploration bonuses into the sub-gradient of the objective function to handle the exploration-exploitation trade-off. It achieves a regret of $\tilde{O}(\sqrt{T})$, matching the state-of-the-art (SoTA) in more restricted settings (Moreno et al., 2024), while obtaining a closed-form solution.

**Contribution 2.** We extend our approach to incorporate bandit feedback on the objective function. We first consider the RL case where $F^t(\mu) := \langle \ell^t, \mu \rangle$. Bandit feedback in this setting means that the agent only observes the loss function in the state-action pairs they visit during each episode, i.e. $(\ell_n^t(x_n^t, a_n^t))_{n\in[N]}$ where $(x_n^t, a_n^t)_{n\in[N]}$ is the agent's trajectory. We obtain the optimal regret of $\tilde{O}(\sqrt{T})$ in this setting. We then address for the first time the general CURL problem under more strict bandit feedback. In this setting, the learner only has access to the value of the objective function evaluated on the state-action distribution sequence induced by the agent's policy, i.e., $F^t(\mu^{\pi^t,p})$. We propose two algorithms for this setting and show that they achieve sub-linear regret. One algorithm requires that the MDP is known, while the other, under the assumption that the probability transition kernel is lower bounded by a positive constant, operates in the setting where the MDP is estimated progressively from observed trajectories. We rely on gradient estimation techniques from the bandit convex optimization literature, even as the peculiar structure of our constraint set and uncertainty regarding the true transition kernel present some unique challenges.

### 1.1. Related Work

**Offline CURL.** An extensive line of work focus on the offline version of CURL (Problem (1)), where the objective function is known and fixed. The methodologies proposed by (Zhang et al., 2020; 2021; Barakat et al., 2023) rely on policy gradient techniques, requiring the estimation of $F$'s gradient concerning the policy $\pi$, a task often complex. Taking a different approach, Zahavy et al. (2021) cast the CURL problem as a min-max game using Fenchel duality, demonstrating that conventional RL algorithms can be tailored to fit the CURL framework. Recently, Geist et al. (2022) established that CURL is a specific instance of mean-field games. Moreover, Moreno et al. (2024) undertake a convexification of Problem (1) and propose a mirror descent algorithm with a non-standard Bregman divergence. Mutti et al. (2022a; 2023) study the gap between evaluating agent performance over infinite realizations versus finite trials and question the classic CURL formulation in Eq. (1). However, they show that non-Markovian policies can be necessary to optimize the finite trials objective, which entails an increased computational burden. On the other hand, the occupancy-measure-based formulation that we study can be solved efficiently, and allows direct comparison with methods from the CURL literature such as (Moreno et al., 2024). Moreover, in application scenarios with many homogeneous agents, a mean-field approach can justify this choice, as we discuss in Sec. 1.

**Online CURL.** To the best of our knowledge, Greedy MD-CURL from (Moreno et al., 2024) is the only regret minimization algorithm designed for online CURL. However, it only achieves sublinear regret when the system dynamics follow the form $x_{n+1} = g_n(x_n, a_n, \varepsilon_n)$, where $g_n$ is a known deterministic function, and $\varepsilon_n$ is an external noise with an unknown distribution independent of $(x_n, a_n)$, which significantly limits its applicability, as we empirically show in Sec. 5. This assumption simplifies the problem, as the algorithm only needs to learn the noise distribution, which can be done independently of the policy, eliminating the need for exploration. In contrast, our approach does not assume any specific form for the dynamics, which introduces the challenge of developing a policy that minimizes

*Table 1.* Comparisons of SoTA finite-horizon tabular MDPs methods. MD stands for Mirror Descent, KL for Kullback-Leibler divergence and $\Gamma$ is defined in Eq. (4). MD + $(\cdot)$ indicates the regularization added to the MD iteration. MD on $\pi$ indicates a policy optimization approach in which MD iterations are performed on policies instead of state-action distributions (occupancy-measures).

| | Algorithm | Optimal regret in $T$ | CURL | Closed-form | Exploration | No model assumption | Adversarial Losses | Bandit feedback |
|---|---|---|---|---|---|---|---|---|
| (Jin et al., 2020) | MD + KL | ✓ | ✗ | ✗ | UCRL | ✓ | ✓ | ✓ |
| (Moreno et al., 2024) | MD + $\Gamma$ | ✓ | ✓ | ✓ | None | ✗ | ✓ | ✗ |
| (ours) | MD + $\Gamma$ | ✓ | ✓ | ✓ | Bonus | ✓ | ✓ | ✓ |
| (Luo et al., 2021) | MD on $\pi$ | ✓ | ✗ | ✓ | Bonus | ✓ | ✓ | ✓ |

total loss while simultaneously enabling sufficient exploration to improve estimates of the transition kernels. The technical novelty we introduce to overcome this challenge are well-designed exploration bonuses detailed in Sec. 3.

The work of (Rosenberg & Mansour, 2019b) also studies convex performance criteria in adversarial MDPs, but under a different setup. They use *fixed*, *known* convex functions applied to linear losses, i.e., $F(\langle \mu, \ell^t \rangle)$. In contrast, our setting generalizes this by allowing the convex function $F$ itself to be adversarial and applied directly to the occupancy measure, i.e., $F^t(\mu)$.

**RL approaches.** *Model-optimistic methods* construct a set of plausible MDPs by forming confidence bounds around the empirical transition kernels, then select the policy that maximizes the expected reward in the best feasible MDP. A key example of this approach is UCRL (Upper Confidence RL) methods (Jaksch et al., 2008; Zimin & Neu, 2013; Rosenberg & Mansour, 2019b; Jin et al., 2020). While these methods offer strong theoretical guarantees, they are often difficult to implement due to the complexity of optimizing over all plausible MDPs. While we believe these approaches could be generalized to CURL, their computational complexity has led us to propose an alternative method. *Value-optimistic methods* are value-based approaches that compute optimistic value functions, rather than optimistic models, using dynamic programming. An example is UCB-VI (Azar et al., 2017). However, these methods are limited to stochastic losses. *Policy-optimization (PO) methods* directly optimize the policy and are widely used in RL due to their faster performance and closed-form solutions. Recently, Luo et al. (2021) achieved SoTA regret for PO methods with adversarial losses and bandit feedback by introducing *dilated bonuses*, which satisfy a *dilated* Bellman equation and are added to the $Q$-function. However, their approach cannot be applied here due to CURL's non-linearity (the expectation of the trajectory appears inside the objective function) which invalidates the Bellman's equations.

We achieve our results by computing local bonuses and adding them to the (sub-)gradient of the objective function in each OMD instance as exploration bonuses. This is more computationally efficient than model-optimistic approaches and addresses the exploration issues in previous online CURL methods. We believe our analysis is of independent interest, as it also offers a new way to study RL approaches over occupancy measures, while providing closed-form solutions. See Table 1 for comparisons.

## 2. Problem Formulation

### 2.1. Setting

For a finite set $\mathcal{S}$, $|\mathcal{S}|$ represents its cardinality, while $\Delta_{\mathcal{S}}$ denotes the $|\mathcal{S}|$-dimensional simplex. For all $d \in \mathbb{N}$ we denote $[d] := \{1, \dots, d\}$. We let $\|\cdot\|_1$ be the $L_1$ norm, and for all $v := (v_n)_{n\in[N]}$, such that $v_n \in \mathbb{R}^{\mathcal{X}\times\mathcal{A}}$ we define $\|v\|_{\infty,1} := \sup_{1\leqslant n\leqslant N} \|v_n\|_1$. We denote by $\|\cdot\|_{1,\infty}$ its dual. Let $\Pi := (\Delta_{\mathcal{A}})^{\mathcal{X}\times N}$ denote the set of policies. We consider an episodic MDP as introduced in Sec. 1. We assume that the initial state-action pair of an agent is sampled from a fixed distribution $\mu_0 \in \Delta_{\mathcal{X}\times\mathcal{A}}$ at the beginning of each episode. At time step $n \in [N]$, the agent moves to a state $x_n \sim p_n(\cdot|x_{n-1}, a_{n-1})$, and chooses an action $a_n \sim \pi_n(\cdot|x_n)$ by means of a policy $\pi_n : \mathcal{X} \to \Delta_{\mathcal{A}}$. When the agent follows a policy $\pi := (\pi_n)_{n\in[N]}$ for an episode in an environment described by the MDP with a transition kernel $p$, this induces a state-action distribution, which we denote by $\mu^{\pi,p} := (\mu_n^{\pi,p})_{n\in[N]}$, that can be calculated recursively for all $(n, x, a) \in [N] \times \mathcal{X} \times \mathcal{A}$, by

$$\begin{aligned} \mu_0^{\pi,p}(x,a) &= \mu_0(x,a) \\ \mu_n^{\pi,p}(x,a) &= \sum_{(x',a')} \mu_{n-1}^{\pi,p}(x',a') p_n(x|x',a') \pi_n(a|x). \end{aligned} \tag{2}$$

We define the set of all state-action distribution sequences satisfying the dynamics of the MDP as

$$\mathcal{M}_{\mu_0}^p := \Big\{ \mu \in (\Delta_{\mathcal{X}\times\mathcal{A}})^N \,\Big|\, \sum_{a'\in\mathcal{A}} \mu_n(x',a') = \sum_{x\in\mathcal{X}, a\in\mathcal{A}} p_n(x'|x,a)\mu_{n-1}(x,a) \;, \forall x' \in \mathcal{X}, \forall n \in [N] \Big\}. \tag{3}$$

For any $\mu \in \mathcal{M}_{\mu_0}^p$, there is a strategy $\pi$ such that $\mu^{\pi,p} = \mu$. It suffices to take $\pi_n(a|x) \propto \mu_n(x,a)$ when the normal-

ization factor is non-zero, and arbitrarily defined otherwise. Let $\mathcal{M}_{\mu_0}^{p,*}$ be the subset of $\mathcal{M}_{\mu_0}^{p}$ where the corresponding policies $\pi$ satisfy $\pi_n(a|x) \neq 0$ for all $(x, a)$. For any two probability transition kernels $p, q$, we define $\Gamma : \mathcal{M}_{\mu_0}^{p} \times \mathcal{M}_{\mu_0}^{q,*} \to \mathbb{R}$ such that, for all $\mu, \mu' \in \mathcal{M}_{\mu_0}^{p} \times \mathcal{M}_{\mu_0}^{q,*}$ with policies $\pi$, $\pi'$,

$$\Gamma(\mu, \mu') := \sum_{n=1}^{N} \mathbb{E}_{(x,a)\sim\mu_n(\cdot)}\Big[\log\Big(\tfrac{\pi_n(a|x)}{\pi'_n(a|x)}\Big)\Big]. \quad (4)$$

In the online extension of CURL, the objective function for episode $t$ is denoted as $F^t := \sum_{n=1}^{N} f_n^t$, where $f_n^t : \Delta_{\mathcal{X}\times\mathcal{A}} \to \mathbb{R}$ is convex and $L$-Lipschitz with respect to the $\|\cdot\|_1$ norm (hence $F^t$ is $L_F$-Lipschitz with respect to the norm $\|\cdot\|_{\infty,1}$ with $L_F := LN$). The objective function $F^t$ is unknown to the learner in the start of episode $t$. In this paper, we examine three types of objective function feedback: *Full-information*: In this case, $F^t$ is fully disclosed to the learner at the end of episode $t$, and is treated in Sec. 3.2. *Bandit in RL*: Here, $F^t(\mu) := \langle \ell^t, \mu\rangle$, and the learner observes the loss function only for the state-action pairs visited, i.e., $(\ell_n^t(x_n^t, a_n^t))_{n\in[N]}$, which is covered in Sec. 4.1. *Bandit in CURL*: In this scenario, the learner only has access to the objective function evaluated on the state-action distribution sequence induced by the agent's policy, i.e., $F^t(\mu^{\pi^t,p})$, and is treated in Sec. 4.2.

The learner's goal is to compute a sequence of strategies $(\pi^t)_{t\in[T]}$, where $T$ represents the total number of episodes, that minimizes their total loss $L_T := \sum_{t=1}^{T} F^t(\mu^{\pi^t,p})$. The learner's performance is evaluated by comparing it to any policy $\pi \in (\Delta_{\mathcal{A}})^{\mathcal{X}\times N}$ using the static regret:

$$R_T(\pi) := \sum_{t=1}^{T} F^t(\mu^{\pi^t,p}) - F^t(\mu^{\pi,p}). \quad (5)$$

We assume the probability transition kernel $p$ is unknown to the learner. Hence, to minimize its total loss, the learner must optimize the objective function while simultaneously learn the environment dynamics, facing an exploration-exploitation dilemma. The interaction between the learner and the environment proceeds in episodes. At each episode $t$, the learner selects a policy $\pi^t$, sends it to the agent, and observes its trajectory $o^t := (x_0^t, a_0^t, \ldots, x_N^t, a_N^t)$. The learner uses this observation to compute an estimation of the probability transition kernel $\widehat{p}^{t+1}$. At the end of episode $t$, the learner receives one of the three feedbacks described above for the objective function $F^t$, and then calculates the policy for the next episode, $\pi^{t+1}$, based on $\pi^t, \widehat{p}^{t+1}$, and the feedback on $F^t$.

### 2.2. Preliminary Results

The results in this section are either known or extensions of existing results needed for the analysis.

Since the probability transition kernel is unknown, we propose an online mirror descent (OMD) instance that optimizes over the state-action distributions induced by the estimated MDP as if it was the true model. This approach differs from the model-optimistic methods for RL discussed in Sec. 1.1 where each iteration is performed over the union of all state-action distribution sets induced by MDPs within a confidence set around the estimated model, which results in a computationally expensive optimization problem per iteration. Lemma 2.1 presents an auxiliary result concerning the quality of the state-action distribution sequence $(\mu^t)_{t\in[T]}$ when $\mu^t$ is the solution of Eq. (6), an OMD instance on the set of state-action distributions induced by a transition kernel $q^t$. It extends the upper bound result from (Moreno et al., 2024) for OMD with smoothly varying constraint sets to any sequence of bounded vectors $(z^t)_{t\in[T]}$ and any sequence of smootlhy varying transitions $(q^t)_{t\in[T]}$.

**Lemma 2.1.** *Let $(q^t)_{t\in[T]}$ be a sequence of probability transition kernels and $(z^t)_{t\in[T]}$ a sequence of vectors in $\mathbb{R}^{N\times|\mathcal{X}|\times|\mathcal{A}|}$, such that $\max_{t\in[T]}\|z^t\|_{1,\infty} \leqslant \zeta$. Initialize $\pi_n^1(a|x) := 1/|\mathcal{A}|$. For $t \in [T]$, let $\tilde{\pi}^t := \frac{t}{t+1}\pi^t + \frac{1}{t+1}|\mathcal{A}|^{-1}$ be a smoothed version of the policy and compute iteratively*

$$\mu^{t+1} \in \mathop{\arg\min}_{\mu\in\mathcal{M}_{\mu_0}^{q^{t+1}}} \tau\langle z^t, \mu\rangle + \Gamma(\mu, \mu^{\tilde{\pi}^t,q^t}). \quad (6)$$

*Then, there is a $\tau > 0$ such that, for any sequence $(\nu^t)_{t\in[T]}$, with $\nu^t := \nu^{\pi,q^t}$ for a common policy $\pi$,*

$$\sum_{t=1}^{T}\langle z^t, \mu^t - \nu^t\rangle \leqslant O\big(\zeta N\sqrt{V_T|\mathcal{X}|\log(|\mathcal{A}|)T\log(T)}\big),$$

*where* $V_T \geqslant 1 + \max_{(n,x,a)} \sum_{t=1}^{T-1} \|q_n^t(\cdot|x,a) - q_n^{t+1}(\cdot|x,a)\|_1.$

This lemma is proved in App. C. It is known (Moreno et al., 2024) that for the divergence $\Gamma$ defined in Eq. (4), Eq. (6) has a closed-form solution for the policy (see App. A.2).

**Learning the model.** Since the learner does not know the probability transition kernel, it must estimate $p$ from the agents' trajectories. Below we present the empirical way for estimating the transition and a well-known result (Lem. 2.2) on its quality using Hoeffding's inequality. Let $N_n^t(x,a) = \sum_{s=1}^{t-1}\mathbb{1}_{\{x_n^s=x, a_n^s=a\}}$, $M_n^t(x'|x,a) = \sum_{s=1}^{t-1}\mathbb{1}_{\{x_{n+1}^s=x', x_n^s=x, a_n^s=a\}}$. The learner's estimate for the transition kernel at the end of episode $t-1$, to be used in episode $t$, is as follows

$$\widehat{p}_{n+1}^t(x'|x,a) := \frac{M_n^t(x'|x,a)}{\max\{1, N_n^t(x,a)\}}. \quad (7)$$

**Lemma 2.2** (Lem. 17 of Jaksch et al., 2008)**.** *For any $0 < \delta < 1$, with a probability of at least $1-\delta$,*

$$\|p_n(\cdot|x,a) - \widehat{p}_n^t(\cdot|x,a)\|_1 \leqslant \sqrt{\frac{2|\mathcal{X}|\log\Big(\frac{|\mathcal{X}||\mathcal{A}|NT}{\delta}\Big)}{\max\{1, N_{n-1}^t(x,a)\}}}$$

*holds simultaneously for all $(t,n,x,a) \in [T]\times[N]\times\mathcal{X}\times\mathcal{A}$.*

These results suffice for analyzing CURL with full-information feedback (Sec. 3). For bandit feedback, more refined tools are needed. In bandit RL, we need Bernstein's inequality to bound the $L_1$ distance (Lem. D.2). In bandit CURL, we also need a bound on the Kullback-Leibler (KL) divergence (Lem. E.3), which requires the Laplace (add-one) estimator (Eq. (51)), as the KL of the empirical one can be unbounded.

## 3. Exploration Bonus in CURL

We now present our novel approach for online CURL with adversarial losses and unknown dynamics.

### 3.1. Limitations of previous approaches

The performance measure of a learner playing a sequence of strategies $(\pi^t)_{t\in[T]}$ is given by the static regret defined in Eq. (5). Using the estimate of the probability transition kernel $\widehat{p}^t$ computed by the learner, the static regret can be further decomposed as follows

$$\begin{aligned} R_T(\pi) &= \sum_{t=1}^T F^t(\mu^{\pi^t,p}) - F^t(\mu^{\pi^t,\widehat{p}^t}) \\ &+ \sum_{t=1}^T F^t(\mu^{\pi^t,\widehat{p}^t}) - F^t(\mu^{\pi,p}) \\ &\leqslant \underbrace{\sum_{t=1}^T \langle \nabla F^t(\mu^{\pi^t,p}), \mu^{\pi^t,p} - \mu^{\pi^t,\widehat{p}^t}\rangle}_{R_T^{\text{MDP}}} \qquad (8) \\ &+ \underbrace{\sum_{t=1}^T \langle \nabla F^t(\mu^{\pi^t,\widehat{p}^t}), \mu^{\pi^t,\widehat{p}^t} - \mu^{\pi,p}\rangle}_{R_T^{\text{policy}}}, \end{aligned}$$

where the inequality comes from the convexity of $F^t$. Let $\xi_n^t(x,a) := \|p_n(\cdot|x,a) - \widehat{p}_n^t(\cdot|x,a)\|_1$. The term $R_T^{\text{MDP}}$, accounts for the error in estimating the MDP, and satisfies $R_T^{\text{MDP}} = \tilde{O}(\sqrt{T})$ with high probability. This is a classic result (see Neu et al., 2012). We first show that

$$R_T^{\text{MDP}} \leqslant L \sum_{t=1}^T \sum_{n=1}^N \sum_{i=0}^{n-1} \sum_{x,a} \mu_i^{\pi^t,p}(x,a) \xi_{i+1}^t(x,a). \qquad (9)$$

Then, using Lem. 2.2 and that $N_n^t(x,a)$ increases with the empirical version of the state-action distribution $\mu_n^{\pi^t,p}(x,a)$ we achieve the final bound (see App. B.2). The second term, $R_T^{\text{policy}}$, depends on the algorithm used to derive the policies. As mentioned in Sec. 1.1, model-optimistic approaches could be adapted to CURL, but they are computationally expensive. To achieve low complexity, we explore potential problems that might arise from the absence of explicit exploration. We decompose this regret term as follows:

$$\begin{aligned} R_T^{\text{policy}} &= \underbrace{\sum_{t=1}^T \langle \nabla F^t(\mu^{\pi^t,\widehat{p}^t}), \mu^{\pi^t,\widehat{p}^t} - \mu^{\pi,\widehat{p}^t}\rangle}_{R_T^{\text{policy/MD}}} \\ &+ \underbrace{\sum_{t=1}^T \langle \nabla F^t(\mu^{\pi^t,\widehat{p}^t}), \mu^{\pi,\widehat{p}^t} - \mu^{\pi,p}\rangle}_{R_T^{\text{policy/MDP}}}. \end{aligned}$$

Assume the learner computes its policy sequence $(\pi^t)_{t\in[T]}$ by solving Eq. (6) with $q^{t+1} := \widehat{p}^{t+1}$ and $z^t := \nabla F^t(\mu^{\pi^t,\widehat{p}^t})$. Hence, from Lem. 2.1, $R_T^{\text{policy/MD}} = \tilde{O}(\sqrt{T})$ (Lemmas A.3 and A.4 in the Appendix demonstrate that $\sum_{t=1}^T \|\widehat{p}^{t+1}(\cdot|x,a) - \widehat{p}^t(\cdot|x,a)\|_1 \leqslant e\log(T)$. By hypothesis, $\|\nabla F^t(\mu^{\pi^t,\widehat{p}^t})\|_{1,\infty} \leqslant L_F$. Hence, we meet all the assumptions from Lem. 2.1). But the term $R_T^{\text{policy/MDP}}$ poses a challenge. It can be decomposed as $R_T^{\text{MDP}}$ in Eq. (9). However, the state-action distribution multiplying $\xi_{i+1}^t(x,a)$ would either be $\mu_i^{\pi,p}(x,a)$ or $\mu_i^{\pi,\widehat{p}^{t+1}}(x,a)$, and neither is related to $N_i^t(x,a)$. Consequently, we do not have the same convergence effect as $R_T^{\text{MDP}}$. In fact, this term can become prohibitively large. Without exploration, previous work using similar analysis (Moreno et al., 2024) only achieved optimal regret under strong model assumptions, limiting its applicability in realistic scenarios.

### 3.2. CURL with full-information feedback

We outline our idea to overcome previous limitations presented in Subsec. 3.1. Let $b^t := (b_n^t)_{n\in[N]}$ be a sequence of vectors, to be properly defined later, such that $b_n^t \in \mathbb{R}^{\mathcal{X}\times\mathcal{A}}$. We assume that $\pi^t$ is the policy inducing $\mu^t$ computed as in Eq. (6) with $q^t := \widehat{p}^t$, but instead of considering $z^t = \nabla F^t(\mu^{\pi^t,\widehat{p}^t})$ as the (sub-)gradient of MD to be used in episode $t+1$, we let $z^t := \nabla F^t(\mu^{\pi^t,\widehat{p}^t}) - b^t$, i.e.,

$$\mu^{t+1} := \underset{\mu\in\mathcal{M}_{\mu_0}^{\widehat{p}^{t+1}}}{\arg\min} \Big\{ \tau\langle \nabla F^t(\mu^{\pi^t,\widehat{p}^t}) - b^t, \mu\rangle + \Gamma(\mu, \tilde{\mu}^t) \Big\}.$$

If we assume that $b^t$ is such that, for all $t \in [T]$ and for some $\zeta > 0$, $\|\nabla F^t(\mu^{\pi^t,\widehat{p}^t}) - b^t\|_{1,\infty} \leqslant \zeta$, then by Lem. 2.1 and by adding and subtracting the bonus vector, we would have that $R_T^{\text{policy}}$ is bounded by

$$\tilde{O}(\sqrt{T}) + \textstyle\sum_{t=1}^T \langle b^t, \mu^{\pi^t,\widehat{p}^t} - \mu^{\pi,\widehat{p}^t}\rangle + R_T^{\text{policy/MDP}}. \quad (10)$$

Let $C_\delta := \sqrt{2|\mathcal{X}|\log(|\mathcal{X}||\mathcal{A}|NT/\delta)}$, and for all $n \in \{0,[N]\}$, $(x,a) \in \mathcal{X}\times\mathcal{A}$, let

$$b_n^t(x,a) := L(N-n)\frac{C_\delta}{\sqrt{\max\{1, N_n^t(x,a)\}}}. \qquad (11)$$

Note that $\|b_n^t\|_\infty \leqslant LNC_\delta$, ensuring that the hypothesis of Lem. 2.1 remains valid for this sequence. Decomposing

$R_T^{\text{policy/MDP}}$ as we do for $R_T^{\text{MDP}}$ in Eq. (9), and then applying Lem. 2.2, we get that for any $\delta \in (0,1)$, with probability at least $1-\delta$, $R_T^{\text{policy/MDP}}$ is bounded by

$$\begin{aligned} LC_\delta \sum_{t=1}^{T} \sum_{n=0}^{N-1} (N-n) \sum_{x,a} \frac{\mu_n^{\pi,\widehat{p}^t}(x,a)}{\sqrt{\max\{1, N_n^t(x,a)\}}} \\ = \sum_{t=1}^{T} \langle \mu^{\pi,\widehat{p}^t}, b^t \rangle. \end{aligned} \tag{12}$$

By replacing Eq. (12) in Eq. (10), the additive property in the decomposition allows us to cancel out the problematic regret term $R_T^{\text{policy/MDP}}$. As a result, we obtain that $R_T^{\text{policy}} \leqslant \tilde{O}(\sqrt{T}) + \sum_{t=1}^{T} \langle b^t, \mu^{\pi^t,\widehat{p}^t} \rangle$. All that remains is to analyze the new term due to the added bonus, $\sum_{t=1}^{T} \langle b^t, \mu^{\pi^t,\widehat{p}^t} \rangle$, which we do in Prop. 3.1.

**Proposition 3.1.** *Let $(b^t)_{t\in[T]}$ be the bonus vector in Eq. (11). For any $\delta' \in (0,1)$, with probability $1-3\delta'$,*

$$\sum_{t=1}^{T} \langle b^t, \mu^{\pi^t,\widehat{p}^t} \rangle = \tilde{O}\big(LN^3 |\mathcal{X}|^{3/2} \sqrt{|\mathcal{A}|T}\big).$$

With all the ingredients in place, we introduce our new method, *Bonus O-MD-CURL*, in Alg. 1. The main result is in Thm. 3.2 and its proof is in App. B.2. In terms of $T$ and $|\mathcal{A}|$, our result matches the optimal one in RL from (Jin et al., 2020), but we have additional factors of $N$ and $\sqrt{|\mathcal{X}|}$ that are due to using bonuses and to our approach to deal with adversarial convex RL.

**Theorem 3.2.** *Running Alg. 1 for online CURL with unknown transition kernel, full-information feedback, where $F^t := \sum_{n=1}^{N} f_n^t$ is convex and each $f_n^t$ is L-Lipschitz under $\|\cdot\|_1$, ensures that, with probability at least $1-6\delta$ for any $\delta \in (0,1)$, the optimal choice of $\tau$ achieves, for any $\pi \in \Pi$,*

$$R_T(\pi) = \tilde{O}\big(LN^3 |\mathcal{X}|^{3/2} \sqrt{|\mathcal{A}|T}\big).$$

**Algorithm 1** Bonus O-MD-CURL (Full-information)

**Input:** number of episodes $T$, initial policy $\pi^1 \in \Pi$, initial state-action distribution $\mu_0$ and state-action distribution sequence $\mu^1 = \tilde{\mu}^1 = \mu^{\pi^1,\widehat{p}^1}$ with $\widehat{p}_n^1(\cdot|x,a) = 1/|\mathcal{X}|$, learning rate $\tau > 0$.
**Init.:** $\forall (n,x,a,x')$, $N_n^1(x,a) = M_n^1(x'|x,a) = 0$
**for** $t = 1, \dots, T$ **do**
  agent starts at $(x_0^t, a_0^t) \sim \mu_0(\cdot)$
  **for** $n = 1, \dots, N$ **do**
    Env. draws new state $x_n^t \sim p_n(\cdot|x_{n-1}^t, a_{n-1}^t)$
    Update counts

$$N_{n-1}^{t+1}(x_{n-1}^t, a_{n-1}^t) = N_{n-1}^t(x_{n-1}^t, a_{n-1}^t) + 1$$
$$M_{n-1}^{t+1}(x_n^t|x_{n-1}^t, a_{n-1}^t) = M_{n-1}^t(x_n^t|x_{n-1}^t, a_{n-1}^t) + 1$$

    Agent chooses an action $a_n^t \sim \pi_n^t(\cdot|x_n^t)$
  **end for**
  Compute bonus sequence as in Eq. (11)
  Observe objective function $F^t$
  Compute $\mu^{\pi^t,\widehat{p}^t}$ as in Eq. (2)
  Update transition estimate as in Eq. (7)
  Compute the $\pi^{t+1}$ associated to the solution of Eq. 6 with $z^t := -\nabla F^t(\mu^{\pi^t,\widehat{p}^t}) + b^t$ and $q^{t+1} = \widehat{p}^{t+1}$
  Compute $\tilde{\pi}^{t+1}$ (Lem. 2.1), and $\tilde{\mu}^{t+1} := \mu^{\tilde{\pi}^{t+1},\widehat{p}^{t+1}}$
**end for**

## 4. Bandit Feedback

### 4.1. Bandit feedback with bonus in RL

We generalize Alg. 1 to handle the RL case with bandit feedback. Our aim is not to improve the existing algorithms for bandit RL; rather, we show that our new methodology and analysis for CURL achieves comparable results to the SoTA in bandit RL. In this case, an adversary selects a sequence of loss functions $(\ell^t)_{t\in[T]}$, with $\ell^t := (\ell_n^t)_{n\in[N]}$, where $\ell_n^t : \mathcal{X} \times \mathcal{A} \to [0,1]$, and the objective function is given by $F^t(\mu) := \langle \ell^t, \mu \rangle = \sum_{n=1}^{N} \langle \ell_n^t, \mu_n \rangle$. Note that now the gradient of $F^t$ with respect to $\mu$ is always equal to $\ell^t$ due to the linearity of the objective function. Bandit feedback in this setting implies that the learner observes the loss function only for the state-action pairs visited by the agent during each episode, i.e., $(\ell_n^t(x_n^t, a_n^t))_{n\in[N]}$ where $(x_n^t, a_n^t)_{n\in[N]}$ is the agent's trajectory.

We define Alg. 2 in App. D, a version of *Bonus O-MD-CURL* where for each OMD update we take $z^t := \widehat{\ell}^t - b^t$, with $\widehat{\ell}^t$ an importance-weighted estimator of $\ell^t$ defined in Eq. (40) and $b^t$ the bonus vector defined in Eq. (11). Thm. 4.1 states that Alg. 2 achieves the regret bound of $\tilde{O}(\sqrt{T})$ known to be the optimal for RL with bandit feedback (Jin et al., 2020). For the proof and for an overview of approaches for bandit RL see App. D.

**Theorem 4.1.** *Playing Alg. 2 for RL with adversarial losses $(\ell^t)_{t\in[T]}$, unknown transition kernel, and bandit feedback, obtains with high probability for any policy $\pi \in \Pi$,*

$$R_T(\pi) = \tilde{O}\big(N^3 |\mathcal{X}|^{3/2} \sqrt{|\mathcal{A}|T} + N^{3/2} |\mathcal{X}|^{5/4} |\mathcal{A}| \sqrt{T}\big).$$

### 4.2. CURL with bandit feedback

Returning back to the CURL framework, we now assume that $F^t \colon \Delta_{\mathcal{X}\times\mathcal{A}} \to [0,N]$ can be any convex, $L$-Lipschitz function with respect to $\|\cdot\|_1$. In contrast to Sec. 3, we assume here that after executing a policy $\pi^t$ we observe $F^t(\mu^{\pi^t,p})$ instead of $\nabla F^t(\mu^{\pi^t,p})$. We will consider both the case when the MDP is known in advance and when it needs (as in previous sections) to be estimated progressively from observed trajectories.

**Main challenges.** This problem can be broadly categorized as a bandit convex optimization (BCO) problem. This places us in a more challenging domain compared to the

bandit feedback setting in the standard RL problem, where the gradient of the loss function is identical for any point in $(\Delta_{\mathcal{X}\times\mathcal{A}})^N$ and is easier to estimate. Moreover, as a BCO problem, the present setting still exhibits distinctive challenges. One being the peculiar nature of our decision set $\mathcal{M}_{\mu_0}^p$ and how it impedes the efficacy of some standard gradient estimation techniques as we explain below. Another issue arises when the MDP is not known as that induces uncertainty over the true set of permissible occupancy measures. This incomplete knowledge of the decision set is atypical in the BCO literature and introduces multiple sources of bias for any adopted method.

### 4.2.1. Entropic Regularization Method

Our first approach is to extend our MD-based algorithm from Sec. 3, supposing still that the MDP is not known. Since the algorithm required knowledge of the gradient $\nabla F^t(\mu^{\pi^t,p})$, we propose to estimate it by querying $F^t$ at a random perturbation of $\mu^{\pi^t,p}$, a standard approach in the convex bandit literature popularized by Flaxman et al. (2005). This method yields $T^{3/4}$ regret under convex and Lipschitz conditions, and is incapable of doing better (Hu et al., 2016). Although more advanced algorithms and analyses achieve $\sqrt{T}$ regret (Hazan & Li, 2016; Bubeck et al., 2021; Fokkema et al., 2024), they are arguably less practical, more complicated, and have worse dimension dependence. For $d \in \mathbb{Z}_+$, we denote by $\mathbb{B}^d$ and $\mathbb{S}^d$ the unit ball and sphere respectively in $\mathbb{R}^d$, and by $\mathbb{1}_d \in \mathbb{R}^d$ the vector with all entries equal to one. Let $k\colon \mathcal{S} \to \mathbb{R}$ be a convex function, where $\mathcal{S} \subseteq \mathbb{R}^d$ is a convex set satisfying $\mathbb{B}^d \subseteq \mathcal{S}$. Fix some $\delta \in (0,1)$. The approach of Flaxman et al. (2005) relies on the observation that $\frac{(1-\delta)d}{\delta}\mathbb{E}_{\boldsymbol{u}\in\mathbb{S}^d}[k((1-\delta)x+\delta\boldsymbol{u})\boldsymbol{u}] \approx \nabla k(x)$. Hence, $\frac{(1-\delta)d}{\delta}k((1-\delta)x+\delta\boldsymbol{u})\boldsymbol{u}$ (for some $\boldsymbol{u}$ uniformly sampled from $\mathbb{S}^d$) can be used as a one-point stochastic surrogate for the gradient. Applying this idea to our problem presents several challenges. Mainly, $\mathcal{M}_{\mu_0}^p$ has an empty interior in $\mathbb{R}^{N|\mathcal{X}||\mathcal{A}|}$. This can be addressed, assuming for the moment that the kernel $p$ is known, by defining a bijection $\Lambda_p\colon (\mathcal{M}_{\mu_0}^p)^- \to \mathcal{M}_{\mu_0}^p$, where $(\mathcal{M}_{\mu_0}^p)^- \subseteq \mathbb{R}^{N|\mathcal{X}|(|\mathcal{A}|-1)}$ is a representation of the constraint set in a lower-dimensional space where it is possible for its interior to be non-empty, see App. E.1 for more details. Next, we need to specify a (hyper)sphere that is contained in $(\mathcal{M}_{\mu_0}^p)^-$, which would allow us to use the aforementioned spherical estimation technique while remaining inside the feasible set of occupancy measures. To guarantee the existence of such an object, we rely on the following assumption (discussed further below).

**Assumption 4.2.** There exists a value $\varepsilon > 0$ such that $p_n(x'|x,a) \geqslant \varepsilon$ for all $x, x' \in \mathcal{X}^2$, $a \in \mathcal{A}$, and $n \in [N]$.

Under this assumption, we show in App. E.2.1 that for $\kappa := \varepsilon/(|\mathcal{A}|-1+\sqrt{|\mathcal{A}|-1})$, it holds that $\kappa\mathbb{1}_{N|\mathcal{X}|(|\mathcal{A}|-1)} + \kappa\mathbb{B}^{N|\mathcal{X}|(|\mathcal{A}|-1)} \subseteq (\mathcal{M}_{\mu_0}^p)^-$. For any $\boldsymbol{v} \in \mathbb{B}^{N|\mathcal{X}|(|\mathcal{A}|-1)}$, define $\zeta^{\boldsymbol{v},p} := \Lambda_p(\kappa\mathbb{1}_{N|\mathcal{X}|(|\mathcal{A}|-1)} + \kappa\boldsymbol{v})$. Motivated by the preceding discussion, we use (a simple transformation of)

$$\tfrac{1-\delta}{\delta\kappa}N|\mathcal{X}|(|\mathcal{A}|-1)F^t\big((1-\delta)\mu^t + \delta\zeta^{\boldsymbol{u}^t,p}\big)\boldsymbol{u}^t$$

as a surrogate for $\nabla F^t(\mu^t)$, where $\boldsymbol{u}^t$ is sampled uniformly from $\mathbb{S}^{N|\mathcal{X}|(|\mathcal{A}|-1)}$. What remains is to address the issue that the true kernel $p$ is unknown. Similarly to the full information case, we compute an estimate $\widehat{p}^t$ at each round to be used in place of the true kernel, and we employ bonuses to explore. One difference is that we rely on a slightly altered transition kernel estimator (see App. E.2.2) to ensure that $\widehat{p}^t$ too satisfies the condition of Asm. 4.2. Another discrepancy to be accounted for in the analysis is that although we compute $\pi^t$ relying on $\widehat{p}^t$ (in particular, $\pi^t$ is the policy induced by $(1-\delta)\mu^t + \delta\zeta^{\boldsymbol{u}^t,\widehat{p}^t} \in \mathcal{M}_{\mu_0}^{\widehat{p}^t}$), we observe $F^t(\mu^{\pi^t,p})$, the evaluation of $\pi^t$ in the true environment. This induces an extra source of bias in the gradient estimator. We summarize our approach in Alg. 3 in App. E.2.3, and prove the following result in App. E.2.5:

**Theorem 4.3.** *Under Asm. 4.2, Alg. 3 with a suitable tuning of $\tau$, $\delta$, and $(\alpha_t)_{t\in[T]}$ satisfies for any policy $\pi \in \Pi$ that*

$$\mathbb{E}\left[R_T(\pi)\right] = \tilde{\mathcal{O}}\big(\sqrt{L(L+1)/\varepsilon}|\mathcal{X}|^{5/4}|\mathcal{A}|^{5/4}N^3T^{3/4}\big)\,.$$

The main shortcoming of this method is its reliance on the restrictive Asm. 4.2, which also affects the regret guarantee through its dependence on $\varepsilon$. This assumption is not necessary to guarantee that $(\mathcal{M}_{\mu_0}^p)^-$ has a non-empty interior; it suffices instead to assume that every state is reachable at every step, as we do later. Enforcing Asm. 4.2 only serves as a simple way to enable the construction of a sampling sphere with a certain radius. One can construct a different sampling sphere (or ellipsoid) without this assumption; nevertheless, the magnitude of the gradient estimator (which is featured in the current regret bound) would still scale with the reciprocal of the radius of that sphere, the permissible values for which depend on the structure of the MDP and can be arbitrarily small. It seems then that the current approach leads to an inevitable degradation of the bound subject to the structure of the MDP.

### 4.2.2. Self-concordant regularization Method

Fortunately, we can adopt a more principled approach via the use of self-concordant regularization, which is a common technique in bandit convex (and linear) optimization (see, e.g., Abernethy et al., 2008; Saha & Tewari, 2011; Hazan & Levy, 2014), and has been used for online learning in MDPs in different (linear) settings (Lee et al., 2020; Cohen et al., 2021; Van der Hoeven et al., 2023). We show in App. E.3 that $(\mathcal{M}_{\mu_0}^p)^-$ is a convex polytope specified as the intersection of $N|\mathcal{X}||\mathcal{A}|$ half-spaces. We define $\psi_{\text{lb}}\colon (\mathcal{M}_{\mu_0}^p)^- \to \mathbb{R}$ as the standard logarithmic barrier for

$(\mathcal{M}_{\mu_0}^p)^-$ (see Nemirovski, 2004, Cor. 3.1.1) which is a $\vartheta$-self-concordant barrier (see Nemirovski, 2004, Def. 3.1.1) for $(\mathcal{M}_{\mu_0}^p)^-$ with $\vartheta = N|\mathcal{X}||\mathcal{A}|$. The second approach we adopt here is to run mirror descent directly on $(\mathcal{M}_{\mu_0}^p)^-$ as the decision set and take $\psi_{\text{lb}}$ as the regularizer in place of the entropic regularizer that induces $\Gamma$. Let $\xi$ belong to the interior of $(\mathcal{M}_{\mu_0}^p)^-$, which we assume is not empty. Property I in (Nemirovski, 2004, Sec. 2.2) implies that $\xi + (\nabla^2\psi_{\text{lb}}(\xi))^{-1/2}\mathbb{B}^{NS(A-1)} \subseteq (\mathcal{M}_{\mu_0}^p)^-$ . Hence, we can construct an ellipsoid—entirely contained in $(\mathcal{M}_{\mu_0}^p)^-$—around any point in int$(\mathcal{M}_{\mu_0}^p)^-$. Let $\xi^t$ be the output of mirror descent at round $t$ and $U_t := (\nabla^2\psi_{\text{lb}}(\xi_t))^{-1/2}$. We can then use the following as a surrogate for the gradient of $F^t \circ \Lambda_p$ at $\xi^t$ (see also Saha & Tewari, 2011):

$$\tfrac{(1-\delta)}{\delta} N|\mathcal{X}|(|\mathcal{A}|-1)F^t\big(\Lambda_p\big(\xi^t + \delta U_t \boldsymbol{u}^t\big)\big)U_t^{-1}\boldsymbol{u}^t$$

with $\boldsymbol{u}^t$ again sampled uniformly from $\mathbb{S}^{N|\mathcal{X}|(|\mathcal{A}|-1)}$. The eigenvalues of $U_t$ correspond to the lengths of the semi-axes of the ellipsoid used at round $t$, which could be arbitrarily small and lead again to the gradient surrogate having large magnitude. However, thanks to the relationship between $\xi^t$ and $U_t$, a local norm analysis of mirror descent (see, e.g., Lem. 6.16 in Orabona, 2019) absolves the regret of any dependence on the properties of $U_t$. Unfortunately, due to technical barriers, this log-barrier-based approach is not readily extendable to the setting where the decision set can change over time (in particular, it is not clear whether an analogue for Lem. 2.1 can hold in this case). Hence, we restrict its application *only to the case when the MDP is known*, see Alg. 4 in App. E.3. We state next a regret bound for this algorithm (proved in App. E.3.2), which requires the following less restrictive assumption in place of Asm. 4.2.

**Assumption 4.4.** For every state $x \in \mathcal{X}$ and step $n \in [N]$, there exists a policy $\pi$ such that $\sum_{a\in\mathcal{A}} \mu_n^{\pi,p}(x,a) > 0$.

Note that this can be imposed without loss of generality since the MDP is known; defining $\mathcal{X}_n \subseteq \mathcal{X}$ as the subset of states reachable at step $n$, one can represent occupancy measures as sequences of distributions in $(\Delta_{\mathcal{X}_n\times\mathcal{A}})_{n\in[N]}$.

**Theorem 4.5.** *Under Asm. 4.4, Alg. 4 with a suitable tuning of $\tau$ and $\delta$ satisfies for any policy $\pi \in \Pi$ that*

$$\mathbb{E}\left[R_T(\pi)\right] = \tilde{\mathcal{O}}\big(\sqrt{L}N^{7/4}\left(|\mathcal{X}||\mathcal{A}|T\right)^{3/4}\big)\,.$$

Though holding only for the known MDP case, this bound maintains the $T^{3/4}$ rate of Thm. 4.3 while eliminating its reliance on Asm. 4.2 and its undesirable dependence on the MDP's structure. We leave extending this result to unknown MDPs and designing practical approaches enjoying the optimal $\sqrt{T}$ rate for future work.

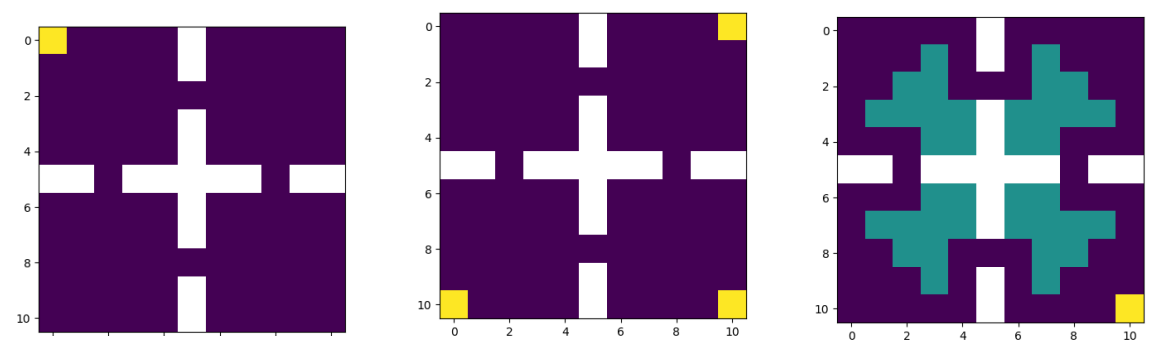

*Figure 1.* [left] Initial agent distribution; [middle] The three targets from *multi-objectives*; [right] The *constrained MDP* (reward in yellow, constraints in blue).

## 5. Experiments

We evaluate Bonus O-MD-CURL on the *multi-objective* and *constrained MDP* tasks from (Geist et al., 2022), which use fixed objective functions and fixed probability kernels across time steps. Adversarial and bandit MDPs are harder to implement due to challenges in finding optimal stationary policies, and there is a a lack of experimental validation in the literature. We focus on evaluating how well the additive bonus helps the algorithm to learn the environment. We also compare it to Greedy MD-CURL from (Moreno et al., 2024). The state space is an $11 \times 11$ four-room grid world, with a single door connecting adjacent rooms. The agent can choose to stay still or move right, left, up, or down, as long as there are no walls blocking the path: $x_{n+1} = x_n + a_n + \varepsilon_n$. The external noise $\varepsilon_n$ is a perturbation that can move the agent to a neighboring state with some probability. The initial distribution is a Dirac delta at the upper left corner of the grid, as in Fig. 1 [left]. We take $N = 40$, $\tau = 0.01$, and 5 repetitions per experiment.[1]

*Multi-objectives:* The goal is to concentrate the distribution on three targets by the final step $N$, as in Fig. 1 [middle]. The objective function is defined as $f_n(\mu_n^{\pi,p}) := -\sum_{k=1}^3(1 - \langle\mu_n^{\pi,p}, e^k\rangle)^2$, where $e^k \in \mathbb{R}^{|\mathcal{X}|}$ is a vector with a 1 at the target state and 0 elsewhere. *Constrained MDPs:* The goal is to concentrate the state distribution on the yellow target in Fig. 1 [right] while avoiding the constraint states in blue. The objective function is defined as $f_n(\mu_n^{\pi,p}) := -\langle r, \mu_n^{\pi,p}\rangle + (\langle\mu_n^{\pi,p}, c\rangle)^2$, where $r, c \in \mathbb{R}_+^{|\mathcal{X}|\times|\mathcal{A}|}$. Here, $r$ and $c$ are zero everywhere except at the target and constraint states respectively.

To compute the regret, we compare against the oracle optimal policy, which can be closely approximated when the dynamics are known. For the **Multi-objective** task, Fig. 2 displays the state distribution at the final time step after 50 iterations for Bonus O-MD-CURL [up, left], and Greedy MD-CURL [up,right], and plot the log-loss [down,left] and regret [down,right] after 1000 iterations. We see that Bonus O-MD-CURL reaches the targets much faster than Greedy MD-CURL. As for the **Constrained MDP** task, Fig. 3 dis-

[1]The code to reproduce the empirical results are available at: https://github.com/biancammoreno/Convex_RL

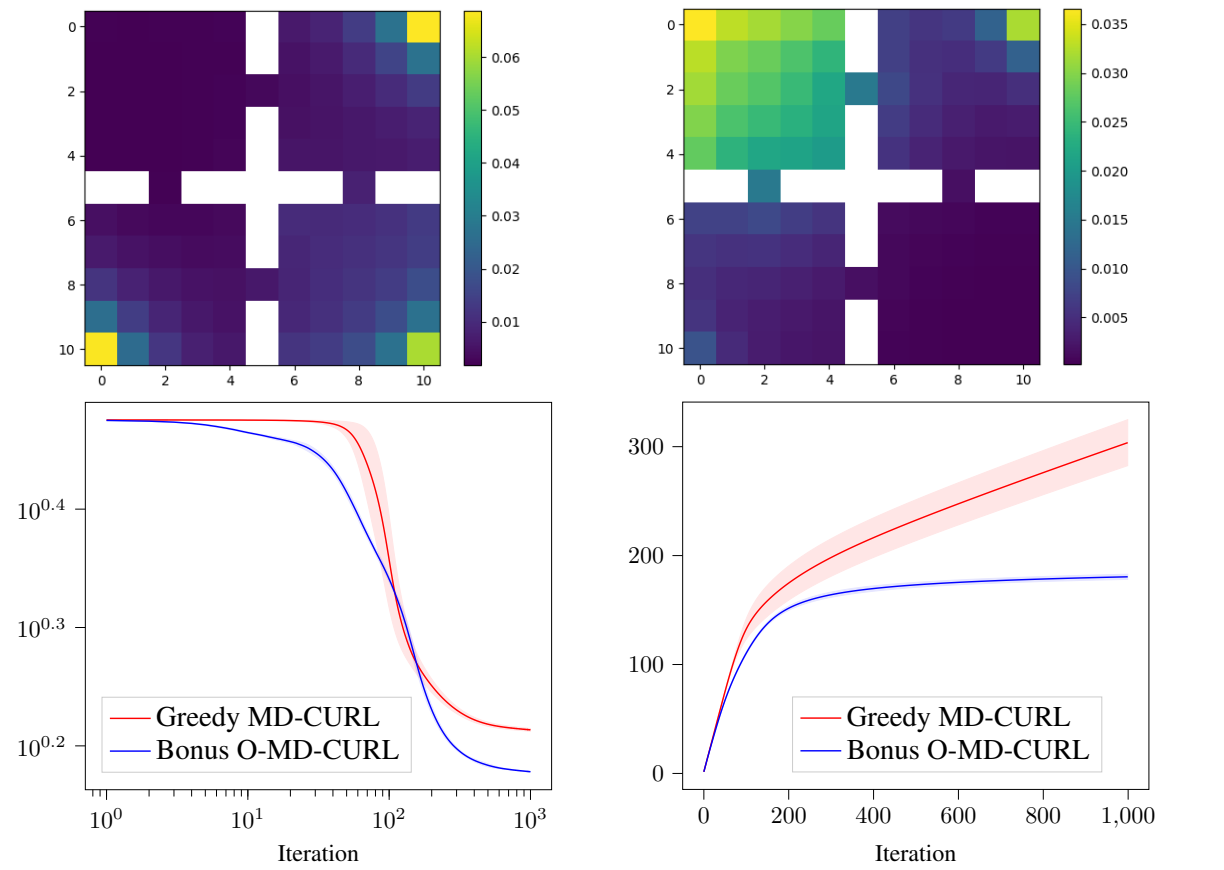

*Figure 2.* Multi-objective: distribution at $N = 40$ after 50 iters. for Bonus O-MD-CURL [up,left], Greedy MD-CURL [up,right]; log-loss [down,left] and regret [down,right] for $10^3$ iters.

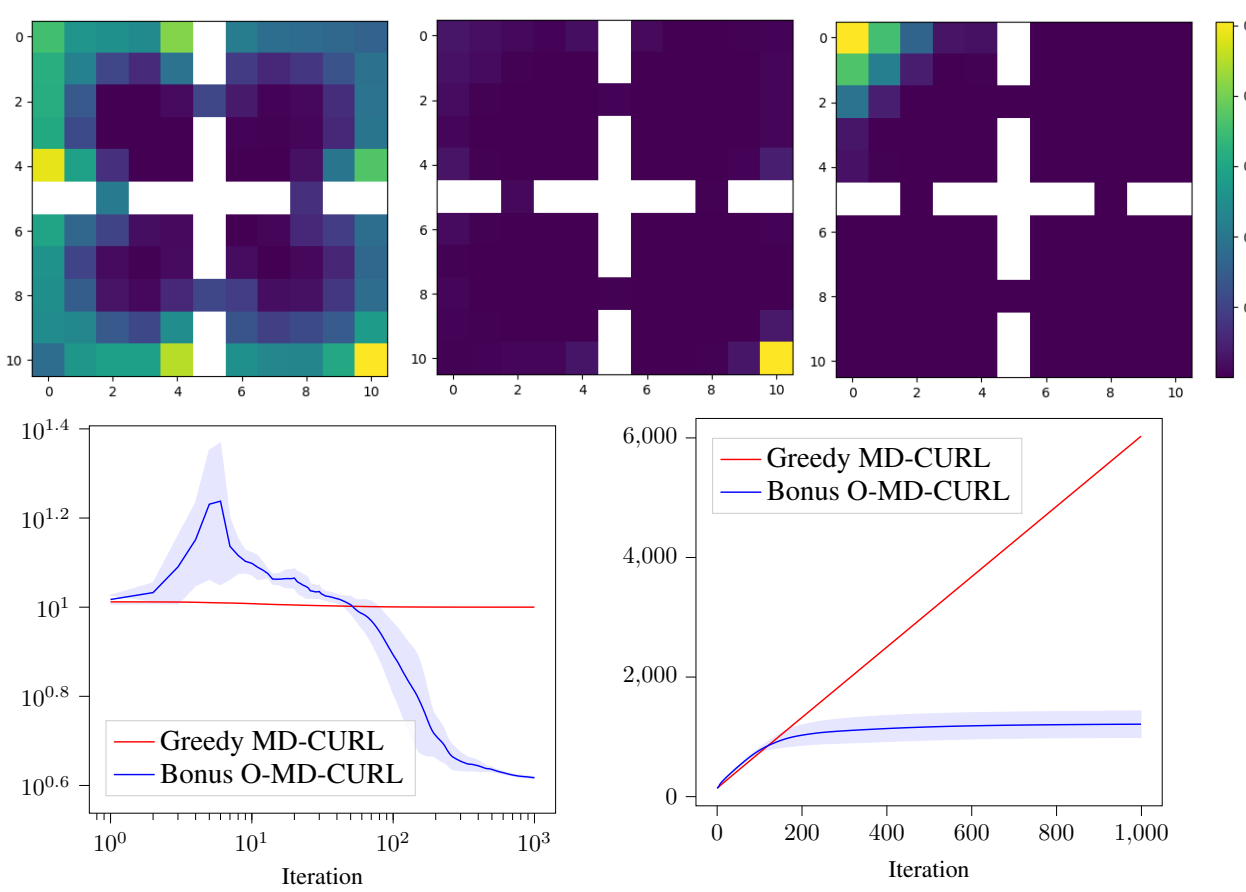

*Figure 3.* Constrained MDP after $10^3$ iters.: sum distributions over all time steps $n \in [40]$ at [up,left]; distribution at the last time step $N = 40$ for Bonus O-MD-CURL [up,center], and Greedy MD-CURL [up,right]; the log-loss [down, left] and regret [down,right].

plays the log-sum of all state distributions for all time steps $n \in [40]$ at iteration 1000 for Bonus O-MD-CURL [up,left]; the state distribution at the last time step $n = 40$ after 1000 iterations for Bonus O-MD-CURL [up,center], and Greedy MD-CURL [up,right]; and the log-loss [down,left] and regret [down,right]. In this case, Greedy MD-CURL fails to reach the target state even after 1000 iterations, while Bonus O-MD-CURL successfully reaches the target state avoiding constrained states to minimize cost thanks to the additive bonuses. These examples empirically demonstrate the value of the additive bonus in tasks requiring exploration.

## Impact Statement

This work is of a theoretical nature, we do not foresee any notable societal consequences.

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

# A. Auxiliary Results

## A.1. Auxiliary lemmas

**Lemma A.1.** *For* $0 < \delta < 1$,

$$\sum_{t=1}^{T}\sum_{n=1}^{N}\sum_{i=0}^{n-1}\sum_{x,a}\mu_i^{\pi^t,p}(x,a)\|p_{i+1}(\cdot|x,a)-\widehat{p}_{i+1}^t(\cdot|x,a)\|_1$$
$$\leqslant 3|\mathcal{X}|N^2\sqrt{2|\mathcal{A}|T\log\left(\frac{|\mathcal{X}||\mathcal{A}|NT}{\delta}\right)}+2|\mathcal{X}|N^2\sqrt{2T\log\left(\frac{N}{\delta}\right)}$$

*with probability at least* $1-2\delta$.

*Proof.* Let $\xi_n^t(x,a) := \|p_n(\cdot|x,a)-\widehat{p}_n^t(\cdot|x,a)\|_1$. We denote by $o^t := (x_n^t, a_n^t)_{n\in[N]}$ the trajectory of the agent at episode $t$ when playing policy $\pi^t$. Let $\widehat{\mu}_n^{\pi^t,p}(x,a) := \mathbb{1}_{\{(x_n^t,a_n^t)=(x,a)\}}$ be the empirical state-action distribution computed from the agent's trajectory. We consider the following decomposition:

$$\sum_{t=1}^{T}\sum_{n=1}^{N}\sum_{i=0}^{n-1}\sum_{x,a}\mu_i^{\pi^t,p}(x,a)\xi_{i+1}^t(x,a) = \underbrace{\sum_{t=1}^{T}\sum_{n=1}^{N}\sum_{i=0}^{n-1}\sum_{x,a}\widehat{\mu}_i^{\pi^t,p}(x,a)\xi_{i+1}^t(x,a)}_{(1)}$$
$$+\underbrace{\sum_{t=1}^{T}\sum_{n=1}^{N}\sum_{i=0}^{n-1}\sum_{x,a}\left(\mu_n^{\pi^t,p}-\widehat{\mu}_i^{\pi^t,p}(x,a)\right)\xi_{i+1}^t(x,a)}_{(2)}.$$

**Term $(1)$ analysis.** We start by analysing the first term. Using Lem. 2.2, we have that for $\delta\in(0,1)$, with probability $1-\delta$,

$$(1)=\sum_{t=1}^{T}\sum_{n=1}^{N}\sum_{i=0}^{n-1}\sum_{x,a}\widehat{\mu}_i^{\pi^t,p}(x,a)\xi_{i+1}^t(x,a)\leqslant\sqrt{2|\mathcal{X}|\log\left(\frac{|\mathcal{X}||\mathcal{A}|NT}{\delta}\right)}\sum_{t=1}^{T}\sum_{n=1}^{N}\sum_{i=0}^{n-1}\sum_{x,a}\frac{\mu_i^{\pi^t,p}(x,a)}{\sqrt{\max\{1,N_i^t(x,a)\}}}.$$

Using Lem. 19 from (Jaksch et al., 2008), we have that for all $i\in[N]$ and $(x,a)\in\mathcal{X}\times\mathcal{A}$,

$$\sum_{t=1}^{T}\frac{\widehat{\mu}_i^{\pi^t,p}(x,a)}{\sqrt{\max\{1,N_i^t(x,a)\}}}\leqslant(\sqrt{2}+1)\sqrt{N_i^T(x,a)}.$$

Therefore, using Jensen's inequality and that $\sum_{(x,a)}N_i^T(x,a)=T$ for all $i\in[N]$, we have that

$$\begin{aligned}\sum_{t=1}^{T}\sum_{n=1}^{N}\sum_{i=0}^{n-1}\sum_{x,a}\frac{\widehat{\mu}_i^{\pi^t,p}(x,a)}{\sqrt{\max\{1,N_i^t(x,a)\}}}&\leqslant 3\sum_{n=1}^{N}\sum_{i=0}^{n-1}\sum_{x,a}\sqrt{N_i^T(x,a)}\\&\leqslant 3\sum_{n=1}^{N}\sum_{i=0}^{n-1}\sqrt{|\mathcal{X}||\mathcal{A}|T}\\&\leqslant 3N^2\sqrt{|\mathcal{X}||\mathcal{A}|T}.\end{aligned}\tag{13}$$

Substituting this inequality into the upper bound for term $(1)$ yields

$$\begin{aligned}(1)&\leqslant\sqrt{2|\mathcal{X}|\log\left(\frac{|\mathcal{X}||\mathcal{A}|NT}{\delta}\right)}3N^2\sqrt{|\mathcal{X}||\mathcal{A}|T}\\&=3|\mathcal{X}|N^2\sqrt{2|\mathcal{A}|T\log\left(\frac{|\mathcal{X}||\mathcal{A}|NT}{\delta}\right)}.\end{aligned}\tag{14}$$

**Term** $(2)$ **analysis.** We now analyse the second term. Let $\mathcal{F}^t := \sigma(o^1, \ldots, o^{t-1})$ be the filtration generated by the trajectories of the agent from the first episode, up to the end of episode $t-1$. Note that $\xi_{n+1}^t(x,a)$ is $\mathcal{F}^t$ measurable, as it only depends on observations up to episode $t-1$. Therefore,

$$\mathbb{E}[\xi_{n+1}^t(x,a)\widehat{\mu}_n^{\pi^t,p}(x,a)|\mathcal{F}_n^t] = \xi_{n+1}^t(x,a)\mathbb{E}[\widehat{\mu}_n^{\pi^t,p}(x,a)|\mathcal{F}_n^t] = \xi_{n+1}^t(x,a)\mu_n^{\pi^t,p}(x,a).$$

For all $n \in [N]$, let $M_n^0 = 0$ and for all $t \in [T]$,

$$M_n^t := \sum_{s=1}^{t}\sum_{x,a}\big(\mu_n^{\pi^s,p}(x,a) - \widehat{\mu}_n^{\pi^s,p}(x,a)\big)\xi_{n+1}^s(x,a).$$

From the observation above, $(M_n^t)_{t\in[T]}$ is a martingale sequence with respect to the filtration $\mathcal{F}^t$. Furthermore, as by definition $|\xi_{n+1}^t(x,a)| \leqslant 2$,

$$|M_n^t - M_n^{t-1}| \leqslant \sum_{x\in\mathcal{X}}\left|\sum_{a\in\mathcal{A}}\big(\mu_n^{\pi^t,p}(x,a) - \widehat{\mu}_n^{\pi^t,p}(x,a)\big)\xi_{n+1}^t(x,a)\right| \leqslant 2|\mathcal{X}|.$$

Therefore, by Azuma-Hoeffding, we have that for any $\varepsilon > 0$,

$$\mathbb{P}\big(M_n^T \geqslant \varepsilon\big) \leqslant \exp\left(\frac{-\varepsilon^2}{8|\mathcal{X}|^2 T}\right).$$

Applying the union bound on all $n \in [N]$, we then have that for any $\delta \in (0,1)$, with probability at least $1-\delta$,

$$M_n^T \leqslant 2|\mathcal{X}|\sqrt{2T\log\left(\frac{N}{\delta}\right)}$$

holds simultaneously for all $n \in [N]$.

Substituting this inequality into term $(2)$ and summing over $n \in [N]$ and $i \in [n-1]$, we obtain, with probability at least $1-\delta$, that

$$(2) = \sum_{n=1}^{N}\sum_{i=0}^{n-1} M_i^T \leqslant 2|\mathcal{X}|N^2\sqrt{2T\log\left(\frac{N}{\delta}\right)}. \tag{15}$$

**Final step.** Combining the upper bounds for term $(1)$ from Eq. (14) and term $(2)$ from Eq. (15), we obtain, with probability at least $1-2\delta$, that

$$\sum_{t=1}^{T}\sum_{n=1}^{N}\sum_{i=0}^{n-1}\sum_{x,a}\mu_i^{\pi^t,p}(x,a)\xi_{i+1}^t(x,a) \leqslant 3|\mathcal{X}|N^2\sqrt{2|\mathcal{A}|T\log\left(\frac{|\mathcal{X}||\mathcal{A}|NT}{\delta}\right)} + 2|\mathcal{X}|N^2\sqrt{2T\log\left(\frac{N}{\delta}\right)},$$

concluding the proof.

□

**Lemma A.2.** *For any* $0 < \delta < 1$,

$$\sum_{t=1}^{T}\sum_{n=0}^{N}(N-n)\sum_{x,a}\frac{\mu_n^{\pi^t,p}(x,a)}{\sqrt{\max\{1, N_n^t(x,a)\}}} \leqslant 3N^2\sqrt{|\mathcal{X}||\mathcal{A}|T} + |\mathcal{X}|N^2\sqrt{2T\log\left(\frac{N}{\delta}\right)},$$

*holds with probability at least* $1-\delta$.

*Proof.* Recall that we denote by $(x_n^t, a_n^t)_{n\in\{0,[N]\}}$ the trajectory of the agent during episode $t$, when playing policy $\pi^t$, and that we define by $\widehat{\mu}^{\pi^t,p}(x,a) := \mathbb{1}_{\{(x_n^t,a_n^t)=(x,a)\}}$ as the empirical state-action distribution computed from the trajectory of

the agent. We consider the following decomposition:

$$\sum_{t=1}^{T}\sum_{n=0}^{N}(N-n)\sum_{x,a}\frac{\mu_n^{\pi^t,p}(x,a)}{\sqrt{\max\{1,N_n^t(x,a)\}}} = \underbrace{\sum_{t=1}^{T}\sum_{n=0}^{N}(N-n)\sum_{x,a}\frac{\widehat{\mu}_n^{\pi^t,p}(x,a)}{\sqrt{\max\{1,N_n^t(x,a)\}}}}_{(1)} + \underbrace{\sum_{t=1}^{T}\sum_{n=0}^{N}(N-n)\sum_{x,a}\frac{\left(\mu_n^{\pi^t,p}-\widehat{\mu}_n^{\pi^t,p}(x,a)\right)}{\sqrt{\max\{1,N_n^t(x,a)\}}}}_{(2)} \tag{16}$$

**Term $(1)$ analysis.** Using the same decomposition of term $(1)$ of Lem. A.1 in Eq. (13) we have that

$$\sum_{t=1}^{T}\sum_{n=0}^{N}(N-n)\sum_{x,a}\frac{\widehat{\mu}_n^{\pi^t,p}(x,a)}{\sqrt{\max\{1,N_n^t(x,a)\}}} \leqslant 3N^2\sqrt{|\mathcal{X}||\mathcal{A}|T}. \tag{17}$$

**Term $(2)$ analysis.** The analysis of term $(2)$ follows a similar approach to the analysis of term $(2)$ in Lem. A.1, with the key difference being that, instead of carrying the term related to the difference between the true probability transition and the estimated one, we now have the term $1/\sqrt{\max\{1,N_n^t(x,a)\}}$.

Let $\mathcal{F}^t := \sigma(o^1,\dots,o^{t-1})$ be the filtration generated by the trajectories of the agent from the first episode, up to the end of episode $t-1$. Note that $1/\sqrt{\max\{1,N_n^t(x,a)\}}$ is $\mathcal{F}^t$ measurable, as it only depends from observations of time step $n$ up to episode $t-1$. Therefore,

$$\mathbb{E}[1/\sqrt{\max\{1,N_n^t(x,a)\}}\widehat{\mu}_n^{\pi^t,p}(x,a)|\mathcal{F}_n^t] = 1/\sqrt{\max\{1,N_n^t(x,a)\}}\mu_n^{\pi^t,p}(x,a).$$

For all $n\in[N]$, let $M_n^0=0$ and for all $t\in[T]$,

$$M_n^t := \sum_{s=1}^{t}(N-n)\sum_{x,a}\left(\mu_n^{\pi^s,p}(x,a)-\widehat{\mu}_n^{\pi^s,p}(x,a)\right)1/\sqrt{\max\{1,N_n^t(x,a)\}}.$$

From the observation above, $(M_n^t)_{t\in[T]}$ is a martingale sequence with respect to the filtration $\mathcal{F}^t$. Furthermore, as by definition $|1/\sqrt{\max\{1,N_n^t(x,a)\}}|\leqslant 1$,

$$|M_n^t - M_n^{t-1}| \leqslant (N-n)\sum_{x\in\mathcal{X}}\left|\sum_{a\in\mathcal{A}}\left(\mu_n^{\pi^t,p}(x,a)-\widehat{\mu}_n^{\pi^t,p}(x,a)\right)\xi_n^t(x,a)\right| \leqslant (N-n)|\mathcal{X}|.$$

Therefore, by Azuma-Hoeffding, we have that for any $\varepsilon>0$,

$$\mathbb{P}\left(M_n^T \geqslant \varepsilon\right) \leqslant \exp\left(\frac{-\varepsilon^2}{2|\mathcal{X}|^2(N-n)^2T}\right).$$

Applying the union bound on all $n\in[N]$, we then have that for any $\delta\in(0,1)$, with probability at least $1-\delta$,

$$M_n^T \leqslant |\mathcal{X}|N\sqrt{2T\log\left(\frac{N}{\delta}\right)}.$$

Summing over $n\in[N]$, we have that with probability at least $1-\delta$,

$$(2) = \sum_{n=0}^{N} M_n^T \leqslant |\mathcal{X}|N^2\sqrt{2T\log\left(\frac{N}{\delta}\right)}. \tag{18}$$

**Joining terms** (1) **and** (2)**.** To conclude, we replace the final upper bounds of the terms (1) and (2) of Eq. (17) and (18) respectively in the decomposition of Eq. (16), and we obtain that, for any $\delta \in (0,1)$, with probability at least $1-\delta$,

$$\sum_{t=1}^{T}\sum_{n=0}^{N}(N-n)\sum_{x,a}\frac{\mu_n^{\pi^t,p}(x,a)}{\sqrt{\max\{1, N_n^t(x,a)\}}} \leqslant 3N^2\sqrt{|\mathcal{X}||\mathcal{A}|T} + |\mathcal{X}|N^2\sqrt{2T\log\left(\frac{N}{\delta}\right)},$$

concluding the proof. □

**Lemma A.3.** *For all $n \in [N]$, $(x,a,x') \in \mathcal{X}\times\mathcal{A}\times\mathcal{X}$, and $t \in [T]$, let $\widehat{p}_{n+1}^t(x'|x,a)$ be defined as in Eq. (7). Hence,*

$$\|\widehat{p}_{n+1}^{t+1}(\cdot|x,a) - \widehat{p}_{n+1}^{t}(\cdot|x,a)\|_1 \leqslant \frac{\mathbb{1}_{\{x_n^t=x, a_n^t=a\}}}{\max\{1, N_n^{t+1}(x,a)\}}.$$

*Proof.* From the definition of the estimator $\widehat{p}^t$, we have that

$$\widehat{p}_{n+1}^{t+1}(x'|x,a) = \frac{1}{\max\{1, N_n^{t+1}(x,a)\}}\big(N_n^t(x,a)\widehat{p}_{n+1}^t(x'|x,a) + \mathbb{1}_{\{x_{n+1}^t=x', x_n^t=x, a_n^t=a\}}\big).$$

Therefore,

$$\begin{aligned}|\widehat{p}_{n+1}^{t+1}(x'|x,a) - \widehat{p}_{n+1}^{t}(x'|x,a)| &= \frac{1}{\max\{1, N_n^{t+1}(x,a)\}}\Big|\mathbb{1}_{\{x_{n+1}^t=x', x_n^t=x, a_n^t=a\}} - \widehat{p}_{n+1}^t(x'|x,a)\big(N_n^{t+1}(x,a) - N_n^t(x,a)\big)\Big| \\ &= \frac{1}{\max\{1, N_n^{t+1}(x,a)\}}\Big|\mathbb{1}_{\{x_{n+1}^t=x', x_n^t=x, a_n^t=a\}} - \widehat{p}_{n+1}^t(x'|x,a)\mathbb{1}_{\{x_n^t=x, a_n^t=a\}}\Big|.\end{aligned}$$

Summing over $x' \in \mathcal{X}$ we then have that

$$\|\widehat{p}_{n+1}^{t+1}(\cdot|x,a) - \widehat{p}_{n+1}^{t}(\cdot|x,a)\|_1 \leqslant \frac{\mathbb{1}_{\{x_n^t=x, a_n^t=a\}}}{\max\{1, N_n^{t+1}(x,a)\}},$$

concluding the proof.

□

**Lemma A.4.** *For $(n,x,a) \in [N]\times\mathcal{X}\times\mathcal{A}$, let $(q^t)_{t\in[T]}$ be a sequence of probability transition kernels with $q^t := (q_n^t)_{n\in[N]}$ such that*

$$\|q_n^{t+1}(\cdot|x,a) - q_n^t(\cdot|x,a)\|_1 \leqslant \frac{c\mathbb{1}_{\{x_{n-1}^t=x, a_{n-1}^t=a\}}}{\max\{1, N_{n-1}^{t+1}(x,a)\}}$$

*for some constant $c > 0$. Then,*

$$\sum_{t=1}^{T}\|q_n^{t+1}(\cdot|x,a) - q_n^t(\cdot|x,a)\|_1 \leqslant ec\log(T)\,.$$

*Proof.* We have that

$$\sum_{t=1}^{T}\|q_n^{t+1}(\cdot|x,a) - q_n^t(\cdot|x,a)\|_1 \leqslant c\sum_{t=1}^{T}\frac{\mathbb{1}_{\{x_{n-1}^t=x, a_{n-1}^t=a\}}}{\max\{1, N_{n-1}^{t+1}(x,a)\}} = c\sum_{t=1}^{N_{n-1}^{T+1}(x,a)}\frac{1}{t} \leqslant c\sum_{t=1}^{T}\frac{1}{t} \leqslant c\log(eT) \overset{T\geqslant 2}{\leqslant} ec\log(T)\,.$$

□

**Lemma A.5.** *Let $(q^t)_{t\in[T]}$ be a sequence of probability transition kernels, i.e., $q^t := (q_n^t)_{n\in[N]}$ such that for any state-action pair $(x,a)$ and any step $n \in [N]$, $\sum_{t=1}^T \|q_n^{t+1}(\cdot|x,a) - q_n^t(\cdot|x,a)\|_1 \leqslant c\log(T)$ for some constant $c > 0$. Then, for any sequence of policies $(\pi^t)_{t\in[T]}$,*

$$\sum_{t=1}^{T}\|\mu^{\pi^t,q^{t+1}} - \mu^{\pi^t,q^t}\|_{\infty,1} \leqslant c|\mathcal{X}||\mathcal{A}|N\log(T)\,.$$

*While for a fixed policy $\pi$,*

$$\sum_{t=1}^{T}\|\mu^{\pi,q^{t+1}} - \mu^{\pi,q^t}\|_{\infty,1} \leqslant c|\mathcal{X}|N\log(T)\,.$$

*Proof.* Using Lem. B.1 we obtain that

$$\begin{aligned}\sum_{t=1}^{T} \|\mu^{\pi^t,q^{t+1}} - \mu^{\pi^t,q^t}\|_{\infty,1} &\leqslant \sum_{t=1}^{T} \sup_{n\in[N]} \sum_{i=0}^{n-1} \sum_{x,a} \mu_i^{\pi^t,q^t}(x,a) \|q_{i+1}^{t+1}(\cdot|x,a) - q_{i+1}^{t}(\cdot|x,a)\|_1 \\ &= \sum_{t=1}^{T} \sum_{n=0}^{N-1} \sum_{x,a} \mu_n^{\pi^t,q^t}(x,a) \|q_{n+1}^{t+1}(\cdot|x,a) - q_{n+1}^{t}(\cdot|x,a)\|_1 \\ &\leqslant \sum_{n=0}^{N-1} \sum_{x,a} \sum_{t=1}^{T} \|q_{n+1}^{t+1}(\cdot|x,a) - q_{n+1}^{t}(\cdot|x,a)\|_1 \leqslant c|\mathcal{X}||\mathcal{A}|N\log(T)\,.\end{aligned}$$

While for a fixed policy $\pi$,

$$\begin{aligned}\sum_{t=1}^{T} \|\mu^{\pi,q^{t+1}} - \mu^{\pi,q^t}\|_{\infty,1} &\leqslant \sum_{t=1}^{T} \sum_{n=0}^{N-1} \sum_{x,a} \mu_n^{\pi,q^t}(x,a) \|q_{n+1}^{t+1}(\cdot|x,a) - q_{n+1}^{t}(\cdot|x,a)\|_1 \\ &\leqslant \sum_{t=1}^{T} \sum_{n=0}^{N-1} \sum_{x,a} \pi_n(a|x) \|q_{n+1}^{t+1}(\cdot|x,a) - q_{n+1}^{t}(\cdot|x,a)\|_1 \\ &\leqslant c \sum_{n=0}^{N-1} \sum_{x,a} \pi_n(a|x)\log(T) = c|\mathcal{X}|N\log(T)\,.\end{aligned}$$

□

**Lemma A.6.** *Consider a sequence of policies $(\pi^t)_{t\in[T]}$, and define a smoothed version of each policy $\tilde{\pi}^t$ for all $t \in [T]$ as $\tilde{\pi}^t := (1-\alpha_t)\pi^t + \frac{\alpha_t}{|\mathcal{A}|}$, where $\alpha_t \in (0,1)$. Let $p$ and $q$ be two probability transition kernels, denoted as $p := (p_n)_{n\in[N]}$ and $q := (q_n)_{n\in[N]}$, respectively. Therefore, for all $t \in [T]$,*

$$\|\mu^{\pi^t,p} - \mu^{\tilde{\pi}^t,q}\|_{\infty,1} \leqslant \sum_{i=0}^{N-1} \sum_{x,a} \mu_i^{\pi^t,p}(x,a) \|p_{i+1}(\cdot|x,a) - q_{i+1}(\cdot|x,a)\|_1 + 2N\alpha_t.$$

*Proof.* See Lem. D.4 from (Moreno et al., 2024). □

### A.2. Building a closed-form solution for each OMD iteration

In this subsection we argue that the MD optimization problem solved at each iteration in Lem. 2.1 has a closed-form solution. Define the convex function $G^t(\mu) := \tau\langle z^t, \mu\rangle + \Gamma(\mu, \tilde{\mu}^t)$, for $\tau > 0$.

Optimizing a convex objective function over policies is equivalent to optimizing it over state-action distributions in $\mathcal{M}_{\mu_0}^p$. Therefore, the optimization problem solved in Lem. 2.1 over the state-action distributions induced by $q^{t+1}$ is equivalent to minimizing the same function over the space of policies:

$$\underbrace{\min_{\mu\in\mathcal{M}_{\mu_0}^{q^{t+1}}} G^t(\mu)}_{(i)\text{: state-action problem}} \equiv \underbrace{\min_{\pi\in(\Delta_{\mathcal{A}})^{\mathcal{X}\times N}} G^t(\mu^{\pi,q^{t+1}})}_{(ii)\text{: policy problem}}\,. \tag{19}$$

In Thm. 4.1 of (Moreno et al., 2024), it is shown that for each episode $t \in [T]$, an optimal policy for the problem

$$\min_{\pi\in(\Delta_{\mathcal{A}})^{\mathcal{X}\times N}} G^t(\mu^{\pi,q^{t+1}}) := \tau\langle z^t, \mu\rangle + \Gamma(\mu, \tilde{\mu}^t), \tag{20}$$

defined in Eq. (19), denoted by $\pi^{t+1}$, can be computed using an auxiliary sequence of functions $(\tilde{Q}_n^t)_{n\in[N]}$, where $\tilde{Q}_n^t : \mathcal{X}\times\mathcal{A} \to \mathbb{R}$. The sequence starts with $Q_N^t(x,a) = -z_N^t(x,a)$, and for $n \in \{N,\ldots,1\}$, the following recursion is

used:

$$\pi_{n+1}^{t+1}(a|x) = \frac{\tilde{\pi}_{n+1}^{t}(a|x)\exp\big(\tau \tilde{Q}_{n+1}^{t}(x,a)\big)}{\sum_{a'} \tilde{\pi}_{n+1}^{t}(a'|x)\exp\big(\tau \tilde{Q}_{n+1}^{t}(x,a')\big)},$$

$$\tilde{Q}_{n}^{t}(x,a) = -z_n^t(x,a) + \sum_{x'} q_{n+1}^{t+1}(x'|x,a)\sum_{a'} \pi_{n+1}^{t+1}(a'|x')\Big[-\frac{1}{\tau}\log\Big(\frac{\pi_{n+1}^{t+1}(a'|x')}{\tilde{\pi}_{n+1}^{t}(a'|x')}\Big) + \tilde{Q}_{n+1}^{t}(x',a')\Big].$$

The core idea of the proof is to show that, due to the specific divergence used (defined in Eq. (4)), Eq. (20) can be solved using dynamic programming. For further details, the reader is referred to Appendix B of (Moreno et al., 2024). A similar result was also obtained by (Cammardella et al., 2023), though they approached the optimization problem using Lagrangian multipliers instead of dynamic programming.

Problem $(i)$ of Eq. (19) is convex, and the theoretical analysis are given in Lem. 2.1. Thanks to the equivalence between problems $(i)$ and $(ii)$ in Eq. (19), we can use the analysis of problem $(i)$ to provide theoretical guarantees for the closed-form solution policy of problem $(ii)$.

## B. Missing Results and Proofs

**Lemma B.1.** *For any strategy $\pi \in (\Delta_{\mathcal{A}})^{\mathcal{X}\times N}$, for any two probability kernels $p = (p_n)_{n\in[N]}$ and $q = (q_n)_{n\in[N]}$ such that $p_n, q_n : \mathcal{X}\times\mathcal{A}\times\mathcal{X} \to [0,1]$, and $n \in [N]$,*

$$\|\mu_n^{\pi,p} - \mu_n^{\pi,q}\|_1 \leqslant \sum_{i=0}^{n-1}\sum_{x,a}\mu_i^{\pi,p}(x,a)\|p_{i+1}(\cdot|x,a) - q_{i+1}(\cdot|x,a)\|_1.$$

*Proof.* From the definition of a state-action distribution sequence induced by a policy $\pi$ in a probability kernel $p$ in Eq. (2), we have that for all $(x,a) \in \mathcal{X}\times\mathcal{A}$ and $n \in [N]$,

$$\mu_n^{\pi,p}(x,a) = \sum_{x',a'}\mu_{n-1}^{\pi,p}(x',a')p_n(x|x',a')\pi_n(a|x).$$

Thus,

$$\begin{aligned}
\|\mu_n^{\pi,p} - \mu_n^{\pi,q}\|_1 &= \sum_{x,a}\big|\mu_n^{\pi,p}(x,a) - \mu_n^{\pi,q}(x,a)\big| \\
&= \sum_{x,a}\sum_{x',a'}\big|\mu_{n-1}^{\pi,p}(x',a')p_n(x|x',a') - \mu_{n-1}^{\pi,q}(x',a')q_n(x|x',a')\big|\pi_n(a|x) \\
&= \sum_{x}\sum_{x',a'}\big|\mu_{n-1}^{\pi,p}(x',a')p_n(x|x',a') - \mu_{n-1}^{\pi,q}(x',a')q_n(x|x',a')\big| \\
&= \sum_{x}\sum_{x',a'}\big|\mu_{n-1}^{\pi,p}(x',a')p_n(x|x',a') - \mu_{n-1}^{\pi,p}(x',a')q_n(x|x',a') \\
&\qquad + \mu_{n-1}^{\pi,p}(x',a')q_n(x|x',a') - \mu_{n-1}^{\pi,q}(x',a')q_n(x|x',a')\big| \\
&\leqslant \sum_{x',a'}\mu_{n-1}^{\pi,p}(x',a')\|p_n(\cdot|x',a') - q_n(\cdot|x',a')\|_1 + \sum_{x',a'}\big|\mu_{n-1}^{\pi,p}(x',a') - \mu_{n-1}^{\pi,q}(x',a')\big| \\
&= \sum_{x',a'}\mu_{n-1}^{\pi,p}(x',a')\|p_n(\cdot|x',a') - q_n(\cdot|x',a')\|_1 + \|\mu_{n-1}^{\pi,p} - \mu_{n-1}^{\pi,q}\|_1.
\end{aligned}$$

Since for $n=0$, $\|\mu_0^{\pi,p} - \mu_0^{\pi,q}\|_1 = 0$, by induction we get that

$$\|\mu_n^{\pi,p} - \mu_n^{\pi,q}\|_1 \leqslant \sum_{i=0}^{n-1}\sum_{x',a'}\mu_i^{\pi,p}(x',a')\|p_{i+1}(\cdot|x',a') - q_{i+1}(\cdot|x',a')\|_1.$$

□

### B.1. Proof of Prop. 3.1

*Proof.* In the analysis, we explicitly write the term $n = 0$ separately from the other $n \in [N]$. We begin with the following decomposition:

$$\sum_{t=1}^{T}\langle b^t, \mu^{\pi^t,\widehat{p}^t}\rangle + \sum_{t=1}^{T}\langle b_0^t, \mu_0\rangle = \underbrace{\sum_{t=1}^{T}\langle b^t, \mu^{\pi^t,\widehat{p}^t} - \mu^{\pi^t,p}\rangle}_{(1)} + \underbrace{\sum_{t=1}^{T}\langle b^t, \mu^{\pi^t,p}\rangle + \sum_{t=1}^{T}\langle b_0^t, \mu_0\rangle}_{(2)}.$$

**Term $(1)$ analysis.** Using Holder's inequality, we have that

$$(1) \leqslant \sum_{t=1}^{T}\sum_{n=1}^{N}\|b_n^t\|_\infty\|\mu_n^{\pi^t,\widehat{p}^t} - \mu_n^{\pi^t,p}\|_1.$$

From the definition of the bonus sequence, we have that for all $n \in [N]$, $\|b_n^t\|_\infty \leqslant L(N-n)C_\delta$. Hence,

$$\begin{aligned}
(1) &\leqslant LC_\delta \sum_{t=1}^{T}\sum_{n=1}^{N}(N-n)\sum_{i=0}^{n-1}\sum_{x,a}\mu_i^{\pi^t,p}(x,a)\|p_{i+1}(\cdot|x,a) - \widehat{p}_{i+1}^t(\cdot|x,a)\|_1 \\
&\leqslant LC_\delta|\mathcal{X}|N^3\left[3\sqrt{2|\mathcal{A}|T\log\left(\frac{|\mathcal{X}||\mathcal{A}|NT}{\delta}\right)} + 2\sqrt{2T\log\left(\frac{N}{\delta}\right)}\right]
\end{aligned}$$

where the first inequality comes from Lem. B.1, and the second inequality is achieved for any $\delta \in (0,1)$, with probability at least $1-2\delta$, using Lem. A.1.

**Term $(2)$ analysis.** Using the definition of the bonus sequence in equation (11), and recalling that the initial state-action distribution $\mu_0$ is always the same, we have that, for any $\delta \in (0,1)$, with probability at least $1-\delta$,

$$\begin{aligned}
(2) &= LC_\delta\sum_{t=1}^{T}\sum_{n=0}^{N}(N-n)\sum_{x,a}\frac{\mu_n^{\pi^t,p}(x,a)}{\sqrt{\max\{1, N_n^t(x,a)\}}} \\
&\leqslant LC_\delta N^2\left[3\sqrt{|\mathcal{X}||\mathcal{A}|T} + |\mathcal{X}|\sqrt{2T\log\left(\frac{N}{\delta}\right)}\right],
\end{aligned}$$

where the inequality comes from Lem. A.2.

**Joining the upper bounds in term $(1)$ and $(2)$.** Putting both upper bounds together we get that for any $\delta \in (0,1)$, with probability at least $1-3\delta$, and from the definition of $C_\delta$,

$$\begin{aligned}
\sum_{t=1}^{T}\langle b^t, \mu^{\pi^t,\widehat{p}^t}\rangle + \sum_{t=1}^{T}\langle b_0^t, \mu_0\rangle &\leqslant LC_\delta|\mathcal{X}|N^3\left[3\sqrt{2|\mathcal{A}|T\log\left(\frac{|\mathcal{X}||\mathcal{A}|NT}{\delta}\right)} + 2\sqrt{2T\log\left(\frac{N}{\delta}\right)}\right] \\
&\quad + LC_\delta N^2\left[3\sqrt{|\mathcal{X}||\mathcal{A}|T} + |\mathcal{X}|\sqrt{2T\log\left(\frac{N}{\delta}\right)}\right] \\
&= O\left(LN^3|\mathcal{X}|^{3/2}\sqrt{|\mathcal{A}|T}\log\left(\frac{|\mathcal{X}||\mathcal{A}|NT}{\delta}\right)\right).
\end{aligned}$$

□

### B.2. Proof of Thm. 3.2 (Main result)

For proving the main result we join together all the pieces we presented in the main paper and the appendix.

*Proof.* We start by decomposing the regret and using the convexity of the objective function obtaining that

$$
\begin{aligned}
R_T(\pi) &= \sum_{t=1}^T F^t(\mu^{\pi^t,p}) - F^t(\mu^{\pi^t,\widehat{p}^t}) + \sum_{t=1}^T F^t(\mu^{\pi^t,\widehat{p}^t}) - F^t(\mu^{\pi,p}) \\
&\leqslant \underbrace{\sum_{t=1}^T \langle \nabla F^t(\mu^{\pi^t,p}), \mu^{\pi^t,p} - \mu^{\pi^t,\widehat{p}^t}\rangle}_{R_T^{\text{MDP}}} + \underbrace{\sum_{t=1}^T \langle \nabla F^t(\mu^{\pi^t,\widehat{p}^t}), \mu^{\pi^t,\widehat{p}^t} - \mu^{\pi,p}\rangle}_{R_T^{\text{policy}}}.
\end{aligned}
$$

We analyse each term separately:

**Analysis of $R_T^{\text{MDP}}$.** We begin by analyzing the term $R_T^{\text{MDP}}$, which represents the cost incurred due to not knowing the true probability kernel. First, we apply Hoeffding's inequality, and the fact that $f_n^t$ is $L$-Lipschitz with respect to the norm $\|\cdot\|_1$. Following, we apply Lem. B.1, obtaining that

$$
\begin{aligned}
R_T^{\text{MDP}} &\leqslant \sum_{t=1}^T \sum_{n=1}^N \|\nabla f_n^t(\mu^{\pi^t,p})\|_\infty \|\mu_n^{\pi^t,p} - \mu_n^{\pi^t,\widehat{p}^t}\|_1 \\
&\leqslant L \sum_{t=1}^T \sum_{n=1}^N \sum_{i=0}^{n-1} \sum_{x,a} \mu_i^{\pi^t,p}(x,a) \|p_{i+1}(\cdot|x,a) - \widehat{p}_{i+1}^t(\cdot|x,a)\|_1.
\end{aligned}
$$

We can now apply Lem. A.1 to obtain that for any $\delta \in (0,1)$, with probability at least $1-2\delta$,

$$
R_T^{\text{MDP}} \leqslant L3|\mathcal{X}|N^2 \sqrt{2|\mathcal{A}|T \log\left(\frac{|\mathcal{X}||\mathcal{A}|NT}{\delta}\right)} + L2|\mathcal{X}|N^2\sqrt{2T\log\left(\frac{N}{\delta}\right)}. \tag{21}
$$

**Analysis of $R_T^{\text{policy}}$.** To analyse $R_T^{\text{policy}}$ we further decompose it as

$$
R_T^{\text{policy}} = \underbrace{\sum_{t=1}^T \langle \nabla F^t(\mu^{\pi^t,\widehat{p}^t}) - b^t, \mu^{\pi^t,\widehat{p}^t} - \mu^{\pi,\widehat{p}^t}\rangle}_{R_T^{\text{policy/MD}}} + \underbrace{\sum_{t=1}^T \langle b^t, \mu^{\pi^t,\widehat{p}^t} - \mu^{\pi,\widehat{p}^t}\rangle + \sum_{t=1}^T \langle \nabla F^t(\mu^{\pi^t,\widehat{p}^t}), \mu^{\pi,\widehat{p}^t} - \mu^{\pi,p}\rangle}_{R_T^{\text{policy/bonus}}},
$$

where recall that $b^t := (b_n^t)_{n\in[N]}$ is the bonus vector defined in Eq. (11).

**Analysis of $R_T^{\text{policy/MD}}$.** We begin by addressing the term that accounts for the regret incurred by using online Mirror Descent with changing constraint sets.

From Lemmas A.3 and A.4, we know that the probability sequence $(\widehat{p}^t)_{t\in[T]}$ satisfies the condition that $\sum_{t=1}^T \|\widehat{p}_n^{t+1} - \widehat{p}_n^t\|_1 \leqslant c\log(T)$ for $c = e$. Additionally, at each time step $t$, since $F^t$ is $L_F$-Lipschitz with respect to the norm $\|\cdot\|_{\infty,1}$, we have $\|\nabla F^t(\mu)\|_{1,\infty} \leqslant L_F = LN$ for any state-action distribution $\mu$. From the definition of the bonus vector, we also have that $\|b^t\|_{1,\infty} \leqslant LN^2C_\delta$. Consequently, $\|\nabla F^t(\mu) - b^t\|_{1,\infty} \leqslant 2LN^2C_\delta$. Therefore, as we compute $\mu^{t+1}$ by solving

$$
\mu^{t+1} := \underset{\mu\in\mathcal{M}_{\mu_0}^{\widehat{p}^{t+1}}}{\arg\min} \Big\{\tau\langle \nabla F^t(\mu^{\pi^t,\widehat{p}^t}) - b^t, \mu\rangle + \Gamma(\mu, \tilde{\mu}^t)\Big\},
$$

by applying Lem. 2.1 with $\nu^t := \mu^{\pi,\widehat{p}^t}$, $\zeta = 2LN^2C_\delta$, and the sequence of probability transition kernels $(\widehat{p}^t)_{t\in[T]}$, we obtain that for the optimal parameter $\tau = \sqrt{\frac{b}{\zeta^2 T}}$, where

$$
b := N\bigg(\log(T)\Big(e|\mathcal{X}||\mathcal{A}| + 4\Big) + \log(|\mathcal{A}|) + 2Ne|\mathcal{X}|\log(T)^2\log(|\mathcal{A}|)\bigg),
$$

and recalling that $C_\delta := \sqrt{2|\mathcal{X}|\log\big(|\mathcal{X}||\mathcal{A}|NT/\delta\big)}$,

$$
\begin{aligned}
R_T^{\text{policy/MD}} &\leqslant 2\zeta\sqrt{bT} + \zeta N e|\mathcal{X}|\log(T) \\
&\leqslant 2LN^2\sqrt{2|\mathcal{X}|\log\left(\frac{|\mathcal{X}||\mathcal{A}|TN}{\delta}\right)}\big(2\sqrt{bT} + N\log(T)e|\mathcal{X}|\big) \\
&= O\bigg(LN^2|\mathcal{X}|\sqrt{T\log\left(\frac{|\mathcal{X}||\mathcal{A}|NT}{\delta}\right)}\Big(\sqrt{N|\mathcal{A}|\log(T)} + N\sqrt{\log(|\mathcal{A}|)}\log(T)\Big)\bigg).
\end{aligned}
\tag{22}
$$

**Analysis of $R_T^{\text{policy/bonus}}$.** We start by analysing the second term of the sum in $R_T^{\text{policy/bonus}}$. For any $\delta \in (0,1)$, with probability at least $1-\delta$, we have that

$$
\begin{aligned}
\sum_{t=1}^T \langle \nabla F^t(\mu^{\pi^t,\widehat{p}^t}), \mu^{\pi,\widehat{p}^t} - \mu^{\pi,p}\rangle &\leqslant \sum_{t=1}^T\sum_{n=1}^N \|\nabla f_n^t(\mu^{\pi,p})\|_\infty \|\mu_n^{\pi,p} - \mu_n^{\pi,\widehat{p}^t}\|_1 \\
&\leqslant L\sum_{t=1}^T\sum_{n=1}^N\sum_{i=0}^{n-1}\sum_{x,a}\mu_i^{\pi,\widehat{p}^t}(x,a)\|p_{i+1}(\cdot|x,a) - \widehat{p}_{i+1}^t(\cdot|x,a)\|_1 \\
&\leqslant L\sum_{t=1}^T\sum_{n=1}^N\sum_{i=0}^{n-1}\sum_{x,a}\mu_i^{\pi,\widehat{p}^t}(x,a)\frac{C_\delta}{\sqrt{\max\{1, N_i^t(x,a)\}}} \\
&= L\sum_{t=1}^T\sum_{n=0}^{N-1}(N-n)\sum_{x,a}\mu_n^{\pi,\widehat{p}^t}(x,a)\frac{C_\delta}{\sqrt{\max\{1, N_n^t(x,a)\}}} \\
&= \sum_{t=1}^T \langle b^t, \mu^{\pi,\widehat{p}^t}\rangle + \sum_{t=1}^T\langle b_0^t, \mu_0\rangle
\end{aligned}
$$

where the first inequality follows from Holder's inequality, the second from the fact that $f_n^t$ is $L$-Lipschitz with respect to the norm $\|\cdot\|_1$ and Lem. B.1, the third from the concentration bound in Lem. 2.2 where we define $C_\delta := \sqrt{2|\mathcal{X}|\log\big(|\mathcal{X}||\mathcal{A}|NT/\delta\big)}$, and the last equality comes from the definition of the bonus vector in Eq. (11).

Replacing it at the $R_T^{\text{policy/bonus}}$ term we have that

$$
\begin{aligned}
R_T^{\text{policy/bonus}} &= \sum_{t=1}^T\langle b^t, \mu^{\pi^t,\widehat{p}^t} - \mu^{\pi,\widehat{p}^t}\rangle + \sum_{t=1}^T\langle \nabla F^t(\mu^{\pi^t,\widehat{p}^t}), \mu^{\pi,\widehat{p}^t} - \mu^{\pi,p}\rangle \\
&\leqslant \sum_{t=1}^T\langle b^t, \mu^{\pi^t,\widehat{p}^t} - \mu^{\pi,\widehat{p}^t}\rangle + \sum_{t=1}^T\langle b^t, \mu^{\pi,\widehat{p}^t}\rangle + \sum_{t=1}^T\langle b_0^t, \mu_0\rangle \\
&\leqslant \sum_{t=1}^T\langle b^t, \mu^{\pi^t,\widehat{p}^t}\rangle + \sum_{t=1}^T\langle b_0^t, \mu_0\rangle.
\end{aligned}
$$

Lastly, we apply Prop. 3.1 to achieve that, for any $\delta \in (0,1)$, with probability at least $1-4\delta$,

$$
R_T^{\text{policy/bonus}} \leqslant \sum_{t=1}^T\langle b^t, \mu^{\pi^t,\widehat{p}^t}\rangle + \sum_{t=1}^T\langle b_0^t, \mu_0\rangle = O\bigg(LN^3|\mathcal{X}|^{3/2}\sqrt{|\mathcal{A}|T}\log\left(\frac{|\mathcal{X}||\mathcal{A}|NT}{\delta}\right)\bigg). \tag{23}
$$

**Final upper bound on $R_T^{\text{policy}}$.** Joining the upper bounds on $R_T^{\text{policy/MD}}$ and $R_T^{\text{policy/bonus}}$ from Eq.s (22) and (23) respectively, we achieve that for any $\delta \in (0,1)$, with probability at least $1-4\delta$, ignoring logarithmic terms,

$$
R_T^{\text{policy}} \leqslant \tilde{O}\big(LN^3|\mathcal{X}|^{3/2}\sqrt{|\mathcal{A}|T}\big). \tag{24}
$$

**Joining the upper bounds on $R_T^{\text{MDP}}$ and on $R_T^{\text{policy}}$.** Note that the terms in the upper bound on $R_T^{\text{policy}}$ from Eq. (24) dominate those in the upper bound on $R_T^{\text{MDP}}$ from Eq. (21). Therefore, when combining both terms to complete the upper bound on the regret, we obtain that, with probability $1-6\delta$,

$$R_T(\pi) \leqslant \tilde{O}\big(LN^3|\mathcal{X}|^{3/2}\sqrt{|\mathcal{A}|T}\big),$$

concluding the proof.

□

## C. Proof of Lem. 2.1: Online Mirror Descent with Varying Constraint Sets

Before stating the proof we recall a few results from Bregman divergences and in particular the divergence $\Gamma$ defined in Eq. (4) that are used throughout the proof.

To simplify notation, for any probability measure $\eta \in \Delta_E$, where $E$ is any finite space, we define the neg-entropy function, using the convention that $0\log(0) = 0$, as $\phi(\eta) := \sum_{x\in E} \eta(x)\log\eta(x)$. For any $\mu := (\mu_n)_{n\in[N]} \in (\Delta_{\mathcal{X}\times\mathcal{A}})^N$, we define $\rho_n(x) := \sum_{a\in\mathcal{A}} \mu_n(x,a)$ for all $n \in [N]$ and $x \in \mathcal{X}$, representing the marginal distribution over the state space. The function inducing the divergence $\Gamma$, defined in Eq. (4), is given by

$$\psi(\mu) := \sum_{n=1}^{N} \phi(\mu_n) - \sum_{n=1}^{N} \phi(\rho_n). \tag{25}$$

By definition of a Bregman divergence, for any two probability transition kernels $p, q$, for all $\mu \in \mathcal{M}_{\mu_0}^{p}$ and $\mu' \in \mathcal{M}_{\mu_0}^{q,*}$, where $\mathcal{M}_{\mu_0}^{q,*}$ is the subset of $\mathcal{M}_{\mu_0}^{q}$ where the corresponding policies $\pi$ satisfy $\pi_n(a|x) \neq 0$, we then have that

$$\Gamma(\mu,\mu') := \psi(\mu) - \psi(\mu') - \langle\nabla\psi(\mu'), \mu-\mu'\rangle. \tag{26}$$

Additionally, for any probability transition kernel $p$, the function $\psi$ is 1-strongly convex with respect to $\|\cdot\|_{\infty,1}$ within $\mathcal{M}_{\mu_0}^{p}$ (see Thm. 4.1 from (Moreno et al., 2024)). Consequently, a consequence from a known property of Bregman divergences (Shalev-Shwartz, 2012) is that, for any $\mu \in \mathcal{M}_{\mu_0}^{p}$ and $\mu' \in \mathcal{M}_{\mu_0}^{p,*}$,

$$\Gamma(\mu,\mu') \geqslant \frac{1}{2}\|\mu-\mu'\|_{\infty,1}^2. \tag{27}$$

*Lemma.* Let $(q^t)_{t\in[T]}$ be a sequence of probability transition kernels, and $(z^t)_{t\in[T]}$ a sequence of vectors in $\mathbb{R}^{N\times|\mathcal{X}|\times|\mathcal{A}|}$, such that $\|z^t\|_{1,\infty} \leqslant \zeta$ for all $t \in [T]$. Initialize $\pi_n^1(a|x) := 1/|\mathcal{A}|$ as the uniform policy. For every $t \in [T]$, let $\tilde{\pi}^t := (1-\alpha_t)\pi^t + \alpha_t|\mathcal{A}|^{-1}$ be a smoothed version of the policy with $\alpha_t := 1/(t+1)$ and $\tilde{\mu}^t := \mu^{\tilde{\pi}^t, q^t}$. For each $t \in [T]$, compute iteratively

$$\mu^{t+1} \in \operatorname*{arg\,min}_{\mu\in\mathcal{M}_{\mu_0}^{q^{t+1}}} \tau\langle z^t, \mu\rangle + \Gamma(\mu, \tilde{\mu}^t). \tag{28}$$

Hence, there is a $\tau > 0$ such that, for any sequence $(\nu^t)_{t\in[T]}$, with $\nu^t := \nu^{\pi,q^t}$ for a common policy $\pi$,

$$\textstyle\sum_{t=1}^{T}\langle z^t, \mu^t - \nu^t\rangle \leqslant O\big(\zeta N\sqrt{V_T|\mathcal{X}|\log(|\mathcal{A}|)T\log(T)}\big),$$

where $V_T \geqslant 1 + \max_{(n,x,a)}\sum_{t=1}^{T-1}\|q_n^t(\cdot|x,a) - q_n^{t+1}(\cdot|x,a)\|_1$.

*Proof.* Throughout this proof, for all $t \in [T]$ we denote by $\pi^t$ the policy inducing $\mu^t$, meaning that $\mu^t := \mu^{\pi^t,q^t}$ and $\tilde{\mu}^t := \mu^{\tilde{\pi}^t,q^t}$. We assume here that $\max_{(n,x,a)}\sum_{t=1}^{T-1}\|q_n^t(\cdot|x,a) - q_n^{t+1}(\cdot|x,a)\|_1 \leqslant c\log(T)$ for $c$ a constant, as this is the case for all the transition estimators we use to obtain the main results of the article.

As $\mathcal{M}_{\mu_0}^{q^{t+1}}$ is a convex set (only linear constraints), the optimality conditions and the definition of a Bregman divergence in Eq. (26) imply that for all $\nu^{t+1} \in \mathcal{M}_{\mu_0}^{q^{t+1}}$,

$$\langle\tau z^t + \nabla\psi(\mu^{t+1}) - \nabla\psi(\tilde{\mu}^t), \nu^{t+1} - \mu^{t+1}\rangle \geqslant 0.$$

Re-arranging the terms and using the three points inequality for Bregman divergences (Bubeck, 2015) we get that,

$$\tau\langle z^t, \mu^{t+1} - \nu^{t+1}\rangle \leqslant \langle \nabla\psi(\mu^{t+1}) - \nabla\psi(\tilde{\mu}^t), \nu^{t+1} - \mu^{t+1}\rangle = \Gamma(\nu^{t+1}, \tilde{\mu}^t) - \Gamma(\nu^{t+1}, \mu^{t+1}) - \Gamma(\mu^{t+1}, \tilde{\mu}^t).$$

Therefore, by adding and subtracting $\tau\langle z^t, \mu^t - \nu^t\rangle$ on the left-hand side,

$$\begin{aligned}\tau\langle z^t, \mu^{t+1} - \nu^{t+1}\rangle + \tau\langle z^t, \mu^t - \nu^t\rangle - \tau\langle z^t, \mu^t - \nu^t\rangle &\leqslant \Gamma(\nu^{t+1}, \tilde{\mu}^t) - \Gamma(\nu^{t+1}, \mu^{t+1}) - \Gamma(\mu^{t+1}, \tilde{\mu}^t)\\ \Rightarrow \tau\langle z^t, \mu^t - \nu^t\rangle \leqslant \tau\langle z^t, \nu^{t+1} - \nu^t\rangle + \tau\langle z^t, \mu^t - \mu^{t+1}\rangle &+ \Gamma(\nu^{t+1}, \tilde{\mu}^t) - \Gamma(\nu^{t+1}, \mu^{t+1}) - \Gamma(\mu^{t+1}, \tilde{\mu}^t).\end{aligned}$$

Then, by summing over $t \in [T]$, we obtain that

$$\begin{aligned}\sum_{t=1}^T \langle z^t, \mu^t - \nu^t\rangle \leqslant\; &\underbrace{\frac{1}{\tau}\sum_{t=1}^T \left[\tau\langle z^t, \mu^t - \mu^{t+1}\rangle - \Gamma(\mu^{t+1}, \tilde{\mu}^t)\right]}_{A} + \underbrace{\frac{1}{\tau}\sum_{t=1}^T \left[\Gamma(\nu^{t+1}, \tilde{\mu}^t) - \Gamma(\nu^{t+1}, \mu^{t+1})\right]}_{B}\\ &+ \underbrace{\sum_{t=1}^T \langle z^t, \nu^{t+1} - \nu^t\rangle}_{C}.\end{aligned} \tag{29}$$

The term $A$ arises due to our lack of knowledge of $z^t$ at the beginning of episode $t$ for all episodes (adversarial losses hypothesis). To address this, we employ Young's inequality and the strong convexity of $\Gamma$. For the term $B$, in the classic Online Mirror Descent proof (Shalev-Shwartz, 2012), where the set of constraints is fixed, the sum of the differences between the Bregman divergences telescopes (as would be the case with a fixed $\nu$). However, because we are dealing with time-varying constraint sets, this telescoping effect does not occur in our situation. We will now proceed to derive an upper bound for each term, starting with term $C$ that is straightforward.

**Step $0$: upper bound on $C$.** Applying Holder's inequality, Lem. A.5 with a fixed policy $\pi$, and the hypothesis that $\|z^t\|_{1,\infty} \leqslant \zeta$,

$$C = \sum_{t=1}^T \langle z^t, \nu^{t+1} - \nu^t\rangle \leqslant \sum_{t=1}^T \|z^t\|_{1,\infty}\|\nu^{t+1} - \nu^t\|_{\infty,1} \leqslant \zeta c|\mathcal{X}|N\log(T). \tag{30}$$

**Step $1$: upper bound on $B$.** We now analyse the second term of the sum in Eq. (29). To make the Bregman divergence terms telescope we add and subtract $\Gamma(\nu^t, \mu^t) - \Gamma(\nu^t, \tilde{\mu}^t)$, obtaining

$$\begin{aligned}\sum_{t=1}^T \Gamma(\nu^{t+1}, \tilde{\mu}^t) - \Gamma(\nu^{t+1}, \mu^{t+1}) = &\underbrace{\sum_{t=1}^T \Gamma(\nu^{t+1}, \tilde{\mu}^t) - \Gamma(\nu^t, \tilde{\mu}^t)}_{(i)} + \underbrace{\sum_{t=1}^T \Gamma(\nu^t, \tilde{\mu}^t) - \Gamma(\nu^t, \mu^t)}_{(ii)}\\ &+ \underbrace{\sum_{t=1}^T \Gamma(\nu^t, \mu^t) - \Gamma(\nu^{t+1}, \mu^{t+1})}_{(iii)}.\end{aligned} \tag{31}$$

We analyze each term. Using the definition of a Bregman divergence induced by $\psi$ in Eq. (26) we get that

$$\begin{aligned}(i) &= \sum_{t=1}^T \psi(\nu^{t+1}) - \psi(\tilde{\mu}^t) - \langle \nabla\psi(\tilde{\mu}^t), \nu^{t+1} - \tilde{\mu}^t\rangle - \psi(\nu^t) + \psi(\tilde{\mu}^t) + \langle \nabla\psi(\tilde{\mu}^t), \nu^t - \tilde{\mu}^t\rangle\\ &= \sum_{t=1}^T \psi(\nu^{t+1}) - \psi(\nu^t) + \sum_{t=1}^T \langle \nabla\psi(\tilde{\mu}^t), \nu^t - \nu^{t+1}\rangle\\ &\leqslant -\psi(\nu^1) + \sum_{t=1}^T \|\nabla\psi(\tilde{\mu}^t)\|_{1,\infty}\|\nu^t - \nu^{t+1}\|_{\infty,1},\end{aligned}$$

where in the last inequality we use the telescoping nature of the first term and applied Hölder's inequality to the second term. Recall that for $v := (v_n)_{n\in[N]}$ such that $v_n \in \mathbb{R}^{\mathcal{X}\times\mathcal{A}}$, we defined $\|v\|_{\infty,1} := \sup_{n\in[N]} \|v_n\|_1$. We now also define $\|\omega\|_{1,\infty} := \sup_v\{|\langle\omega, v\rangle|, \|v\|_{\infty,1} \leqslant 1\} = \sum_{n=1}^N \sup_{x,a} |\omega_n(x,a)|$ as the respective dual norm.

With our choice of Bregman divergence, and given that

$$\tilde{\pi}^t := (1-\alpha_t)\pi^t + \alpha_t \frac{1}{|\mathcal{A}|},$$

for each $n \in [N], (x,a) \in \mathcal{X}\times\mathcal{A}$, $|\nabla\psi(\tilde{\mu}^t)(n,x,a)| = |\log(\tilde{\pi}_n^t(a|x))| \leqslant \log(|\mathcal{A}|/\alpha_t)$.

From the Lemma hypothesis, there is a common policy $\pi$ such that for all $t \in [T]$, $\nu^t := \nu^{\pi,q^t}$. Hence, from the result above,

$$\begin{aligned}
(i) &\leqslant -\psi(\nu^1) + \sum_{t=1}^T N \log\left(\frac{|\mathcal{A}|}{\alpha_t}\right)\|\nu^{\pi,q^t} - \nu^{\pi,q^{t+1}}\|_{\infty,1} \\
&\leqslant -\psi(\nu^1) + N\log\left(\frac{|\mathcal{A}|}{\min_{t\in[T]}\alpha_t}\right) c|\mathcal{X}|N\log(T),
\end{aligned}$$

where the last inequality comes from Lem. A.5 with a fixed $\pi$.

As for the second term, using our definition of $\Gamma$, we obtain that

$$\begin{aligned}
(ii) &= \sum_{t=1}^T \sum_{n,x,a} \nu_n^{\pi,q^t}(x,a)\log\left(\frac{\pi_n(a|x)}{\tilde{\pi}_n^t(a|x)}\right) - \sum_{n,x,a}\nu_n^{\pi,q^t}(x,a)\log\left(\frac{\pi_n(a|x)}{\pi_n^t(a|x)}\right) \\
&= \sum_{t=1}^T \sum_{n,x,a} \nu_n^{\pi,q^t}(x,a)\log\left(\frac{\pi_n^t(a|x)}{\tilde{\pi}_n^t(a|x)}\right) \\
&= \sum_{t=1}^T \sum_{n,x,a} \nu_n^{\pi,q^t}(x,a)\log\left(\frac{\pi_n^t(a|x)}{(1-\alpha_t)\pi_n^t(a|x) + \alpha_t/|\mathcal{A}|}\right) \\
&\leqslant N\sum_{t=1}^T(-\log(1-\alpha_t)) \leqslant 2N\sum_{t=1}^T \alpha_t,
\end{aligned}$$

where the last inequality is valid if $0 \leqslant \alpha_t \leqslant 0.5$.

The third term telescopes, hence, since $-\Gamma(\nu^{T+1}, \mu^{T+1}) \leqslant 0$ because a Bregman divergence is always non-negative,

$$(iii) \leqslant \Gamma(\nu^1, \mu^1).$$

Before adding back the three terms, note that, for $\pi_n^1(a|x) = 1/|\mathcal{A}|$, we have $\Gamma(\nu^1,\mu^1) - \psi(\nu^1) = -\psi(\mu^1)$. Furthermore, $-\psi(\mu^1) \leqslant N\log(|\mathcal{A}|)$. Therefore,

$$\Gamma(\nu^1,\mu^1) - \psi(\nu^1) \leqslant N\log(|\mathcal{A}|). \tag{32}$$

Summing over our bounds and using the Inequality (32), we get that $B$ is upper bounded as

$$\begin{aligned}
\frac{1}{\tau}\sum_{t=1}^T\left[\Gamma(\nu^{t+1},\tilde{\mu}^t) - \Gamma(\nu^{t+1},\mu^{t+1})\right] &\leqslant \frac{1}{\tau}\left[(i)+(ii)+(iii)\right] \\
&\leqslant \frac{N}{\tau}\log(|\mathcal{A}|) + \frac{N^2 c|\mathcal{X}|}{\tau}\log\left(\frac{|\mathcal{A}|}{\min_{t\in[T]}\alpha_t}\right)\log(T) + \frac{2N}{\tau}\sum_{t=1}^T\alpha_t.
\end{aligned} \tag{33}$$

**Step 2: Upper bound on $A$.** It remains to upper bound term $A$ from Eq. (29),

$$A = \frac{1}{\tau}\left[\sum_{t=1}^T \tau\langle z^t, \mu^t - \mu^{t+1}\rangle - \Gamma(\mu^{t+1},\tilde{\mu}^t)\right], \tag{34}$$

representing what we pay for not knowing the loss function in advance. For that we use Young's inequality (Beck & Teboulle, 2003): for any $\sigma > 0$ to be optimized later, and for each episode $t \in [T]$,

$$\tau\langle z^t, \mu^t - \mu^{t+1}\rangle - \Gamma(\mu^{t+1}, \tilde{\mu}^t) \leqslant \frac{\tau^2\|z^t\|_{1,\infty}^2}{2\sigma} + \frac{\sigma}{2}\|\mu^t - \mu^{t+1}\|_{\infty,1}^2 - \Gamma(\mu^{t+1}, \tilde{\mu}^t). \tag{35}$$

From the definition of $\Gamma$ in Eq. (4), we have that

$$\Gamma(\mu^{t+1}, \tilde{\mu}^t) = \sum_{n=1}^{N}\sum_{(x,a)} \mu_n^{t+1}(x,a)\log\left(\frac{\pi_n^{t+1}(a|x)}{\tilde{\pi}^t(a|x)}\right) = \Gamma(\mu^{t+1}, \mu^{\tilde{\pi}^t, q^{t+1}}).$$

From the strong convexity of $\psi$, as $\mu^{t+1} \in \mathcal{M}_{\mu_0}^{q^{t+1}}$ and $\mu^{\tilde{\pi}^t, q^{t+1}} \in \mathcal{M}_{\mu_0}^{q^{t+1},*}$, we then have from Eq. (27) that

$$\Gamma(\mu^{t+1}, \tilde{\mu}^t) = \Gamma(\mu^{t+1}, \mu^{\tilde{\pi}^t, q^{t+1}}) \geqslant \frac{1}{2}\|\mu^{t+1} - \mu^{\tilde{\pi}^t, q^{t+1}}\|_{\infty,1}^2. \tag{36}$$

Using the fact that for any vectors $a, b, c \in \mathbb{R}^d$ and for any norm $\|\cdot\|$, the inequality $\|a-b\|^2 \leqslant 2(\|a-c\|^2 + \|b-c\|^2)$ holds, we then have by Eq. (36)

$$\begin{aligned}
\frac{1}{4}\|\mu^t - \mu^{t+1}\|_{\infty,1}^2 - \Gamma(\mu^{t+1}, \tilde{\mu}^t) &\leqslant \frac{1}{4}\|\mu^t - \mu^{t+1}\|_{\infty,1}^2 - \frac{1}{2}\|\mu^{\tilde{\pi}^t, q^{t+1}} - \mu^{t+1}\|_{\infty,1}^2 \\
&\leqslant \frac{1}{2}\big(\|\mu^t - \mu^{\tilde{\pi}^t, q^{t+1}}\|_{\infty,1}^2 + \|\mu^{\tilde{\pi}^t, q^{t+1}} - \mu^{t+1}\|_{\infty,1}\big) - \frac{1}{2}\|\mu^{\tilde{\pi}^t, q^{t+1}} - \mu^{t+1}\|_{\infty,1}^2 \\
&= \frac{1}{2}\|\mu^t - \mu^{\tilde{\pi}^t, q^{t+1}}\|_{\infty,1}^2.
\end{aligned} \tag{37}$$

For any $n \in [N]$, we have $\|\mu_n^t - \mu_n^{\tilde{\pi}^t, q^{t+1}}\|_1 \leqslant 2$. Using this result along with Lem. A.6 for $p = \widehat{p}^t$ and $q = \widehat{p}^{t+1}$, we derive the first inequality below. To obtain the second inequality, we apply Lem. A.5 with the sequence of policies $(\pi^t)_{t \in [T]}$.

$$\begin{aligned}
\sum_{t=1}^{T}\|\mu^t - \mu^{\tilde{\pi}^t, q^{t+1}}\|_{\infty,1}^2 &\leqslant 2\sum_{t=1}^{T}\sup_{n \in [N]}\sum_{i=0}^{n-1}\sum_{x,a}\mu_i^t(x,a)\|q_{i+1}^t(\cdot|x,a) - q_{i+1}^{t+1}(\cdot|x,a)\|_1 + 4N\sum_{t=1}^{T}\alpha_t \\
&\leqslant 2c|\mathcal{X}||\mathcal{A}|N\log(T) + 4N\sum_{t=1}^{T}\alpha_t.
\end{aligned} \tag{38}$$

Therefore, summing Eq. (35) over $t \in [T]$ with $\sigma = 1/2$, and plugging the inequality above, yields

$$\sum_{t=1}^{T}\tau\langle z^t, \mu^t - \mu^{t+1}\rangle - \Gamma(\mu^{t+1}, \tilde{\mu}^t) \leqslant \tau^2\sum_{t=1}^{T}\|z^t\|_{1,\infty}^2 + c|\mathcal{X}||\mathcal{A}|N\log(T) + 2N\sum_{t=1}^{T}\alpha_t.$$

Using that $\|z^t\|_{1,\infty} \leqslant \zeta$ and dividing by $\tau$ entails:

$$A \leqslant \tau\zeta^2 T + \frac{N}{\tau}\left(c|\mathcal{X}||\mathcal{A}|\log(T) + 2\sum_{t=1}^{T}\alpha_t\right). \tag{39}$$

**Conclusion.** Finally, by replacing the final bounds of Eqs. (33), (39), and (30), we obtain

$$\begin{aligned}
\sum_{t=1}^{T}\langle z^t, \mu^t - \nu^t\rangle &\leqslant A + B + C \\
&\leqslant \tau T\zeta^2 + \frac{N}{\tau}\left(c|\mathcal{X}||\mathcal{A}|\log(T) + 2\sum_{t=1}^{T}\alpha_t\right) + \frac{N}{\tau}\log(|\mathcal{A}|) \\
&\quad + \frac{N^2 c|\mathcal{X}|}{\tau}\log\left(\frac{|\mathcal{A}|}{\min_{t \in [T]}\alpha_t}\right)\log(T) + \frac{2N}{\tau}\sum_{t=1}^{T}\alpha_t + \zeta N c|\mathcal{X}|\log(T).
\end{aligned}$$

In particular, for $\alpha_t = 1/(t+1)$,

$$\sum_{t=1}^{T} \langle z^t, \mu^t - \nu^t \rangle \leqslant \tau T \zeta^2 + \frac{1}{\tau} \underbrace{N\Big[\log(T)\Big(c|\mathcal{X}||\mathcal{A}|+4\Big) + \log(|\mathcal{A}|) + 2Nc|\mathcal{X}|\log(T)^2\log(|\mathcal{A}|)\Big]}_{=:b} + \zeta Nc|\mathcal{X}|\log(T).$$

Optimising over $\tau = \sqrt{\frac{b}{\zeta^2 T}}$,

$$\sum_{t=1}^{T} \langle z^t, \mu^t - \nu^t \rangle \leqslant 2\zeta\sqrt{bT} + \zeta Nc|\mathcal{X}|\log(T) = O\Big(\zeta\sqrt{cN|\mathcal{X}||\mathcal{A}|T\log(T)} + \zeta N\sqrt{c|\mathcal{X}|\log(|\mathcal{A}|)T}\log(T) + \zeta Nc|\mathcal{X}|\log(T)\Big),$$

concluding the proof.

□

## D. Bandit Feedback with Bonus in RL

**Notations.** Throughout App. D, we define the trajectory observed by the learner in episode $t$ as $o^t := (x_n^t, a_n^t, \ell_n^t(x_n^t, a_n^t))_{n\in[N]}$. Let $\mathcal{F}^t$ denote the $\sigma$-algebra generated by the observations up to episode $t$, i.e., $\mathcal{F}^t := \sigma(o^1, \ldots, o^{t-1})$. We use $\mathbb{E}_t$ to represent the conditional expectation with respect to the observations up to episode $t$, i.e., $\mathbb{E}_t[\cdot] := \mathbb{E}[\cdot|\mathcal{F}^t]$.

**Overview of existing approaches.** To adapt Alg. 1 to the bandit case, we need to estimate the loss function for each MD update. A classic choice using importance sampling is:

$$\hat{\ell}_n^t(x,a) = \frac{\ell_n^t(x,a)}{\mu_n^{\pi^t,p}(x,a)} \mathbb{1}_{\{x_n^t = x, a_n^t = a\}}.$$

This update is unbiased, as $\mathbb{E}[\mathbb{1}_{\{x_n^t=x,a_n^t=a\}}] = \mathbb{E}[\mathbb{E}[\mathbb{1}_{\{x_n^t=x,a_n^t=a\}}|\mathcal{F}^t]] = \mu_n^{\pi^t,p}(x,a)$. However, since we do not know the true transition probability $p$, we cannot use this estimate directly. In (Rosenberg & Mansour, 2019a), they use $\mu^{\pi^t,\hat{p}^t}$ with UC-O-REPS and achieve a regret of $O(T^{3/4})$.

Consider the following confidence set, that is further detailed in Eq. (41),

$$\Omega^t := \{q \mid |q_n(x'|x,a) - \hat{p}_n^t(x'|x,a)| \leqslant \varepsilon_n^t(x'|x,a), \forall (x,a,x') \in \mathcal{X}\times\mathcal{A}\times\mathcal{X}, n \in [N]\}.$$

In (Jin et al., 2020), the authors incorporate a parameter $\gamma$ for implicit exploration, an idea from multi-armed bandits (Neu, 2015), and use the following estimate:

$$\hat{\ell}_n^t(x,a) = \frac{\ell_n^t(x,a)}{\bar{\mu}_n^t(x,a)+\gamma} \mathbb{1}_{\{x_n^t = x, a_n^t = a\}}, \tag{40}$$

where $\bar{\mu}_n^t(x,a) := \max_{q\in\Omega^t} \mu^{\pi^t,q}$. Although this is a biased estimate ($\mu_n^{\pi^t,p}(x,a) \leqslant \bar{\mu}_n(x,a)$), $\Omega^t$ is constructed to ensure that the bias introduced is reasonably small. They also argue that $\bar{\mu}$ can be computed efficiently through dynamic programming. They demonstrate that running UC-O-REPS from (Rosenberg & Mansour, 2019b) with this estimate achieves $\tilde{O}(\sqrt{T})$ regret, improving upon previous results.

In Alg. 2 we detail our method for solving the RL problem with adversarial losses, unknown probability transitions and bandit feedback. We proceed to the regret analysis.

### D.1. Auxiliary lemmas

**Lemma D.1** (Lem. A.2 of (Luo et al., 2021), adapted from Lem. 1 of (Neu, 2015))**.** *Let $(z_n^t(x,a))_{t\in[T]}$ be a sequence of functions $\mathcal{F}_t$-measurable, such that $z_n^t(x,a) \in [0,R]$ for each $(x,a) \in \mathcal{X}\times\mathcal{A}$, and $n \in [N]$. Let $Z_n^t(x,a) \in [0,R]$ be a random variable such that $\mathbb{E}_t[Z_n^t(x,a)] = z_n^t(x,a)$. Then with probability $1-\delta$,*

$$\sum_{t=1}^{T}\sum_{n=1}^{N}\sum_{x,a}\left(\frac{\mathbb{1}_{\{x_n^t=x,a_n^t=a\}}Z_n^t(x,a)}{\bar{\mu}_n^t(x,a)+\gamma} - \frac{\mu_n^{\pi^t,p}(x,a)z_n^t(x,a)}{\bar{\mu}_n^t(x,a)}\right) \leqslant \frac{RN}{2\gamma}\log\left(\frac{N}{\delta}\right).$$

**Algorithm 2** MD-CURL with Additive Bonus for Bandit feedback RL

**Input:** number of episodes $T$, initial policy $\pi^1 \in \Pi$, initial state-action distribution $\mu_0$, initial state-action distribution sequence $\mu^1 = \tilde{\mu}^1 = \mu^{\pi^1, \widehat{p}^1}$ with $\widehat{p}_n^1(\cdot|x,a) = 1/|\mathcal{X}|$ for all $(n,x,a)$, learning rate $\tau > 0$, exploration parameter $\gamma = \tau$ (tuned in the proof of Thm. 4.1), sequence of parameters $(\alpha_t)_{t\in[T]}$ with $\alpha_t = 1/(t+1)$.
**Init.:** $\forall (n,x,a,x')$, $N_n^1(x,a) = M_n^1(x'|x,a) = 0$
**for** $t = 1, \dots, T$ **do**
  Agent starts at $(x_0^t, a_0^t) \sim \mu_0(\cdot)$
  **for** $n = 1, \dots, N$ **do**
    Env. draws new state $x_n^t \sim p_n(\cdot|x_{n-1}^t, a_{n-1}^t)$
    Update counts

$$N_{n-1}^{t+1}(x_{n-1}^t, a_{n-1}^t) \leftarrow N_{n-1}^t(x_{n-1}^t, a_{n-1}^t) + 1$$
$$M_{n-1}^{t+1}(x_n^t|x_{n-1}^t, a_{n-1}^t) \leftarrow M_{n-1}^t(x_n^t|x_{n-1}^t, a_{n-1}^t) + 1$$

    Agent chooses an action $a_n^t \sim \pi_n^t(\cdot|x_n^t)$
    Observe local loss $\ell_n^t(x_n^t, a_n^t)$
  **end for**
  Update transition estimate for all $(n,x,a,x')$: $\widehat{p}_n^{t+1}(x'|x,a) := \frac{M_n^{t+1}(x'|x,a)}{\max\{1, N_n^t(x,a)\}}$
  Compute bonus sequence for all $(n,x,a)$: $b_n^t(x,a) := \frac{(N-n)C_\delta}{\sqrt{\max\{1, N_n^{t+1}(x,a)\}}}$
  Compute optimistic state-action distribution for all $(n,x,a)$: $\bar{\mu}_n^t(x,a) := \max_{q\in\Omega^t} \mu^{\pi^t, q}$, where $\Omega^t$ is defined as in Eq. (41)
  Compute loss estimate for all $(n,x,a)$: $\widehat{\ell}_n^t(x,a) = \frac{\ell_n^t(x,a)}{\bar{\mu}_n^t(x,a)+\gamma} \mathbb{1}_{\{x_n^t = x, a_n^t = a\}}$
  Compute policy $\pi_n^{t+1}(x,a)$ by solving

$$\mu^{t+1} \in \underset{\mu \in \mathcal{M}_{\mu_0}^{\widehat{p}^{t+1}}}{\arg\min} \left\{ \tau \langle \widehat{\ell}^t - b^t, \mu \rangle + \Gamma(\mu, \tilde{\mu}^t) \right\},$$

  which has a closed-form solution for $\pi^{t+1}$ (see Sec. A.2)
  Compute $\tilde{\pi}^{t+1}$, the smooth version of $\pi^{t+1}$:

$$\tilde{\pi}^{t+1} = (1 - \alpha_t)\pi^{t+1} + \alpha_t/|\mathcal{A}|$$

  and the associated state-action distribution $\tilde{\mu}^{t+1} := \mu^{\tilde{\pi}^{t+1}, \widehat{p}^{t+1}}$
**end for**

We define the confidence interval used in Alg. 2 as

$$\Omega^t := \{q \,|\, |q_n(x'|x,a) - \widehat{p}^t_n(x'|x,a)| \leqslant \varepsilon^t_n(x'|x,a),\ \text{for all } (x,a,x') \in \mathcal{X} \times \mathcal{A} \times \mathcal{X}, n \in [N]\}, \tag{41}$$

with

$$\varepsilon^t_n(x'|x,a) := 2\sqrt{\frac{\widehat{p}^t_n(x'|x,a)\log\left(\frac{TN|\mathcal{X}||\mathcal{A}|}{\delta}\right)}{\max\{1, N^t_{n-1}(x,a)\}}} + \frac{14\log\left(\frac{TN|\mathcal{X}||\mathcal{A}|}{\delta}\right)}{3\max\{1, N^t_{n-1}(x,a)\}}.$$

We present two results regarding this confidence set. The first result, based on the empirical Bernstein inequality, shows that the true probability kernel $p$ belongs to this confidence set with high probability. The second is a key lemma from (Jin et al., 2020), which explains how the confidence set shrinks over time. For the proofs, the reader is referred to the original references.

**Lemma D.2** (Empirical Bernstein inequality, Thm. 4 (Maurer & Pontil, 2009)/Lem. 2 (Jin et al., 2020))**.** *With probability at least $1-4\delta$, we have that $p \in \Omega^t$ for all $t \in [T]$.*

**Lemma D.3** (Lem. 4 (Jin et al., 2020))**.** *With probability at least $1-6\delta$, for any collection of transition functions $(p^{x,t})_{x\in\mathcal{X}}$ such that $p^{x,t} \in \Omega^t$ for all $x$, we have*

$$\sum_{t=1}^{T}\sum_{n=1}^{N}\sum_{x,a}|\mu_n^{\pi^t,p^{x,t}}(x,a) - \mu_n^{\pi^t,p}(x,a)| = O\left(N^2|\mathcal{X}|\sqrt{\mathcal{A}T\log\left(\frac{TN|\mathcal{X}||\mathcal{A}|}{\delta}\right)}\right).$$

### D.2. Proof of Thm. 4.1

*Proof.* We start by decomposing the static regret with respect to any policy $\pi \in (\Delta_{\mathcal{A}})^{\mathcal{X}\times N}$ as follows

$$\begin{aligned} R_T(\pi) = &\underbrace{\sum_{t=1}^{T}\langle \ell^t, \mu^{\pi^t,p} - \mu^{\pi^t,\widehat{p}^t}\rangle}_{1, R_T^{\text{MDP}}} + \underbrace{\sum_{t=1}^{T}\langle \ell^t - b^t, \mu^{\pi^t,\widehat{p}^t} - \mu^{\pi,\widehat{p}^t}\rangle}_{2, R_T^{\text{policy/MD}}} \\ &+ \underbrace{\sum_{t=1}^{T}\langle \ell^t, \mu^{\pi,\widehat{p}^t} - \mu^{\pi,p}\rangle - \sum_{t=1}^{T}\langle b^t, \mu^{\pi,\widehat{p}^t}\rangle}_{3, R_T^{\text{policy/MD}}} + \underbrace{\sum_{t=1}^{T}\langle b^t, \mu^{\pi^t,\widehat{p}^t}\rangle}_{4,\ \text{Bonus term}}. \end{aligned} \tag{42}$$

#### D.2.1. Term 1: $R_T^{\text{MDP}}$

The analysis of this term is already provided in App. B.2. Here, we can further leverage the fact that the objective function is linear and that, by definition, $\ell^t_n \in [0,1]$. Therefore, with probability at least $1-2\delta$, we have:

$$R_T^{\text{MDP}} \leqslant 3|\mathcal{X}|N^2\sqrt{2|\mathcal{A}|T\log\left(\frac{|\mathcal{X}||\mathcal{A}|NT}{\delta}\right)} + 2|\mathcal{X}|N^2\sqrt{2T\log\left(\frac{N}{\delta}\right)}. \tag{43}$$

#### D.2.2. Term 2: $R_T^{\text{policy/MD}}$

In practice, the learner plays using the estimated loss function minus the bonus. Hence, $R_T^{\text{policy/MD}}$ accounts for both the bias introduced by the loss estimation and the standard mirror descent regret bound. We start with the following decomposition:

$$\begin{aligned} R_T^{\text{policy/MD}} &= \sum_{t=1}^{T}\langle \ell^t - b^t, \mu^{\pi^t,\widehat{p}^t} - \mu^{\pi,\widehat{p}^t}\rangle \\ &= \underbrace{\sum_{t=1}^{T}\langle \ell^t - \widehat{\ell}^t, \mu^{\pi^t,\widehat{p}^t} - \mu^{\pi,\widehat{p}^t}\rangle}_{\text{Bias terms}} + \underbrace{\sum_{t=1}^{T}\langle \widehat{\ell}^t - b^t, \mu^{\pi^t,\widehat{p}^t} - \mu^{\pi,\widehat{p}^t}\rangle}_{\text{MD term}} \end{aligned}$$

**Mirror Descent term in $R_T^{\text{policy/MD}}$.** We begin by analyzing the error term from applying Mirror Descent with varying constraint sets, which is similar to that of Lem. 2.1 with $z^t = \widehat{\ell}^t - b^t$, and $(\widehat{p}^t)_{t\in[T]}$ as the probability kernel sequence defining the varying constraint sets. However, special attention is needed for the sup norm of the subgradient term since it now involves an estimate of the loss function. Additionally, the optimal learning rate $\tau$ now also depends on both the exploration parameter $\gamma$ and the analysis of the bias terms, we provide a detailed explanation of how the entire analysis is affected below.

In proving Lem. 2.1 in App. C, the regret term for Mirror Descent is split into three terms: term $A$ in Eq. (34), term $B$ in Eq. (31), and term $C$ in Eq. (30). The analysis of term $B$ remains unchanged since this term is independent of the chosen loss function. We focus on what changes for terms $A$ and $C$.

As in the proof of Lem. 2.1, we use again the notation $\mu^t := \mu^{\pi^t,\widehat{p}^t}$ for all $t \in [T]$. From Eq. (34) term $A$ is defined as

$$A = \frac{1}{\tau}\sum_{t=1}^{T}\big[\tau\langle \widehat{\ell}^t - b^t, \mu^t - \mu^{t+1}\rangle - \Gamma(\mu^{t+1}, \tilde{\mu}^t)\big].$$

For a fixed $t$, from Young's inequality we have that

$$\sum_{n=1}^{N}\tau\langle \widehat{\ell}_n^t - b_n^t, \mu_n^t - \mu_n^{t+1}\rangle \leqslant \sum_{n=1}^{N}\tau^2\frac{\|\widehat{\ell}_n^t - b_n^t\|_\infty^2}{2\sigma} + \frac{\sigma}{2}\|\mu_n^t - \mu_n^{t+1}\|_1^2.$$

Following the analysis of term $A$ in App. C, in special Eqs. (36), (37), and (38), we obtain that for $\sigma = 1/2$,

$$A \leqslant \sum_{t=1}^{T}\sum_{n=1}^{N}\tau\|\widehat{\ell}_n^t - b_n^t\|_\infty^2 + \frac{eN|\mathcal{X}||\mathcal{A}|\log(T)}{\tau} + \frac{2N}{\tau}\sum_{t=1}^{T}\alpha_t. \tag{44}$$

From Eq. (30), with the notation $\nu^t := \nu^{\pi,\widehat{p}^t}$, term $C$ is defined as $C = \frac{1}{\tau}\sum_{t=1}^{T}\tau\langle\widehat{\ell}^t - b^t, \nu^{t+1} - \nu^t\rangle$. From Young's inequality with $\sigma = 1/2$ we obtain that

$$\begin{aligned} C &\leqslant \frac{1}{\tau}\sum_{t=1}^{T}\frac{\tau^2\|\widehat{\ell}^t - b^t\|_{1,\infty}^2}{2\sigma} + \frac{1}{\tau}\frac{\sigma}{2}\sum_{t=1}^{T}\|\nu^{t+1} - \nu^t\|_{\infty,1}^2 \\ &\leqslant \sum_{t=1}^{T}\sum_{n=1}^{N}\tau\|\widehat{\ell}_n^t - b_n^t\|_\infty^2 + \frac{eN|\mathcal{X}|\log(T)}{2\tau}. \end{aligned} \tag{45}$$

**Bounding the sup norm of the estimated loss function.** Recall from the definition of the bonus function in Eq. (11), with $L = 1$, that $\|b_n^t\|_\infty \leqslant NC_\delta =: b$ for all $n \in [N]$ and $t \in [T]$. As $\|\widehat{\ell}_n^t - b_n^t\|_\infty^2 \leqslant \|\widehat{\ell}_n^t\|_\infty^2 + \|b_n^t\|_\infty^2$, we can focus on the term involving the sup norm of the estimated loss function.

We apply Lem. D.1 with $Z_n^t(x,a) = \frac{\mathbb{1}_{\{x_n^t = x, a_n^t = a\}}\ell_n^t(x,a)^2}{\bar{\mu}_n^t(x,a)+\gamma}$ and $z_n^t(x,a) = \frac{\mu_n^{\pi^t,p}(x,a)\ell_n^t(x,a)^2}{\bar{\mu}_n^t(x,a)+\gamma}$. Note that $Z_n^t(x,a), z_n^t(x,a) \leqslant \frac{1}{\gamma}$, that $z_n^t(x,a)$ is $\mathcal{F}^t$-measurable, and that $\mathbb{E}_t[Z_n^t(x,a)] = z_n^t(x,a)$. Therefore, with probability $1-\delta$,

$$\begin{aligned} \tau\sum_{t=1}^{T}\sum_{n=1}^{N}\|\widehat{\ell}_n^t\|_\infty^2 &\leqslant \tau\sum_{t=1}^{T}\sum_{n=1}^{N}\sum_{x,a}\widehat{\ell}_n^t(x,a)^2 \\ &= \tau\sum_{t=1}^{T}\sum_{n=1}^{N}\sum_{x,a}\frac{\mathbb{1}_{\{x_n^t=x,a_n^t=a\}}\ell_n^t(x,a)}{\bar{\mu}_n^t(x,a)+\gamma}\frac{\mathbb{1}_{\{x_n^t=x,a_n^t=a\}}\ell_n^t(x,a)}{\bar{\mu}_n^t(x,a)+\gamma} \\ &= \tau\sum_{t=1}^{T}\sum_{n=1}^{N}\sum_{x,a}\frac{\mathbb{1}_{\{x_n^t=x,a_n^t=a\}}Z_n^t(x,a)}{\bar{\mu}_n^t(x,a)+\gamma} \\ &\underbrace{\leqslant}_{\text{Lem. D.1}} \tau\sum_{t=1}^{T}\sum_{n=1}^{N}\sum_{x,a}\frac{\mu_n^{\pi^t,p}(x,a)}{\bar{\mu}_n^t(x,a)}\frac{\mu_n^{\pi^t,p}(x,a)\ell_n^t(x,a)^2}{\bar{\mu}_n^t(x,a)+\gamma} + \tau\frac{1}{\gamma}\frac{N}{2\gamma}\log\left(\frac{N}{\delta}\right). \end{aligned}$$

Lem. D.2 states that the true probability transition $p \in \Omega^t$ with probability $1 - 4\delta$ for all $t \in [T]$. Hence, with probability $1 - 4\delta$, $\bar{\mu}_n^t(x, a) \geqslant \mu_n^{\pi^t,p}(x, a)$. Consequently, by replacing it in the previous inequality, we obtain that with probability $1 - 5\delta$,

$$\tau \sum_{t=1}^{T} \sum_{n=1}^{N} \|\widehat{\ell}_n^t\|_\infty^2 \leqslant \tau \sum_{t=1}^{T} \sum_{n=1}^{N} \sum_{x,a} \ell_n^t(x, a)^2 + \frac{N\tau}{2\gamma^2} \log\left(\frac{N}{\delta}\right)$$
$$\leqslant \tau TN|\mathcal{X}||\mathcal{A}| + \frac{N\tau}{2\gamma^2} \log\left(\frac{N}{\delta}\right),$$

where for the last inequality we use that $\ell_n^t \in [0, 1]$.

Therefore, by replacing it in the upper bound of the terms $A$ and $C$ in Eqs. (44) and (45) respectively, we obtain that

$$A + C \leqslant \tau 4bTN|\mathcal{X}||\mathcal{A}| + \frac{N\tau}{\gamma^2} \log\left(\frac{N}{\delta}\right) + \frac{3eN|\mathcal{X}||\mathcal{A}| \log(T)}{2\tau} + \frac{2N}{\tau} \sum_{t=1}^{T} \alpha_t.$$

The upper bound on term $B$ from Eq. (33) in App. C remains the same:

$$B \leqslant \frac{N}{\tau} \log(|\mathcal{A}|) + \frac{eN^2|\mathcal{X}|}{\tau} \log\left(\frac{|\mathcal{A}|}{\min_{t\in[T]} \alpha_t}\right) \log(T) + \frac{2N}{\tau} \sum_{t=1}^{T} \alpha_t.$$

Thus, setting $\alpha_t = 1/(t + 1)$, the final upper bound on the Mirror Descent term is, with high probability, given by

$$\begin{aligned}&\sum_{t=1}^{T} \langle \widehat{\ell}^t - b^t, \mu^{\pi^t,\widehat{p}^t} - \mu^{\pi,\widehat{p}^t} \rangle \leqslant A + B + C \\ &\leqslant \tau 4bTN|\mathcal{X}||\mathcal{A}| + \frac{N\tau}{\gamma^2} \log\left(\frac{N}{\delta}\right) + 6\frac{eN|\mathcal{X}||\mathcal{A}| \log(T)}{\tau} + \frac{N}{\tau} \log(|\mathcal{A}|) + \frac{eN^2|\mathcal{X}|}{\tau} \log(|\mathcal{A}|T) \log(T).\end{aligned} \tag{46}$$

Before tuning the optimal parameter $\tau$, we must first analyze the bias terms.

**Bias terms.** We now proceed to analyze the bias terms. Our approach is similar to the one used in (Jin et al., 2020), with a key difference: they utilize confidence sets in their Mirror Descent iterations, whereas we perform iterations over the set induced by $\widehat{p}^t$. We start by dividing the bias term in two:

$$\sum_{t=1}^{T} \langle \ell^t - \widehat{\ell}^t, \mu^{\pi^t,\widehat{p}^t} - \mu^{\pi,\widehat{p}^t} \rangle = \underbrace{\sum_{t=1}^{T} \langle \ell^t - \widehat{\ell}^t, \mu^{\pi^t,\widehat{p}^t} \rangle}_{\text{Bias 1}} + \underbrace{\sum_{t=1}^{T} \langle \widehat{\ell}^t - \ell^t, \mu^{\pi,\widehat{p}^t} \rangle}_{\text{Bias 2}}.$$

**Bias 1.** Since $\mu^{\pi^t,\widehat{p}^t}$ is $\mathcal{F}^t$-measurable, we have that $\mathbb{E}_t[\langle \ell^t - \widehat{\ell}^t, \mu^{\pi^t,\widehat{p}^t} \rangle] = \langle \mathbb{E}_t[\ell^t - \widehat{\ell}^t], \mu^{\pi^t,\widehat{p}^t} \rangle$. For any couple $(x, a)$, and for any time step $n \in [N]$,

$$\mathbb{E}_t[\ell_n^t(x, a) - \widehat{\ell}_n^t(x, a)] = \ell_n^t(x, a) - \frac{\ell_n^t(x, a)\mu_n^{\pi^t,p}(x, a)}{\bar{\mu}_n^t(x, a) + \gamma} = \ell_n^t(x, a)\left(\frac{\bar{\mu}_n^t(x, a) + \gamma - \mu_n^{\pi^t,p}(x, a)}{\bar{\mu}_n^t(x, a) + \gamma}\right).$$

Hence,

$$\mathbb{E}_t[\text{Bias 1}] = \sum_{t=1}^{T} \sum_{n=1}^{N} \sum_{x,a} \mu_n^{\pi^t,\widehat{p}^t}(x, a)\ell_n^t(x, a)\left(\frac{\bar{\mu}_n^t(x, a) + \gamma - \mu_n^{\pi^t,p}(x, a)}{\bar{\mu}_n^t(x, a) + \gamma}\right).$$

From Lem. D.2, and from the definition of $\bar{\mu}$, we have that with high probability, $\bar{\mu}_n^t(x, a) \geqslant \mu_n^{\pi^t,\widehat{p}^t}(x, a)$, therefore,

$$\mathbb{E}_t[\text{Bias 1}] \leqslant \sum_{t=1}^{T} \sum_{n=1}^{N} \sum_{x,a} |\bar{\mu}_n^t(x, a) + \gamma - \mu_n^{\pi^t,p}(x, a)| \leqslant \sum_{t=1}^{T} \sum_{n=1}^{N} \sum_{x,a} |\bar{\mu}_n^t(x, a) - \mu_n^{\pi^t,p}(x, a)| + \gamma|\mathcal{X}||\mathcal{A}|NT.$$

Note that $\bar{\mu}_n^t(x,a) = \max_{p\in\Omega^t} \mu_n^{\pi^t,p}(x,a) = \pi_n^t(a|x)\max_{p\in\Omega^t} \rho_n^{\pi^t,p}(x)$, where $\rho_n^{\pi^t,p}(x) := \sum_{a\in\mathcal{A}} \mu_n^{\pi^t,p}(x,a)$. Therefore, for each $x \in \mathcal{X}$, there is a $p^{x,t} \in \Omega^t$ such that $\bar{\mu}_n^t(x,a) = \mu_n^{\pi^t,p^{x,t}}(x,a)$.

From Lem. D.3 we obtain that

$$\sum_{t=1}^T \sum_{n=1}^N \sum_{x,a} |\mu_n^{\pi^t,p^{x,t}}(x,a) - \mu_n^{\pi^t,p}(x,a)| = O\bigg(N^2|\mathcal{X}|\sqrt{|\mathcal{A}|T\log\Big(\frac{|\mathcal{X}||\mathcal{A}|NT}{\delta}\Big)}\bigg).$$

Therefore,

$$\mathbb{E}_t[\text{Bias 1}] = O\bigg(N^2|\mathcal{X}|\sqrt{|\mathcal{A}|T\log\Big(\frac{|\mathcal{X}||\mathcal{A}|NT}{\delta}\Big)}\bigg) + \gamma|\mathcal{X}||\mathcal{A}|NT.$$

As we have that Bias 1 $= \mathbb{E}_t[\text{Bias 1}] + \sum_{t=1}^T \langle \mathbb{E}_t[\hat{\ell}^t] - \hat{\ell}^t, \mu^{\pi^t,\hat{p}^t}\rangle$, all that remains is to treat the second term of the sum. With high probability, $\bar{\mu}_n^t(x,a) \geqslant \mu_n^{\pi^t,\hat{p}^t}(x,a)$, therefore

$$\sum_{n=1}^N \sum_{x,a} \hat{\ell}_n^t(x,a)\mu_n^{\pi^t,\hat{p}^t}(x,a) \leqslant \sum_{n=1}^N \sum_{x,a} \ell_n^t(x,a)\mathbb{1}_{\{x_n^t = x, a_n^t = a\}} \leqslant N.$$

Thus, Azuma's inequality gives us that

$$\sum_{t=1}^T \langle \mathbb{E}_t[\hat{\ell}^t] - \hat{\ell}^t, \mu^{\pi^t,\hat{p}^t}\rangle \leqslant N\sqrt{2T\log\Big(\frac{1}{\delta}\Big)},$$

which is of a smaller order than the terms previously appearing in the bias bound. Hence,

$$\text{Bias 1} = O\bigg(N^2|\mathcal{X}|\sqrt{|\mathcal{A}|T\log\Big(\frac{|\mathcal{X}||\mathcal{A}|NT}{\delta}\Big)}\bigg) + \gamma|\mathcal{X}||\mathcal{A}|NT.$$

**Bias 2.** The result follows directly from Lem. 14 of (Jin et al., 2020) using $\mu^{\pi,\hat{p}^t}$ instead of $\mu^{\pi,p}$:

$$\text{Bias 2} = \sum_{t=1}^T \langle \hat{\ell}^t - \ell^t, \mu^{\pi,\hat{p}^t}\rangle = O\bigg(\frac{N\log(\frac{|\mathcal{X}||\mathcal{A}|N}{\delta})}{\gamma}\bigg).$$

**Optimizing the learning and exploration parameters $\tau$ and $\gamma$.** By joinning the Mirror Descent term from Eq. (46) along with the bounds on Bias 1 and Bias 2 terms, and setting $\gamma = \tau$, we obtain that

$$\begin{aligned} R_T^{\text{policy/MD}} = O\bigg(&\tau bN|\mathcal{X}||\mathcal{A}|T + \frac{N}{\tau}\Big[\log\Big(\frac{N}{\delta}\Big) + |\mathcal{X}||\mathcal{A}|\log(T) + \log(|\mathcal{A}|) + N|\mathcal{X}|\log(|\mathcal{A}|T)\log(T)\Big] \\ &+ N^2|\mathcal{X}|\sqrt{|\mathcal{A}|T\log\Big(\frac{|\mathcal{X}||\mathcal{A}|NT}{\delta}\Big)} + \tau|\mathcal{X}||\mathcal{A}|NT + \frac{N\log(\frac{|\mathcal{X}||\mathcal{A}|N}{\delta})}{\tau}\bigg). \end{aligned}$$

Let $\varphi_1 := (b+1)N|\mathcal{X}||\mathcal{A}|$, and

$$\varphi_2 := N\bigg[\log\Big(\frac{N}{\delta}\Big) + |\mathcal{X}||\mathcal{A}|\log(T) + \log(|\mathcal{A}|) + N|\mathcal{X}|\log(|\mathcal{A}|T)\log(T) + \log\Big(\frac{|\mathcal{X}||\mathcal{A}|N}{\delta}\Big)\bigg].$$

For $\tau \propto \sqrt{\varphi_2/\varphi_1 T}$, recalling that $b := NC_\delta$, and that $C_\delta := \sqrt{2|\mathcal{X}|\log(|\mathcal{X}||\mathcal{A}|NT/\delta)}$, we obtain that, with high probability,

$$R_T^{\text{policy/MD}} = 2\sqrt{\varphi_1\varphi_2 T} = \tilde{O}\big(N^{3/2}|\mathcal{X}|^{5/4}|\mathcal{A}|\sqrt{T} + N^2|\mathcal{X}|^{5/4}\sqrt{|\mathcal{A}|T}\big). \tag{47}$$

### D.2.3. Term 3: $R_T^{\text{POLICY/MDP}}$

The upper bound for this term directly follows from the analysis of adding the bonus term to compensate for insufficient exploration, as discussed in Subsec. 3.2 of the main paper, and is detailed in App. B.2. Thus, we have that with high probability, $R_T^{\text{policy/MDP}} \leqslant 0$.

D.2.4. TERM 4: BONUS TERM

The analysis of this term follows directly from Prop. 3.1 from the main paper: for any $\delta \in (0,1)$, with probability at least $1-3\delta$, we have that

$$\sum_{t=1}^{T} \langle b^t, \mu^{\pi^t, \hat{p}^t} \rangle = \tilde{O}\big(N^3 |\mathcal{X}|^{3/2} \sqrt{|\mathcal{A}| T}\big). \tag{48}$$

### D.3. Final bound

Replacing the upper bound on all four terms from Eq.s (43), (47), (48), and that $R_T^{\text{policy/MDP}} \leqslant 0$ into the regret decomposition in Eq. (42), we obtain that playing Alg. 2 for the RL problem with bandit feedback on adversarial loss functions has, with high probability, a static regret of order

$$R_T(\pi) = \tilde{O}\big(N^3 |\mathcal{X}|^{3/2} \sqrt{|\mathcal{A}| T} + N^{3/2} |\mathcal{X}|^{5/4} |\mathcal{A}| \sqrt{T}\big).$$

□

## E. Curl with Bandit Feedback

**Notation** Let $S$ and $A$ be two positive integers. For convenience and brevity, we suppose in what follows that $\mathcal{X} = [S]$ and $\mathcal{A} = [A]$. Accordingly, we will often use $S$ and $A$ in place of $|\mathcal{X}|$ and $|\mathcal{A}|$ respectively. Let $\dot{A} \in \{A, A-1\}$. For a vector $\xi \in \mathbb{R}^{NS\dot{A}}$ and $(n,x,a) \in [N] \times [S] \times [\dot{A}]$, we use $\xi_n(x,a)$ as a shorthand for $\xi((n-1)S\dot{A} + (x-1)\dot{A} + a)$. Similarly, let $\ddot{A} \in \{A, A-1\}$. Then, for an $NS\dot{A} \times NS\ddot{A}$ matrix $M$, we use $M(n,x,a,n',x',a')$ to denote the item in row $(n-1)S\dot{A} + (x-1)\dot{A} + a$ and column $(n'-1)S\ddot{A} + (x'-1)\ddot{A} + a'$ of $M$. For any $d \in \mathbb{Z}_+$, let $\mathbb{1}_d \in \mathbb{R}^d$ be the vector with all entries equal to one and $\mathbf{I}_d$ the $d \times d$ identity matrix.

### E.1. An alternative representation for the decision sets

In the following, we will fix an arbitrary transition kernel $p := (p_n)_{n \in [N]}$. We recall the notation that for $\zeta \in \mathcal{M}_{\mu_0}^p$, $\rho_n^\zeta(x) := \sum_{a \in \mathcal{A}} \zeta_n(x,a)$ for $(n,x) \in [N] \times \mathcal{X}$, which satisfies $\rho_n^\zeta(x) = \sum_{x',a' \in \mathcal{X} \times \mathcal{A}} \zeta_{n-1}(x',a') p_n(x|x',a')$ for $n \geqslant 2$. At the first step, we define $\rho_1^p(x) := \sum_{x',a' \in \mathcal{X} \times \mathcal{A}} \mu_0(x',a') p_1(x|x',a')$, which satisfies $\rho_1^p(x) = \rho_1^\zeta(x)$ for every $\zeta \in \mathcal{M}_{\mu_0}^p$ and $x \in \mathcal{X}$ since the initial state distribution is the same for all occupancy measures in $\mathcal{M}_{\mu_0}^p$.

We describe here the mapping alluded to in Sec. 4.2.1 of $\mathcal{M}_{\mu_0}^p$ to a lower-dimensional space where it could have a non-empty interior. This is analogous to how one can define a bijective map between the simplex $\Delta_d$ and the set $\{x \in \mathbb{R}^{d-1} : \mathbb{1}_{d-1}^\intercal x \leqslant 1 \text{ and } x_i \geqslant 0\, \forall i \in [d-1]\}$, which is the intersection of the positive orthant of $R^{d-1}$ with the $L_1$ unit ball, see (Jézéquel et al., 2022, Section 2). This can be done since any coordinate $x_{i*}$ of a vector $x \in \Delta_d$ can be recovered from the rest of the coordinates: $x_{i*} = 1 - \sum_{i \neq i*} x_i$. In our case, denoting by $a^*$ the last action in $\mathcal{A}$ (i.e., $a^* = A$, recalling that $\mathcal{A} = [A]$), we will represent the occupancy measures as vectors in $\mathbb{R}^{NS(A-1)}$ by omitting all coordinates that correspond to this action. We can afford to do so, since for any $\mu \in \mathcal{M}_{\mu_0}^p$, we have that $\mu_n(x,a^*) = \rho_n^\mu(x) - \sum_{a \neq a^*} \mu_n(x,a)$ where $\rho_n^\mu$ is recoverable from $\mu_{n-1}$ and given in the first step by the initial state distribution $\rho_1^p(x)$, which does not depend on $\mu$. In the following, we use this idea to define the sought mapping.

Define the $A \times (A-1)$ matrix

$$G := \begin{bmatrix} \mathbf{I}_{A-1} \\ -\mathbb{1}_{A-1}^\intercal \end{bmatrix}$$

and let $H$ be the $NSA \times NS(A-1)$ matrix obtained via taking the direct sum of $NS$ copies of $G$: $H := \bigoplus_{i=1}^{NS} G$.[2] Define $\boldsymbol{w}^{p,1} \in \mathbb{R}^{NSA}$ such that

$$\boldsymbol{w}_n^{p,1}(x,a) := \rho_1^p(x) \mathbb{I}\{n = 1, a = a^*\}\,.$$

Next, for every $2 \leqslant m \leqslant N$, we define $W^{p,m}$ as the $NSA \times NSA$ matrix where

$$W^{p,m}(n,x,a,n',x',a') := \mathbb{I}\{n = m, n' = m-1, a = a^*\} p_m(x|x',a')\,.$$

[2] For an $n \times m$ matrix $M$ and an $n' \times m'$ matrix $M'$, $M \bigoplus M'$ is the $(n+n') \times (m+m')$ block matrix $\begin{bmatrix} M & \mathbf{0} \\ \mathbf{0} & M' \end{bmatrix}$.

Then, we define the $NSA \times NS(A-1)$ matrix

$$B^p := (\mathbf{I}_{NSA} + W^{p,N}) \dots (\mathbf{I}_{NSA} + W^{p,3})(\mathbf{I}_{NSA} + W^{p,2})H$$

and the vector

$$\boldsymbol{\beta}^p := (\mathbf{I}_{NSA} + W^{p,N}) \dots (\mathbf{I}_{NSA} + W^{p,3})(\mathbf{I}_{NSA} + W^{p,2})\boldsymbol{w}^{p,1} .$$

Finally, define the function $\Xi_p \colon \mathbb{R}^{NS(A-1)} \to \mathbb{R}^{NSA}$ where

$$\Xi_p(\xi) := B^p \xi + \boldsymbol{\beta}^p$$

for $\xi \in \mathbb{R}^{NS(A-1)}$.

To explain the semantics of $\Xi_p$, let $\mu \in \mathcal{M}^p_{\mu_0}$ and $\tilde{\mu} \in \mathbb{R}^{NS(A-1)}$ be such that $\tilde{\mu}_n(x,a) := \mu_n(x,a)$ for all $(n,x,a) \in [N] \times \mathcal{X} \times \mathcal{A}\backslash a^*$. It then holds that $\Xi_p(\tilde{\mu}) = \mu$. To see this, note that $H\tilde{\mu}$ expands $\tilde{\mu}$ setting $(H\tilde{\mu})_n(x,a^*) = -\sum_{a \neq a^*} \mu_n(x,a)$. To fully recover $\mu_n(x,a^*)$, what remains is to add $\rho^\mu_n(x)$. This is achieved at $n=1$ by adding $\boldsymbol{w}^{p,1}$ to $H\tilde{\mu}$ since $\boldsymbol{w}^{p,1}_n(x,a^*) = \rho^p_1(x) = \rho^\mu_1(x)$ and $\boldsymbol{w}^{p,1}_n(x,a) = 0$ for $a \neq a^*$. Next, at $n=2$, the matrix $W^{p,2}$ extracts the values $\rho^\mu_2(x)$ when operated on $H\tilde{\mu} + \boldsymbol{w}^{p,1}$ such that $\mu_2(x,a^*)$ is recovered at coordinate $(2,x,a^*)$ of $(\mathbf{I}_{NSA} + W^{p,2})(H\tilde{\mu} + \boldsymbol{w}^{p,1})$. Iterating this procedure until step $N$ allows us to fully recover $\mu$ from $\tilde{\mu}$. While for a generic $\xi \in \mathbb{R}^{NS(A-1)}$, $\Xi_p(\xi)$ is the unique vector in $\mathbb{R}^{NSA}$ satisfying $(\Xi_p(\xi))_n(x,a) = \xi_n(x,a)$ for all $n, x$, and $a \neq a^*$; $(\Xi_p(\xi))_1(x,a^*) = \rho^p_1(x) - \sum_{a \neq a^*}(\Xi_p(\xi))_1(x,a)$ for all $x$; and $(\Xi_p(\xi))_n(x,a^*) = \sum_{x',a'}(\Xi_p(\xi))_{n-1}(x',a')p_n(x|x',a') - \sum_{a \neq a^*}(\Xi_p(\xi))_n(x,a)$ for all $x$ and $n \geqslant 2$.

Note that $B^p$ has full column rank since for any $\xi \in \mathbb{R}^{NS(A-1)}$, $B^p\xi$ is only an expansion of $\xi$; hence, we can define its left pseudo-inverse $(B^p)^+ := ((B^p)^\intercal B^p)^{-1}(B^p)^\intercal$, which satisfies $(B^p)^+ B^p = \mathbf{I}_{NS(A-1)}$. On the other hand, the matrix $B^p(B^p)^+$ projects vectors in $\mathbb{R}^{NSA}$ onto the column space of $B^p$, which is given by

$$\left\{\mu \in \mathbb{R}^{NSA} : \sum_a \mu_n(x,a) = \sum_{x',a'} \mu_{n-1}(x',a')p_n(x|x',a') \; \forall x \in \mathcal{X}, 2 \leqslant n \leqslant N \text{ and } \sum_a \mu_1(x,a) = 0 \; \forall x \in \mathcal{X}\right\}. \quad (49)$$

It is easy to verify that for any $\mu, \mu' \in \mathcal{M}^p_{\mu_0}$, $\mu - \mu'$ lies in the column space of $B^p$ (recall that $\sum_a \mu_1(x,a) = \sum_a \mu'_1(x,a) = \rho^p_1(x)$). Moreover, $\boldsymbol{\beta}^p \in \mathcal{M}^p_{\mu_0}$ as it corresponds to a policy $\pi$ where $\pi_n(a^*|x) = 1$ for all $n$ and $x$. Therefore, for any $\mu \in \mathcal{M}^p_{\mu_0}$, $\mu - \boldsymbol{\beta}^p$ belongs to the column space of $B^p$, and we consequently have that

$$\Xi_p\big((B^p)^+(\mu - \boldsymbol{\beta}^p)\big) = B^p(B^p)^+(\mu - \boldsymbol{\beta}^p) + \boldsymbol{\beta}^p = \mu .$$

Hence, by the definition of $\Xi_p$, $(B^p)^+(\mu - \boldsymbol{\beta}^p)$ coincides with $\mu$ on all coordinates $(n,x,a) \in [N] \times \mathcal{X} \times \mathcal{A}\backslash\{a^*\}$ (since the map $\Xi_p$ only expands the input vector adding the coordinates corresponding to action $a^*$), and is then the only point in $\mathbb{R}^{NS(A-1)}$ that $\Xi_p$ maps to $\mu$. In light of this, we define

$$(\mathcal{M}^p_{\mu_0})^- := \{\xi \in \mathbb{R}^{NS(A-1)} \mid \Xi_p(\xi) \in \mathcal{M}^p_{\mu_0}\} ,$$

the pre-image of $\mathcal{M}^p_{\mu_0}$ under $\Xi_p$. Accordingly, we define $\Lambda_p \colon (\mathcal{M}^p_{\mu_0})^- \to \mathcal{M}^p_{\mu_0}$ as the restriction of $\Xi_p$ to $(\mathcal{M}^p_{\mu_0})^-$; that is,

$$\Lambda_p := \Xi_p|_{(\mathcal{M}^p_{\mu_0})^-} .$$

This then is a bijective function, with $\Lambda_p^{-1}(\mu) = (B^p)^+(\mu - \boldsymbol{\beta}^p)$.

Still, $(\mathcal{M}^p_{\mu_0})^-$ is not guaranteed to have a non-empty interior. Suppose that some state $x^*$ is not reachable at a certain step $n^*$; that is, for every state $x$ and action $a$, $p_{n^*}(x^*|x,a) = 0$ if $n^* \geqslant 2$, or just that $\rho^p_1(x^*) = 0$ if $n^* = 1$. Then, for any $\mu \in \mathcal{M}^p_{\mu_0}$, $\mu_{n^*}(x^*,a) = 0$ for every action $a$. This implies that for every $\xi \in (\mathcal{M}^p_{\mu_0})^-$, $\xi_{n^*}(x^*,a) = 0$ for all $a \neq a^*$ (since these coordinates are preserved under $\Lambda_p$), and hence, $(\mathcal{M}^p_{\mu_0})^-$ has an empty interior. To remedy this, we rely on Asm. 4.4, which is equivalent to imposing that for every state $x$, $\rho^p_1(x) > 0$ and there exists for every step $n$ a state-action pair $(x',a')$ such that $p_{n+1}(x|x',a') > 0$. We show next that this condition is sufficient for $(\mathcal{M}^p_{\mu_0})^-$ to have a non-empty interior. We first present an alternative characterization of $(\mathcal{M}^p_{\mu_0})^-$.

**Lemma E.1.** *It holds that*

$$(\mathcal{M}^p_{\mu_0})^- = \left\{\xi \in \mathbb{R}^{NS(A-1)} \colon B^p\xi \geqslant -\boldsymbol{\beta}^p\right\} .$$

Hence, $(\mathcal{M}^p_{\mu_0})^-$ is a polyhedral set formed by $NSA$ constraints; namely that for $n, x, a \in [N] \times \mathcal{X} \times \mathcal{A}$, $B^p(n, x, a, \cdot, \cdot, \cdot)^\intercal \xi + \boldsymbol{\beta}^p_n(x, a) \geqslant 0$.

*Proof.* Any $\xi \in (\mathcal{M}^p_{\mu_0})^-$ clearly satisfies $B^p\xi \geqslant -\boldsymbol{\beta}^p$ since $B^p\xi + \boldsymbol{\beta}^p = \Lambda_p(\xi) \in \mathcal{M}^p_{\mu_0}$, whose coordinates are non-negative. Conversely, assume that $B^p\xi \geqslant -\boldsymbol{\beta}^p$ for some $\xi \in \mathbb{R}^{NS(A-1)}$, and let $\mu := \Xi_p(\xi) = B^p\xi + \boldsymbol{\beta}^p$. Showing that $\xi \in (\mathcal{M}^p_{\mu_0})^-$ is equivalent, by definition, to showing that $\Xi_p(\xi) \in \mathcal{M}^p_{\mu_0}$. Since $\boldsymbol{\beta}^p \in \mathcal{M}^p_{\mu_0}$ and $B^p\xi$ belongs to the column space of $B^p$ specified in (49), it holds that

$$\sum_a \mu_n(x, a) = \sum_{x', a'} \mu_{n-1}(x', a') p_n(x|x', a')$$

for every $n \geqslant 2$, and that $\sum_a \mu_1(x, a) = \sum_a \boldsymbol{\beta}^p_1(x, a) = \rho_1^{\boldsymbol{\beta}^p}(x) = \rho^p_1(x)$. Then, to show that $\mu \in \mathcal{M}^p_{\mu_0}$, it remains to show that $\mu \in (\Delta_{\mathcal{X}\times\mathcal{A}})^N$. By assumption, $\mu$ only has non-negative coordinates; therefore, we only have to show that $\sum_{x,a} \mu_n(x, a) = 1$ at every $n$. This easily done via induction: $\sum_{x,a} \mu_1(x, a) = \sum_x \rho^p_1(x) = 1$, and for $n \geqslant 2$,

$$\sum_{x,a} \mu_n(x, a) = \sum_x \sum_{x', a'} \mu_{n-1}(x', a') p_n(x|x', a') = \sum_{x', a'} \mu_{n-1}(x', a') \,.$$

□

**Lemma E.2.** $(\mathcal{M}^p_{\mu_0})^-$ *has a non-empty interior if and only if Asm. 4.4 holds.*

*Proof.* Necessity is immediate as argued before. We prove sufficiency utilizing an argument from the proof of Proposition 2.3 in (Wolsey & Nemhauser, 1999). For every step-state-action triple $(n, x, a)$, it is easy to verify that Asm. 4.4 implies the existence of some $\mu \in \mathcal{M}^p_{\mu_0}$ such that $\mu_n(x, a) > 0$. Taking a convex combination with full support of one such occupancy measure for every $(n, x, a)$ results, via the convexity of $\mathcal{M}^p_{\mu_0}$, in an occupancy measure $\mu^* \in \mathcal{M}^p_{\mu_0}$ whose entries are all strictly positive. Hence, $\xi^* := \Lambda_p^{-1}(\mu^*)$ is an interior point of the polyhedral set $(\mathcal{M}^p_{\mu_0})^-$ as it satisfies with strict inequality all the constraints defining it. □

## E.2. Entropic Regularization Approach

### E.2.1. Fitting a Euclidean ball in the constraint set

For the following, fix $\varepsilon \in (0, 1/S)$. From Sec. 4.2.1, recall the definition $\kappa := \varepsilon/(A - 1 + \sqrt{A-1})$. We now show that $\kappa \mathbb{1}_{NS(A-1)} + \kappa \boldsymbol{v} \in (\mathcal{M}^p_{\mu_0})^-$ for any $\boldsymbol{v} \in \mathbb{B}^{NS(A-1)}$ assuming the transition kernel $p := (p_n)_{n\in[N]}$ satisfies the condition of Asm. 4.2; that is, $p_n(x'|x, a) \geqslant \varepsilon$ for all $(n, x, x', a) \in [N] \times \mathcal{X}^2 \times \mathcal{A}$. Take $\zeta^{\boldsymbol{v},p} := \Xi_p(\kappa \mathbb{1}_{NS(A-1)} + \kappa \boldsymbol{v})$. Note that showing that $\kappa \mathbb{1}_{NS(A-1)} + \kappa \boldsymbol{v} \in (\mathcal{M}^p_{\mu_0})^-$ is equivalent to showing that $\zeta^{\boldsymbol{v},p} \in \mathcal{M}^p_{\mu_0}$. In the following, we proceed with the latter.

Note that via Lem. E.1, it suffices to show that $\zeta^{\boldsymbol{v},p}$ is non-negative. We use induction in the following to show more particularly that $\zeta^{\boldsymbol{v},p} \in (\Delta_{\mathcal{X}\times\mathcal{A}})^N$. By the definition of $\zeta^{\boldsymbol{v},p}_n$, we have that for $(n, x) \in [N] \times \mathcal{X}$,

$$\zeta^{\boldsymbol{v},p}_n(x, a) = \frac{\varepsilon}{A - 1 + \sqrt{A-1}}(1 + \boldsymbol{v}_n(x, a)) \; \forall a \in \mathcal{A}\backslash a^* \text{ and } \zeta^{\boldsymbol{v},p}_n(x, a^*) = \rho_n^{\zeta^{\boldsymbol{v},p}}(x) - \sum_{a \neq a^*} \zeta^{\boldsymbol{v},p}_n(x, a) \,, \tag{50}$$

where $\rho_n^{\zeta^{\boldsymbol{v},p}}(x) = \sum_{a', x' \in \mathcal{A}\times\mathcal{X}} \zeta^{\boldsymbol{v},p}_{n-1}(x', a') p_n(x|x', a')$ for $n \geqslant 2$ and $\rho_1^{\zeta^{\boldsymbol{v},p}}(x) = \sum_{a', x' \in \mathcal{A}\times\mathcal{X}} \mu_0(x', a') p_1(x|x', a') = \rho^p_1(x)$. For $a \neq a^*$, clearly $\zeta^{\boldsymbol{v},p}_n(x, a) \geqslant 0$ as $\boldsymbol{v}_n(x, a) \geqslant -1$. Note that at any step $n$ and state $x$, the Cauchy-Schwarz inequality and the fact that $\boldsymbol{v} \in \mathbb{B}^{NS(A-1)}$ yield that

$$\sum_{a \neq a^*} \boldsymbol{v}_n(x, a) \leqslant \sqrt{A-1} \sqrt{\sum_{a \neq a^*} |\boldsymbol{v}_n(x, a)|^2} \leqslant \sqrt{A-1} \,.$$

Hence,

$$\sum_{a \neq a^*} \zeta^{\boldsymbol{v},p}_n(x, a) = \sum_{a \neq a^*} \frac{\varepsilon}{A - 1 + \sqrt{A-1}}(1 + \boldsymbol{v}_n(x, a)) \leqslant \varepsilon \,.$$

On the other hand, Asm. 4.2 implies that $\rho_1^{\zeta_1^{\boldsymbol{v},p}}(x) \geqslant \varepsilon$ for every $x$. Hence, (50) gives that $\zeta_1^{\boldsymbol{v},p}(x,a^*) \geqslant 0$ at every $x$. Moreover, (50) also implies that $\sum_{x,a} \zeta_1^{\boldsymbol{v},p}(x,a) = \sum_x \rho_1^p(x) = 1$, yielding that $\zeta_1^{\boldsymbol{v},p} \in \Delta_{\mathcal{X}\times\mathcal{A}}$. For $n \geqslant 2$, assuming that $\zeta_{n-1}^{\boldsymbol{v},p} \in \Delta_{\mathcal{X}\times\mathcal{A}}$, Asm. 4.2 implies again that $\rho_n^{\zeta^{\boldsymbol{v},p}}(x) \geqslant \varepsilon$ for every $x$. We then get via (50) that $\zeta_n^{\boldsymbol{v},p}(x,a^*) \geqslant 0$ and that $\zeta_n^{\boldsymbol{v},p} \in \Delta_{\mathcal{X}\times\mathcal{A}}$ since $\sum_{x,a} \zeta_n^{\boldsymbol{v},p}(x,a) = \sum_x \rho_n^{\zeta^{\boldsymbol{v},p}}(x) = 1$, which holds again via (50) and the assumption that $\zeta_{n-1}^{\boldsymbol{v},p} \in \Delta_{\mathcal{X}\times\mathcal{A}}$. Induction then establishes that $\zeta^{\boldsymbol{v},p} \in (\Delta_{\mathcal{X}\times\mathcal{A}})^N$ as sought. As mentioned above, this implies via Lem. E.1 that $\zeta^{\boldsymbol{v},p} \in \mathcal{M}_{\mu_0}^p$, or equivalently, that $\kappa \mathbb{1}_{NS(A-1)} + \kappa \boldsymbol{v} \in (\mathcal{M}_{\mu_0}^p)^-$ and $\zeta^{\boldsymbol{v},p} = \Lambda_p(\kappa \mathbb{1}_{NS(A-1)} + \kappa \boldsymbol{v})$.

### E.2.2. Estimating the transition kernel

In this section, we define and analyze an alternative transition kernel estimator to the one given in Eq. (7). What we seek in this new estimator is *(1)* that it estimates well the true transition kernel, with a guarantee similar to that of Lem. 2.2; *(2)* that it drifts across rounds in a controlled manner, satisfying the bound of Lem. A.3 up to a constant; and *(3)* that, at the same time, it satisfies the condition of Asm. 4.2 almost surely, supposing, naturally, that it is satisfied by the true kernel.

To recall the notation, for each round $t \in [T]$, $o^t$ denotes a random trajectory obtained by executing the policy $\pi^t$ in the environment; that is, $o^t := (x_1^t, a_1^t, \ldots, x_N^t, a_N^t)$ where $a_n^t \sim \pi^t(\cdot|x_n^t)$ and $x_n^t \sim p_n(\cdot|x_{n-1}^t, a_{n-1}^t)$.[3] We also recall the definitions

$$N_n^t(x,a) := \sum_{s=1}^{t-1} \mathbb{1}_{\{x_n^s = x, a_n^s = a\}} \qquad \text{and} \qquad M_n^t(x'|x,a) := \sum_{s=1}^{t-1} \mathbb{1}_{\{x_{n+1}^s = x', x_n^s = x, a_n^s = a\}} .$$

Fix $n, x, a \in [N] \times \mathcal{X} \times \mathcal{A}$. As an intermediate step, we compute at the beginning of each round $t$ the Laplace (add-one) estimator for $p_n^t(\cdot|x,a)$; that is, for $x' \in \mathcal{X}$,

$$\tilde{p}_n^t(x'|x,a) := \frac{M_{n-1}^t(x'|x,a) + 1}{N_{n-1}^t(x,a) + S} . \tag{51}$$

To obtain a guarantee on the accuracy of this estimator, we firstly describe a slightly different setting. Let $(\dot{x}_n^s)_{s=1}^T$ be an i.i.d. sequence of states such that $\dot{x}_n^s \sim p_n(\cdot|x,a)$. Then, for $k \in [T]$, we define the Laplace estimator

$$\dot{p}_n^k(x'|x,a) := \frac{1 + \sum_{s=1}^k \mathbb{1}_{\{\dot{x}_n^s = x'\}}}{k + S} .$$

Notice that in our setting, the distribution $\tilde{p}_n^t(\cdot|x,a)$ is equivalent to $\dot{p}_n^{N_{n-1}^t(x,a)}(\cdot|x,a)$, keeping in mind that the number of samples $N_{n-1}^t(x,a)$ is random and dependent on the observed samples. Let $D_{\mathrm{KL}}(p \,\|\, q)$ denote the KL-divergence between distributions (probability mass functions) $p$ and $q$. We derive the following result concerning the divergence between $\tilde{p}^t$ and $p$ using known properties of the Laplace estimator and a union bound argument.

**Lemma E.3.** *For fixed $t, n, x, a \in [T] \times [N] \times \mathcal{X} \times \mathcal{A}$, it holds with probability at least $1 - \delta$ that*

$$D_{\mathrm{KL}}\big(p_n(\cdot|x,a) \,\|\, \tilde{p}_n^t(\cdot|x,a)\big) \leqslant \frac{161S + 6\sqrt{S}\log^{5/2}\frac{ST}{4\delta} + 310}{\max\{1, N_{n-1}^t(x,a)\}} .$$

*Proof.* For a fixed $k \in [T]$, Thm. 2 in (Canonne et al., 2023) and Prop. 1 in (Mourtada & Gaïffas, 2022) imply that

$$P\left(D_{\mathrm{KL}}\big(p_n(\cdot|x,a) \,\|\, \dot{p}_n^k(\cdot|x,a)\big) > \frac{161S + 6\sqrt{S}\log^{5/2}\frac{S}{4\delta} + 310}{k}\right) \leqslant \delta .$$

Via a union bound, we obtain that[4]

$$P\left(D_{\mathrm{KL}}\big(p_n(\cdot|x,a) \,\|\, \tilde{p}_n^t(\cdot|x,a)\big) > \frac{161S + 6\sqrt{S}\log^{5/2}\frac{S}{4\delta} + 310}{\max\{1, N_{n-1}^t(x,a)\}}\right)$$

[3] Recall that $(x_0^t, a_0^t) \sim \mu_0(\cdot,\cdot)$.

[4] Note that if $N_{n-1}^t(x,a) = 0$, then $\tilde{p}_n^t(\cdot|x,a)$ is the uniform distribution and $D_{\mathrm{KL}}\big(p_n(\cdot|x,a) \,\|\, \tilde{p}_n^t(\cdot|x,a)\big) \leqslant \log S$; hence, the bound trivially holds.

$$\leqslant P\left(\exists k \in [T]:\ D_{\mathrm{KL}}\big(p_n(\cdot|x,a) \,\|\, \dot{p}_n^k(\cdot|x,a)\big) > \frac{161S + 6\sqrt{S}\log^{5/2}\frac{S}{4\delta} + 310}{k}\right) \leqslant \delta T\,.$$

The lemma then follows after rescaling $\delta$. ☐

Note that the distribution $\tilde{p}_n^t(\cdot|x,a)$ does not necessarily satisfy the conditions of Asm. 4.2 uniformly. Next, we define for a given $\varepsilon \in [0, 1/S]$ the set

$$\Delta_{\mathcal{X}}^{\varepsilon} := \{x \in \mathbb{R}^d : \mathbb{1}_d^{\mathsf{T}} x = 1 \text{ and } x_i \geqslant \varepsilon\ \forall i \in [d]\} \subseteq \Delta_{\mathcal{X}}\,,$$

which is the set of state distribution assigning probability at least $\varepsilon$ to every state. We then define $\widehat{p}_n^t(\cdot|x,a)$ as the information projection of $\tilde{p}_n^t(\cdot|x,a)$ onto $\Delta_{\mathcal{X}}^{\varepsilon}$; that is,

$$\widehat{p}_n^t(\cdot|x,a) := \operatorname*{arg\,min}_{q \in \Delta_{\mathcal{X}}^{\varepsilon}} D_{\mathrm{KL}}\big(q \,\|\, \tilde{p}_n^t(\cdot|x,a)\big)\,, \tag{52}$$

which exists and is unique since $\Delta_{\mathcal{X}}^{\varepsilon}$ is compact and $D_{\mathrm{KL}}\big(\cdot \,\|\, \tilde{p}_n^t(\cdot|x,a)\big)$ is continuous and strictly convex where it is finite (note that $\tilde{p}_n^t(\cdot|x,a)$ never assigns zero probability to any state; hence, $D_{\mathrm{KL}}\big(q \,\|\, \tilde{p}_n^t(\cdot|x,a)\big)$ is finite for any $q \in \Delta_{\mathcal{X}}^{\varepsilon}$). If $\tilde{p}_n^t(\cdot|x,a)$ is not already in $\Delta_{\mathcal{X}}^{\varepsilon}$, this projection can only bring us closer to $p_n(\cdot|x,a)$ in the $KL$-divergence sense as the following inequality (Cover & Thomas, 2012, Thm. 11.6.1) states:

$$D_{\mathrm{KL}}(p_n(\cdot|x,a) \,\|\, \widehat{p}_n^t(\cdot|x,a)) \leqslant D_{\mathrm{KL}}(p_n(\cdot|x,a) \,\|\, \tilde{p}_n^t(\cdot|x,a)) - D_{\mathrm{KL}}(\widehat{p}_n^t(\cdot|x,a) \,\|\, \tilde{p}_n^t(\cdot|x,a))\,. \tag{53}$$

With this fact in mind, we can arrive at the following result, a parallel of Lem. 2.2.

**Lemma E.4.** *With probability at least $1-\delta$, it holds for all $t, n, x, a \in [T] \times [N] \times \mathcal{X} \times \mathcal{A}$ simultaneously that*

$$\|p_n(\cdot|x,a) - \widehat{p}_n^t(\cdot|x,a)\|_1 \leqslant \sqrt{\frac{322S + 12\sqrt{S}\log^{5/2}\frac{S^2ANT^2}{4\delta} + 620}{\max\{1, N_{n-1}^t(x,a)\}}}\,.$$

*Proof.* The statement is a consequence of (53), Lem. E.3, and Pinsker's inequality; followed by an application of a union bound over all rounds, steps, and state-action pairs. ☐

What remains now is to show that there exists a constant $c > 0$ such that

$$\|\widehat{p}_{n+1}^{t+1}(\cdot|x,a) - \widehat{p}_{n+1}^t(\cdot|x,a)\|_1 \leqslant c\frac{\mathbb{1}_{\{x_n^t = x, a_t^s = a\}}}{\max\{1, N_n^{t+1}(x,a)\}}\,.$$

This can be easily shown to hold for $\tilde{p}^t$, i.e., before the projection step, as states the following lemma.

**Lemma E.5.** *For all $n \in [N-1]$, $(x, a, x') \in \mathcal{X} \times \mathcal{A} \times \mathcal{X}$, and $t \in [T]$; $\tilde{p}_{n+1}^t(x'|x,a)$ as defined in (51) satisfies*

$$\|\tilde{p}_{n+1}^{t+1}(\cdot|x,a) - \tilde{p}_{n+1}^t(\cdot|x,a)\|_1 \leqslant \frac{2\mathbb{1}_{\{x_n^t = x, a_n^t = a\}}}{N_n^{t+1}(x,a) + S}\,.$$

*Proof.* The derivation follows along the same lines as the proof of Lem. A.3. We have that

$$\begin{aligned}
\tilde{p}_{n+1}^{t+1}(x'|x,a) &= \frac{\mathbb{1}_{\{x_{n+1}^t = x', x_n^t = x, a_n^t = a\}} + M_n^t(x'|x,a) + 1}{N_n^{t+1}(x,a) + S} \\
&= \frac{\mathbb{1}_{\{x_{n+1}^t = x', x_n^t = x, a_n^t = a\}}}{N_n^{t+1}(x,a) + S} + \frac{N_n^t(x,a) + S}{N_n^{t+1}(x,a) + S}\tilde{p}_{n+1}^t(x'|x,a)\,.
\end{aligned}$$

Hence,

$$\begin{aligned}
\tilde{p}_{n+1}^{t+1}(x'|x,a) - \tilde{p}_{n+1}^t(x'|x,a) &= \frac{\mathbb{1}_{\{x_{n+1}^t = x', x_n^t = x, a_n^t = a\}}}{N_n^{t+1}(x,a) + S} + \tilde{p}_{n+1}^t(x'|x,a)\frac{N_n^t(x,a) - N_n^{t+1}(x,a)}{N_n^{t+1}(x,a) + S} \\
&= \frac{\mathbb{1}_{\{x_n^t = x, a_n^t = a\}}}{N_n^{t+1}(x,a) + S}\big(\mathbb{1}_{\{x_{n+1}^t = x'\}} - \tilde{p}_{n+1}^t(x'|x,a)\big)\,.
\end{aligned}$$

Finally, we conclude that

$$\begin{aligned}\|\tilde{p}^{t+1}_{n+1}(\cdot|x,a) - \tilde{p}^{t}_{n+1}(\cdot|x,a)\|_1 &= 2 \sum_{x' : \tilde{p}^{t+1}_{n+1}(x'|x,a) \geqslant \tilde{p}^{t}_{n+1}(x'|x,a)} \big(\tilde{p}^{t+1}_{n+1}(x'|x,a) - \tilde{p}^{t}_{n+1}(x'|x,a)\big) \\ &= \frac{2\mathbb{1}_{\{x^t_n = x, a^t_n = a\}}}{N^{t+1}_n(x,a) + S}\big(1 - \tilde{p}^{t}_{n+1}(x^t_{n+1}|x,a)\big) \leqslant \frac{2\mathbb{1}_{\{x^t_n = x, a^t_n = a\}}}{N^{t+1}_n(x,a) + S} .\end{aligned}$$

□

To derive a similar bound for the projected estimator $\widehat{p}^t$, we firstly derive a more explicit characterization of the information projection onto $\Delta^\varepsilon_{\mathcal{X}}$. For a fixed $\varepsilon \in (0, 1/S)$, define the function $g_\varepsilon \colon \mathbb{R} \times \Delta_{\mathcal{X}} \to \mathbb{R}$ as

$$g_\varepsilon(r; p) := \sum_{x\in\mathcal{X}} \max\{r\varepsilon, p(x)\} .$$

**Lemma E.6.** *For any given $p \in \Delta_{\mathcal{X}}$, the map $r \mapsto g_\varepsilon(r;p)$ is $\varepsilon S$-Lipschitz and has a unique fixed point. Moreover, denoting this fixed point by $r^*$, it holds that $r^* \in [1, \max_x p(x)/\varepsilon)$, and that $g_\varepsilon(r;p) > r$ for $r < r^*$ and $g_\varepsilon(r;p) < r$ for $r > r^*$.*

*Proof.* We firstly note that $g_\varepsilon(\cdot; p)$ can be easily verified to be convex. For any $r \in \mathbb{R}$ and any subgradient $h$ of $g_\varepsilon(\cdot;p)$ at $r$, it holds that $|h| \leqslant \varepsilon S$. Hence, the convexity of $g_\varepsilon(\cdot;p)$ implies that $|g_\varepsilon(r;p) - g_\varepsilon(r';p)| \leqslant \varepsilon S|r - r'|$ for any $r, r' \in \mathbb{R}$, or that $g_\varepsilon(\cdot;p)$ is $\varepsilon S$-Lipschitz. This implies, since $\varepsilon S < 1$ by assumption, that $g_\varepsilon(\cdot;p)$ is a contraction mapping; hence, via Banach's fixed point theorem, it admits a unique fixed point $r^* \in \mathbb{R}$. For $r < 1$, it holds that $g_\varepsilon(r;p) \geqslant \sum_x p(x) = 1 > r$. While for $r \geqslant \max_x p(x)/\varepsilon$, $g_\varepsilon(r;p) = r\varepsilon S < r$. Therefore, $r^* \in [1, \max_x p(x)/\varepsilon)$. Moreover, for any $r < r^*$ ($r > r^*$), it must hold that $g_\varepsilon(r;p) > r$ ($g_\varepsilon(r;p) < r$); as otherwise, the intermediate value theorem, applied to $g_\varepsilon(r;p) - r$, would imply the existence of another fixed point, a contradiction. □

Next, we define $r_\varepsilon \colon \Delta_{\mathcal{X}} \to \mathbb{R}$ as the function that maps a distribution $p \in \Delta_{\mathcal{X}}$ to the fixed point of $g_\varepsilon(\cdot;p)$. This function is well-defined as implied by Lem. E.6. We now show that the solution of the information projection problem onto $\Delta^\varepsilon_{\mathcal{X}}$ can be expressed in terms of the function $r_\varepsilon$. For $p \in \Delta_{\mathcal{X}}$, we define $p_\varepsilon \in \Delta^\varepsilon_{\mathcal{X}}$ as

$$p_\varepsilon(x) := \frac{\max\{r_\varepsilon(p)\varepsilon, p(x)\}}{\sum_{x'\in\mathcal{X}} \max\{r_\varepsilon(p)\varepsilon, p(x')\}} = \max\{\varepsilon, p(x)/r_\varepsilon(p)\} .$$

**Lemma E.7.** *For $p \in \Delta_{\mathcal{X}}$, it holds that $p_\varepsilon = \arg\min_{q\in\Delta^\varepsilon_{\mathcal{X}}} D_{\mathrm{KL}}(q \,\|\, p)$.*

*Proof.* We assume without loss of generality that $p(x) > 0$ for all $x \in \mathcal{X}$; as otherwise, we can cast the problem into a lower dimensional one considering only the elements $x \in \mathcal{X}$ for which $p(x) > 0$. Since the constraint set is compact and the objective is continuous and strictly convex, this minimization problem admits a unique optimal solution. We start by rewriting the problem as

$$\begin{aligned} \min_{q\in\mathbb{R}^S} \quad & \sum_{x\in\mathcal{X}} q(x) \log\frac{q(x)}{p(x)} \\ \text{subject to} \quad & \varepsilon - q(x) \leqslant 0 \;\forall x \in \mathcal{X} \\ & \sum_{x\in\mathcal{X}} q(x) - 1 = 0 \end{aligned}$$

Define the Lagrangian

$$L(q,u,v) := \sum_{x\in\mathcal{X}} q(x)\log\frac{q(x)}{p(x)} + \sum_{x\in\mathcal{X}} u(x)(\varepsilon - q(x)) + v\left(\sum_{x\in\mathcal{X}} q(x) - 1\right)$$

for $v \in \mathbb{R}$ and $u \in \mathbb{R}^S_{\geqslant 0}$. We have that

$$\frac{\partial L}{\partial q(x)}(q,u,v) = \log\frac{q(x)}{p(x)} + 1 - u(x) + v .$$

We now show that we can satisfy the KKT conditions by choosing a solution pair $q^*$ and $u^*, v^*$ where

$$q^*(x) := p_\varepsilon(x) = \max\{\varepsilon, p(x)/r_\varepsilon(p)\}\,, \quad u^*(x) := \log \frac{\max\{\varepsilon, p(x)/r_\varepsilon(p)\}}{p(x)/r_\varepsilon(p)}\,, \text{ and } v^* := -1 + \log r_\varepsilon(p)\,.$$

Firstly, $q^*$ indeed belongs to $\Delta^\varepsilon_{\mathcal{X}}$ by the definition of $p_\varepsilon$, and $u^*$ is non-negative. Moreover, whenever $q^*(x) > \varepsilon$, we get that $u^*(x) = 0$; hence, complementary slackness holds. Finally,

$$\frac{\partial L}{\partial q(x)}(q^*, u^*, v^*) = \log \frac{\max\{\varepsilon, p(x)/r_\varepsilon(p)\}}{p(x)} + 1 - \log \frac{\max\{\varepsilon, p(x)/r_\varepsilon(p)\}}{p(x)/r_\varepsilon(p)} - 1 + \log r_\varepsilon(p) = 0\,.$$

Therefore, we conclude that $p_\varepsilon$ is the optimal solution. □

Computing $p_\varepsilon$, or the information projection of $p$ onto $\Delta^\varepsilon_{\mathcal{X}}$, can be performed efficiently. In particular, the following characterization implies that $r_\varepsilon(p)$ can be computed exactly in a finite number of steps by iterating over the set of states.

**Lemma E.8.** *Let $\mathcal{X}^+_p := \{x \in \mathcal{X} \colon g_\varepsilon(p(x)/\varepsilon; p) < p(x)/\varepsilon\}$ and $\mathcal{X}^-_p := \mathcal{X} \backslash \mathcal{X}^+_p$. Then,*

$$r_\varepsilon(p) = \frac{\sum_{x \in \mathcal{X}^+} p(x)}{1 - \varepsilon|\mathcal{X}^-_p|}\,.$$

*Proof.* As stated in the proof of Lem. E.6, for $r \geqslant \max_{x \in \mathcal{X}} p(x)/\varepsilon$, $g_\varepsilon(r; p) = r\varepsilon S < r$; hence $\mathcal{X}^+_p$ is non-empty as it at least includes $\arg\max_{x \in \mathcal{X}} p(x)$. Moreover, from the same lemma, we have that $r_\varepsilon(p) < \min_{x \in \mathcal{X}^+_p} p(x)/\varepsilon$ and $r_\varepsilon(p) \geqslant \max_{x \in \mathcal{X}^-_p} p(x)/\varepsilon$ (if $\mathcal{X}^-_p$ is non-empty). Therefore,

$$r_\varepsilon(p) = g_\varepsilon(r_\varepsilon(p); p) = \sum_{x \in \mathcal{X}} \max\{r_\varepsilon(p)\varepsilon, p(x)\} = r_\varepsilon(p)\varepsilon|\mathcal{X}^-_p| + \sum_{x \in \mathcal{X}^+} p(x)\,.$$

□

The previous lemma also implies that $r_\varepsilon(p) \leqslant (1 - \varepsilon S)^{-1}$. Returning back to our original objective, we show next that $\|p_\varepsilon - q_\varepsilon\|_1$ is no larger than a constant multiple of $\|p - q\|_1$ for any two distributions $p$ and $q$. Towards that end, we first show that $r_\varepsilon$ is Lipschitz continuous.

**Lemma E.9.** *For $\varepsilon \leqslant \frac{1}{2S}$, the function $r_\varepsilon$ is 1-Lipschitz with respect to the $\|\cdot\|_1$ norm; that is,*

$$|r_\varepsilon(p) - r_\varepsilon(q)| \leqslant \|p - q\|_1$$

*for any $p, q \in \Delta_{\mathcal{X}}$.*

*Proof.* Note that, for any fixed $r \in \mathbb{R}$, $g_\varepsilon(r; \cdot)$ is convex; and that for any $p \in \Delta_{\mathcal{X}}$ and subgradient $k$ of $g_\varepsilon(r; \cdot)$ at $p$, it holds that $k$ is non-negative and satisfies $\|k\|_\infty \leqslant 1$. Hence, for any $p, q \in \Delta_{\mathcal{X}}$,

$$|g_\varepsilon(r; p) - g_\varepsilon(r; q)| \leqslant \sum_{x \colon p(x) > q(x)} (p(x) - q(x)) = \sum_{x \colon q(x) > p(x)} (q(x) - p(x)) = \frac{1}{2}\|p - q\|_1\,. \tag{54}$$

Then, we obtain that

$$\begin{aligned} |r_\varepsilon(p) - r_\varepsilon(q)| &= |g_\varepsilon(r_\varepsilon(p); p) - g_\varepsilon(r_\varepsilon(q); q)| \\ &\leqslant |g_\varepsilon(r_\varepsilon(p); p) - g_\varepsilon(r_\varepsilon(q); p)| + |g_\varepsilon(r_\varepsilon(q); p) - g_\varepsilon(r_\varepsilon(q); q)| \\ &\leqslant \varepsilon S|r_\varepsilon(p) - r_\varepsilon(q)| + \frac{1}{2}\|p - q\|_1 \leqslant \frac{1}{2}|r_\varepsilon(p) - r_\varepsilon(q)| + \frac{1}{2}\|p - q\|_1\,, \end{aligned}$$

where the second inequality follows from (54) and Lem. E.6, and the last inequality holds since $\varepsilon \leqslant \frac{1}{2S}$. The lemma then follows after rearranging the last result. □

**Lemma E.10.** *Assuming $\varepsilon \leqslant \frac{1}{2S}$, it holds for any $p, q \in \Delta_{\mathcal{X}}$ that $\|p_\varepsilon - q_\varepsilon\|_1 \leqslant \frac{5}{2}\|p - q\|_1$.*

*Proof.* We have that

$$\begin{aligned} q_\varepsilon(x) &= \frac{\max\{r_\varepsilon(q)\varepsilon, q(x)\}}{r_\varepsilon(q)} \\ &= \frac{\max\{r_\varepsilon(q)\varepsilon, q(x)\} - \max\{r_\varepsilon(p)\varepsilon, p(x)\} + \max\{r_\varepsilon(p)\varepsilon, p(x)\}}{r_\varepsilon(q)} \\ &= \frac{\max\{r_\varepsilon(q)\varepsilon, q(x)\} - \max\{r_\varepsilon(p)\varepsilon, p(x)\}}{r_\varepsilon(q)} + \frac{r_\varepsilon(p)}{r_\varepsilon(q)} p_\varepsilon(x)\,. \end{aligned}$$

Then,

$$q_\varepsilon(x) - p_\varepsilon(x) = \frac{1}{r_\varepsilon(q)}\big(\max\{r_\varepsilon(q)\varepsilon, q(x)\} - \max\{r_\varepsilon(p)\varepsilon, p(x)\} + \big(r_\varepsilon(p) - r_\varepsilon(q)\big)p_\varepsilon(x)\big)\,.$$

Using Lem. E.9 and the fact that

$$\begin{aligned} |\max\{r_\varepsilon(q)\varepsilon, q(x)\} - \max\{r_\varepsilon(p)\varepsilon, p(x)\}| &\leqslant \max\big\{\varepsilon|r_\varepsilon(q) - r_\varepsilon(p)|, |q(x) - p(x)|\big\} \\ &\leqslant \varepsilon|r_\varepsilon(q) - r_\varepsilon(p)| + |q(x) - p(x)|\,, \end{aligned}$$

we obtain that

$$\begin{aligned} \|p_\varepsilon - q_\varepsilon\|_1 &= \sum_x |q_\varepsilon(x) - p_\varepsilon(x)| \\ &\leqslant \frac{1}{r_\varepsilon(q)} \sum_x \big(\varepsilon|r_\varepsilon(q) - r_\varepsilon(p)| + |q(x) - p(x)| + p_\varepsilon(x)|r_\varepsilon(p) - r_\varepsilon(q)|\big) \\ &\leqslant \frac{\|p-q\|_1}{r_\varepsilon(q)}\big(2 + \varepsilon S\big) \leqslant \frac{5}{2}\|p-q\|_1\,, \end{aligned}$$

where the last step uses that $\varepsilon S \leqslant 1/2$ and $r_\varepsilon(q) \geqslant 1$. □

Finally, we arrive at the sought result, a parallel of Lem. A.3.

**Lemma E.11.** *For all $n \in [N-1]$, $(x, a, x') \in \mathcal{X} \times \mathcal{A} \times \mathcal{X}$, and $t \in [T]$; $\widehat{p}^t_{n+1}(x'|x, a)$ as defined in (52) with $\varepsilon \leqslant \frac{1}{2S}$ satisfies*

$$\|\widehat{p}^{t+1}_{n+1}(\cdot|x, a) - \widehat{p}^t_{n+1}(\cdot|x, a)\|_1 \leqslant \frac{5\mathbb{1}_{\{x^t_n = x, a^t_n = a\}}}{N^{t+1}_n(x, a) + S}\,.$$

*Proof.* This is a direct consequence of the definition in (52) and Lems. E.5, E.7 and E.10. □

### E.2.3. THE ALGORITHM

For $\delta \in (0, 1)$, define

$$C'_\delta := \sqrt{322S + 12\sqrt{S}\log^{5/2}\frac{S^2ANT^2}{4\delta} + 620}\,, \tag{55}$$

which is the leading factor in the confidence bound of Lem. E.4. For the purpose of exploration, much like the full information case, we will utilize at each round $t$ a bonus reward vector $b^t \in \mathbb{R}^{NSA}$ to be subtracted from the estimated gradient, where

$$b^t_n(x, a) := L(N-n)\frac{C'_{1/T}}{\sqrt{\max\{1, N^t_n(x, a)\}}} \tag{56}$$

for $(t, n, x, a) \in [T] \times (\{0\} \bigcup [N]) \times \mathcal{X} \times \mathcal{A}$.

Finally, with all its components detailed, we present Alg. 3, our first approach for CURL with bandit feedback. As mentioned in Sec. 4.2.1, the main changes compared to Alg. 1 are the use of spherical estimation to obtain a surrogate for the gradient and the use of a suitably altered transition kernel estimator.

**Algorithm 3** Bonus O-MD-CURL (bandit feedback)

**input:** learning rate $\tau > 0$, perturbation rate $\delta \in (0, 1]$, sequence of exploration parameters $(\alpha_t)_{t\in[T]} \in (0,1)^T$
**initialization:** $\widehat{p}^1_n(x'|x,a) \leftarrow 1/S \;\forall(n,x,x',a)$, $\mu^1 \in \arg\min_{\mu\in\mathcal{M}^{\widehat{p}^1}_{\mu_0}} \psi(\mu)$
**for** $t = 1, \ldots, T$ **do**
  draw $\boldsymbol{u}^t \in \mathbb{S}^{NS(A-1)}$ uniformly at random
  $\zeta^t \leftarrow \zeta^{\boldsymbol{u}^t,\widehat{p}^t} = \Lambda_{\widehat{p}^t}(\kappa\mathbb{1}_{NS(A-1)} + \kappa\boldsymbol{u}^t)$
  $\widehat{\mu}^t \leftarrow (1-\delta)\mu^t + \delta\zeta^t$
  $\pi^t_n(a|x) \leftarrow \widehat{\mu}^t(x,a)/\sum_{a\in\mathcal{A}}\widehat{\mu}^t(x,a)$
  execute $\pi^t$ and observe $F^t(\mu^{\pi^t,p})$ and a sampled trajectory $o^t := (x^t_1, a^t_1, \ldots, x^t_N, a^t_N)$
  $g^t \leftarrow \frac{1-\delta}{\delta\kappa} NS(A-1)F^t(\mu^{\pi^t,p})\boldsymbol{u}^t$
  construct $\mathring{g}^t \in \mathbb{R}^{NSA}$ as $\mathring{g}^t_n(x,a) \leftarrow g^t_n(x,a)$ for $a \neq a^*$ and $\mathring{g}^t_n(x,a^*) \leftarrow 0$
  construct bonus vector $b^t$ as in (56)
  $\tilde{\pi}^t_n(a|x) \leftarrow (1-\alpha_t)\mu^t(x,a)/\sum_{a\in\mathcal{A}}\mu^t(x,a) + \alpha_t/A$
  construct the new estimated kernel $\widehat{p}^{t+1}$ via (51) and (52)
  set $\mu^{t+1} \in \arg\min_{\mu\in\mathcal{M}^{\widehat{p}^{t+1}}_{\mu_0}} \tau\langle\mathring{g}^t - b^t, \mu\rangle + \Gamma(\mu, \mu^{\tilde{\pi}^t,\widehat{p}^t})$
**end for**

#### E.2.4. Auxiliary lemmas

**Lemma E.12.** *For $0 < \delta < 1$ and $\widehat{p}^t$ as defined in (52), it holds that*

$$\sum_{t=1}^T\sum_{n=1}^N\sum_{i=0}^{n-1}\sum_{x,a}\mu_i^{\pi^t,p}(x,a)\|p_{i+1}(\cdot|x,a) - \widehat{p}^t_{i+1}(\cdot|x,a)\|_1 \leqslant 3C'_\delta N^2\sqrt{SAT} + 2SN^2\sqrt{2T\log\Big(\frac{N}{\delta}\Big)}$$

*with probability at least $1-2\delta$.*

*Proof.* This lemma can be proved in the same manner as its full information version Lem. A.1 with only two small changes; we use the bound of Lem. E.4 instead of Lem. B.1 and we modify the definition of the filtration to be $\mathcal{F}_t := \sigma(\boldsymbol{u}^1, o^1, \ldots, \boldsymbol{u}^{t-1}, o^{t-1}, \boldsymbol{u}^t)$. □

**Lemma E.13.** *For any $0 < \delta < 1$,*

$$\sum_{t=1}^T\sum_{n=0}^N(N-n)\sum_{x,a}\frac{\mu_n^{\pi^t,p}(x,a)}{\sqrt{\max\{1, N^t_n(x,a)\}}} \leqslant 3N^2\sqrt{SAT} + SN^2\sqrt{2T\log\left(\frac{N}{\delta}\right)},$$

*holds with probability at least $1-\delta$.*

*Proof.* The proof is the same as for Lem. A.2 (the version proved in the full information case), except that, again, the filtration used in the proof would be defined as $\mathcal{F}_t := \sigma(\boldsymbol{u}^1, o^1, \ldots, \boldsymbol{u}^{t-1}, o^{t-1}, \boldsymbol{u}^t)$. □

**Proposition E.14.** *Let $b^t$ and $\widehat{p}^t$ be as defined in (56) and (52) respectively. Then, for any $\delta \in (0,1)$, with probability at least $1-3\delta$,*

$$\sum_{t=1}^T\big\langle\mu^{\pi^t,\widehat{p}^t}, b^t\big\rangle + \sum_{t=1}^T\big\langle\mu_0, b^t_0\big\rangle \leqslant LC'_{1/T}N^3\left(3C'_\delta\sqrt{SAT} + 2S\sqrt{2T\log\Big(\frac{N}{\delta}\Big)}\right) + LC'_{1/T}N^2\left(3\sqrt{SAT} + S\sqrt{2T\log\left(\frac{N}{\delta}\right)}\right).$$

*Proof.* The proof is the same as that of Prop. 3.1 except that we would rely on Lems. E.12 and E.13 in place of Lems. A.1 and A.2, and use the definition of $b^t$ in (56) instead of (11). □

**Lemma E.15.** *Let $X$ be a random variable taking values in $\mathbb{R}$, $z_1, z_2 \geqslant 0$ be two constants, and $\delta' \in (0,1)$. If $X \leqslant z_2$ uniformly and $P(X > z_1) \leqslant \delta'$, then $\mathbb{E}[X] \leqslant z_1 + \delta' z_2$ .*

*Proof.* Simply, $\mathbb{E}[X] = \mathbb{E}[\mathbb{I}\{X \leqslant z_1\}X] + \mathbb{E}[\mathbb{I}\{X > z_1\}X] \leqslant z_1 + z_2 P(x > z_1) \leqslant z_1 + \delta' z_2$. □

### E.2.5. Regret analysis

The following theorem, a restatement of Thm. 4.3, provides a regret bound for Alg. 3. Recall that we have adopted in this section the shorthand notation $S = |\mathcal{X}|$ and $A = |\mathcal{A}|$.

**Theorem E.16.** *Under Asm. 4.2, Alg. 3 with a suitable tuning of $\tau$, $\delta$, and $(\alpha_t)_{t\in[T]}$ satisfies for any policy $\pi \in (\Delta_{\mathcal{A}})^{\mathcal{X}\times N}$ that*

$$\mathbb{E}\left[R_T(\pi)\right] \lesssim \sqrt{\frac{L(L+1)}{\varepsilon}} S^{5/4} A^{5/4} N^3 T^{3/4} + \frac{L+1}{\varepsilon} S^2 A^{5/2} N^4 \sqrt{T}\,,$$

*where $\lesssim$ signifies that the inequality holds up to factors logarithmic in $T$, $N$, $S$, and $A$.*

*Proof.* Fixing $\pi \in (\Delta_{\mathcal{A}})^{\mathcal{X}\times N}$, we have that

$$\begin{aligned}
\mathbb{E}\left[R_T(\pi)\right] &= \mathbb{E}\sum_{t=1}^{T}\big(F^t(\mu^{\pi^t,p}) - F^t(\mu^{\pi,p})\big) \\
&= \underbrace{\mathbb{E}\sum_{t=1}^{T}\big(F^t(\mu^{\pi^t,p}) - F^t(\mu^{\pi^t,\widehat{p}^t})\big)}_{①} + \underbrace{\mathbb{E}\sum_{t=1}^{T}\big(F^t(\mu^{\pi^t,\widehat{p}^t}) - F^t(\mu^{\pi,\widehat{p}^t})\big)}_{②} + \underbrace{\mathbb{E}\sum_{t=1}^{T}\big(F^t(\mu^{\pi,\widehat{p}^t}) - F^t(\mu^{\pi,p})\big)}_{③}\,.
\end{aligned}$$

It holds with probability at least $1 - \frac{2}{T}$ that

$$\begin{aligned}
① \leqslant L\sum_{t=1}^{T}\big\|\mu^{\pi^t,p} - \mu^{\pi^t,\widehat{p}^t}\big\|_1 &= L\sum_{t=1}^{T}\sum_{n=1}^{N}\big\|\mu_n^{\pi^t,p} - \mu_n^{\pi^t,\widehat{p}^t}\big\|_1 \\
&\leqslant L\sum_{t=1}^{T}\sum_{n=1}^{N}\sum_{i=0}^{n-1}\sum_{x,a}\mu_i^{\pi^t,p}(x,a)\big\|p_{i+1}(\cdot|x,a) - \widehat{p}_{i+1}^t(\cdot|x,a)\big\|_1 \\
&\leqslant 3LN^2\sqrt{SAT}C'_{1/T} + 2LSN^2\sqrt{2T\log(NT)}\,,
\end{aligned}$$

where the first inequality uses the Lipschitz continuity of $F^t$, the second inequality follows from Lem. B.1, and the last inequality follows from Lem. E.12. Hence, since $L\sum_{t=1}^{T}\big\|\mu^{\pi^t,p} - \mu^{\pi^t,\widehat{p}^t}\big\|_1 \leqslant 2NLT$, it holds via Lem. E.15 (with $\delta' = \frac{2}{T}$) that

$$\mathbb{E}\left[①\right] \leqslant 3LN^2\sqrt{SAT}C'_{1/T} + 2LSN^2\sqrt{2T\log(NT)} + 4LN\,. \tag{57}$$

For the third sum, we use again the Lipschitz continuity of $F^t$, Lem. B.1, and Lem. E.4 to get that with probability at least $1 - \frac{1}{T}$,

$$\begin{aligned}
③ \leqslant L\sum_{t=1}^{T}\big\|\mu^{\pi,\widehat{p}^t} - \mu^{\pi,p}\big\|_1 &\leqslant L\sum_{t=1}^{T}\sum_{n=1}^{N}\big\|\mu_n^{\pi,\widehat{p}^t} - \mu_n^{\pi,p}\big\|_1 \\
&\leqslant L\sum_{t=1}^{T}\sum_{n=1}^{N}\sum_{i=0}^{n-1}\sum_{x,a}\mu_i^{\pi,\widehat{p}^t}(x,a)\big\|p_{i+1}(\cdot|x,a) - \widehat{p}_{i+1}^t(\cdot|x,a)\big\|_1 \\
&\leqslant L\sum_{t=1}^{T}\sum_{n=1}^{N}\sum_{i=0}^{n-1}\sum_{x,a}\mu_i^{\pi,\widehat{p}^t}(x,a)\frac{C'_{1/T}}{\sqrt{\max\{1, N_i^t(x,a)\}}} \\
&= L\sum_{t=1}^{T}\sum_{n=0}^{N}(N-n)\sum_{x,a}\mu_n^{\pi,\widehat{p}^t}(x,a)\frac{C'_{1/T}}{\sqrt{\max\{1, N_n^t(x,a)\}}}
\end{aligned}$$

$$
\begin{aligned}
&= \sum_{t=1}^{T} \big\langle \mu^{\pi,\widehat{p}^t}, b^t \big\rangle + \sum_{t=1}^{T} \big\langle \mu_0, b_0^t \big\rangle \\
&= \sum_{t=1}^{T} \big\langle \mu^{\pi^t,\widehat{p}^t}, b^t \big\rangle + \sum_{t=1}^{T} \big\langle \mu_0, b_0^t \big\rangle + \sum_{t=1}^{T} \big\langle \mu^{\pi,\widehat{p}^t} - \mu^{\pi^t,\widehat{p}^t}, b^t \big\rangle .
\end{aligned}
$$

Via Prop. E.14, it holds with probability $1 - \frac{3}{T}$ that

$$
\sum_{t=1}^{T} \big\langle \mu^{\pi^t,\widehat{p}^t}, b^t \big\rangle + \sum_{t=1}^{T} \big\langle \mu_0, b_0^t \big\rangle \leqslant L C'_{1/T} N^3 \big[ 3 C'_{1/T} \sqrt{SAT} + 2S\sqrt{2T\log(NT)} \big] + L C'_{1/T} N^2 \big[ 3\sqrt{SAT} + S\sqrt{2T\log(NT)} \big] .
$$

Hence, chaining these last two results and using a union bound, we get via Lem. E.15 (with $\delta' = \frac{4}{T}$) that

$$
\mathbb{E}\left[ ③ - \sum_{t=1}^{T} \big\langle \mu^{\pi,\widehat{p}^t} - \mu^{\pi^t,\widehat{p}^t}, b^t \big\rangle \right] \leqslant L C'_{1/T} N^3 \big[ 3 C'_{1/T} \sqrt{SAT} + 2S\sqrt{2T\log(NT)} \big] + L C'_{1/T} N^2 \big[ 3\sqrt{SAT} + S\sqrt{2T\log(NT)} \big] + 8LN(1 + C'_{1/T} N) , \tag{58}
$$

where we have used that

$$
③ - \sum_{t=1}^{T} \big\langle \mu^{\pi,\widehat{p}^t} - \mu^{\pi^t,\widehat{p}^t}, b^t \big\rangle \leqslant L \sum_{t=1}^{T} \big\| \mu^{\pi,\widehat{p}^t} - \mu^{\pi,p} \big\|_1 + \sum_{t=1}^{T} \| b^t \|_\infty \| \mu^{\pi,\widehat{p}^t} - \mu^{\pi^t,\widehat{p}^t} \|_1 \leqslant 2LNT(1 + C'_{1/T} N) .
$$

Define $\widehat{F}^t \colon (\Delta_{\mathcal{X}\times\mathcal{A}})^N \to \mathbb{R}$ as

$$
\widehat{F}^t(\mu) = \mathbb{E}_{\boldsymbol{v} \in \mathbb{B}^{NS(A-1)}} \Big[ F^t\big( (1-\delta)\mu + \delta \zeta^{\boldsymbol{v},\widehat{p}^t} \big) \Big] .
$$

As $\widehat{p}^t$ satisfies the condition of Asm. 4.2 by design, $\zeta^{\boldsymbol{v},\widehat{p}^t} \in \mathcal{M}_{\mu_0}^{\widehat{p}^t} \subset (\Delta_{\mathcal{X}\times\mathcal{A}})^N$ as argued in App. E.2.1; thus, $\widehat{F}^t$ is well-defined. Similarly, since $\boldsymbol{u^t} \in \mathbb{S}^{NS(A-1)} \subset \mathbb{B}^{NS(A-1)}$ and $\zeta^t = \zeta^{\boldsymbol{u}^t,\widehat{p}^t}$, it holds that $\zeta^t \in \mathcal{M}_{\mu_0}^{\widehat{p}^t}$. Via the convexity of $\mathcal{M}_{\mu_0}^{\widehat{p}^t}$, the fact that $\mu^t \in \mathcal{M}_{\mu_0}^{\widehat{p}^t}$, and the definition of $\widehat{\mu}^t$; it holds that $\widehat{\mu}^t \in \mathcal{M}_{\mu_0}^{\widehat{p}^t}$. This yields that $\widehat{\mu}^t = \mu^{\pi^t,\widehat{p}^t}$, recalling the definition of $\pi^t$ in Alg. 3. Using the Lipschitz smoothness of $F^t$, we have that

$$
\begin{aligned}
F^t(\mu^{\pi^t,\widehat{p}^t}) - \widehat{F}^t(\mu^t) = F^t(\widehat{\mu}^t) - \widehat{F}^t(\mu^t) &= F^t((1-\delta)\mu^t + \delta\zeta^t) - \mathbb{E}_{\boldsymbol{v} \in \mathbb{B}^{NS(A-1)}} \Big[ F^t\big( (1-\delta)\mu^t + \delta \zeta^{\boldsymbol{v},\widehat{p}^t} \big) \Big] \\
&\leqslant \delta L \mathbb{E}_{\boldsymbol{v} \in \mathbb{B}^{NS(A-1)}} \| \zeta^t - \zeta^{\boldsymbol{v},\widehat{p}^t} \|_1 \leqslant 2\delta LN
\end{aligned}
$$

and that

$$
\begin{aligned}
\widehat{F}^t(\mu^{\pi,\widehat{p}^t}) - F^t(\mu^{\pi,\widehat{p}^t}) &= \mathbb{E}_{\boldsymbol{v} \in \mathbb{B}^{NS(A-1)}} \Big[ F^t((1-\delta)\mu^{\pi,\widehat{p}^t} + \delta\zeta^{\boldsymbol{v},\widehat{p}^t}) \Big] - F^t(\mu^{\pi,\widehat{p}^t}) \\
&\leqslant \delta L \mathbb{E}_{\boldsymbol{v} \in \mathbb{B}^{NS(A-1)}} \| \zeta^{\boldsymbol{v},\widehat{p}^t} - \mu^{\pi,\widehat{p}^t} \|_1 \leqslant 2\delta LN .
\end{aligned}
$$

Hence,

$$
\begin{aligned}
② &= \sum_{t=1}^{T} \big( F^t(\mu^{\pi^t,\widehat{p}^t}) - \widehat{F}^t(\mu^t) + \widehat{F}^t(\mu^t) - \widehat{F}^t(\mu^{\pi,\widehat{p}^t}) + \widehat{F}^t(\mu^{\pi,\widehat{p}^t}) - F^t(\mu^{\pi,\widehat{p}^t}) \big) \\
&\leqslant \sum_{t=1}^{T} \big\langle \nabla \widehat{F}^t(\mu^t), \mu^t - \mu^{\pi,\widehat{p}^t} \big\rangle + 4\delta LNT \\
&= \sum_{t=1}^{T} \big\langle \nabla \widehat{F}^t(\mu^t) - b^t, \mu^t - \mu^{\pi,\widehat{p}^t} \big\rangle + 4\delta LNT + \sum_{t=1}^{T} \big\langle b^t, \mu^{\pi^t,\widehat{p}^t} - \mu^{\pi,\widehat{p}^t} \big\rangle + \sum_{t=1}^{T} \big\langle b^t, \mu^t - \mu^{\pi^t,\widehat{p}^t} \big\rangle .
\end{aligned}
$$

The last term is easily bounded as follows:

$$\sum_{t=1}^{T}\big\langle b^t, \mu^t - \mu^{\pi^t,\widehat{p}^t}\big\rangle = \sum_{t=1}^{T}\big\langle b^t, \mu^t - \widehat{\mu}^t\big\rangle = \delta\sum_{t=1}^{T}\big\langle b^t, \mu^t - \zeta^t\big\rangle \leqslant \delta\sum_{t=1}^{T}\|b^t\|_\infty\|\mu^t - \zeta^t\|_1 \leqslant 2\delta C'_{1/T}LN^2T\,.$$

We then conclude that

$$\mathbb{E}\left[\text{②} + \sum_{t=1}^{T}\big\langle b^t, \mu^{\pi,\widehat{p}^t} - \mu^{\pi^t,\widehat{p}^t}\big\rangle\right] \leqslant \mathbb{E}\sum_{t=1}^{T}\big\langle \nabla\widehat{F}^t(\mu^t) - b^t, \mu^t - \mu^{\pi,\widehat{p}^t}\big\rangle + 4\delta LNT + 2\delta C'_{1/T}LN^2T\,. \tag{59}$$

Then, combining (57), (58), and (59) yields that

$$\begin{aligned}\mathbb{E}\left[R_T(\pi)\right] \leqslant \mathbb{E}\sum_{t=1}^{T}\big\langle \nabla\widehat{F}^t(\mu^t) - b^t, \mu^t - \mu^{\pi,\widehat{p}^t}\big\rangle + 2\delta L(2 + C'_{1/T}N)NT \\ + 3LC'_{1/T}N^3\big[3C'_{1/T}\sqrt{SAT} + 2S\sqrt{2T\log(NT)}\big] + 4LN(3 + 2C'_{1/T}N)\,.\end{aligned} \tag{60}$$

Define $\mathbb{F}^t, \widehat{\mathbb{F}}^t \colon \big(\mathcal{M}_{\mu_0}^{\widehat{p}^t}\big)^- \to \mathbb{R}$ as $\mathbb{F}^t(\xi) := F^t\big(\Lambda_{\widehat{p}^t}(\xi)\big)$ and $\widehat{\mathbb{F}}^t(\xi) := \widehat{F}^t\big(\Lambda_{\widehat{p}^t}(\xi)\big)$. Then, recalling that $\kappa := \frac{\varepsilon}{A-1+\sqrt{A-1}}$,

$$\begin{aligned}\widehat{\mathbb{F}}^t\big(\Lambda_{\widehat{p}^t}^{-1}(\mu^t)\big) = \widehat{F}^t(\mu^t) &= \mathbb{E}_{\boldsymbol{v}\in\mathbb{B}^{NS(A-1)}}\left[F^t\big((1-\delta)\mu^t + \delta\zeta^{\boldsymbol{v},\widehat{p}^t}\big)\right] \\ &= \mathbb{E}_{\boldsymbol{v}\in\mathbb{B}^{NS(A-1)}}\left[\mathbb{F}^t\big(\Lambda_{\widehat{p}^t}^{-1}\big((1-\delta)\mu^t + \delta\zeta^{\boldsymbol{v},\widehat{p}^t}\big)\big)\right] \\ &= \mathbb{E}_{\boldsymbol{v}\in\mathbb{B}^{NS(A-1)}}\left[\mathbb{F}^t\big(\big(B^{\widehat{p}^t}\big)^+\big((1-\delta)\mu^t + \delta\zeta^{\boldsymbol{v},\widehat{p}^t} - \boldsymbol{\beta}^{\widehat{p}^t}\big)\big)\right] \\ &= \mathbb{E}_{\boldsymbol{v}\in\mathbb{B}^{NS(A-1)}}\left[\mathbb{F}^t\big((1-\delta)\big(B^{\widehat{p}^t}\big)^+\big(\mu^t - \boldsymbol{\beta}^{\widehat{p}^t}\big) + \delta\big(B^{\widehat{p}^t}\big)^+\big(\zeta^{\boldsymbol{v},\widehat{p}^t} - \boldsymbol{\beta}^{\widehat{p}^t}\big)\big)\right] \\ &= \mathbb{E}_{\boldsymbol{v}\in\mathbb{B}^{NS(A-1)}}\left[\mathbb{F}^t\big((1-\delta)\Lambda_{\widehat{p}^t}^{-1}(\mu^t) + \delta\Lambda_{\widehat{p}^t}^{-1}(\zeta^{\boldsymbol{v},\widehat{p}^t})\big)\right] \\ &= \mathbb{E}_{\boldsymbol{v}\in\mathbb{B}^{NS(A-1)}}\left[\mathbb{F}^t((1-\delta)\Lambda_{\widehat{p}^t}^{-1}(\mu^t) + \delta\kappa\mathbf{1}_{NS(A-1)} + \delta\kappa\boldsymbol{v})\right]\,,\end{aligned}$$

where the fourth equality follows form the fact that $\Lambda_{\widehat{p}^t}^{-1}(\mu) = \big(B^{\widehat{p}^t}\big)^+\big(\mu - \boldsymbol{\beta}^{\widehat{p}^t}\big)$, and the last equality follows since $\zeta^{\boldsymbol{v},\widehat{p}^t} = \Lambda_{\widehat{p}^t}(\kappa\mathbb{1}_{NS(A-1)} + \kappa\boldsymbol{v})$. Lem. 1 in (Flaxman et al., 2005) and the chain rule imply that

$$\begin{aligned}\nabla\widehat{\mathbb{F}}^t\big(\Lambda_{\widehat{p}^t}^{-1}(\mu^t)\big) &= \frac{1-\delta}{\delta\kappa}NS(A-1)\mathbb{E}_{\boldsymbol{u}\in\mathbb{S}^{NS(A-1)}}\left[\mathbb{F}^t((1-\delta)\Lambda_{\widehat{p}^t}^{-1}(\mu^t) + \delta\kappa\mathbf{1}_{NS(A-1)} + \delta\kappa\boldsymbol{u})\boldsymbol{u}\right] \\ &= \frac{1-\delta}{\delta\kappa}NS(A-1)\mathbb{E}_{\boldsymbol{u}\in\mathbb{S}^{NS(A-1)}}\left[\mathbb{F}^t\big((1-\delta)\Lambda_{\widehat{p}^t}^{-1}(\mu^t) + \delta\Lambda_{\widehat{p}^t}^{-1}\big(\zeta^{\boldsymbol{u},\widehat{p}^t}\big)\big)\boldsymbol{u}\right] \\ &= \frac{1-\delta}{\delta\kappa}NS(A-1)\mathbb{E}_{\boldsymbol{u}\in\mathbb{S}^{NS(A-1)}}\left[F^t((1-\delta)\mu^t + \delta\zeta^{\boldsymbol{u},\widehat{p}^t})\boldsymbol{u}\right] \\ &= \frac{1-\delta}{\delta\kappa}NS(A-1)\mathbb{E}_{\boldsymbol{u}^t\in\mathbb{S}^{NS(A-1)}}\left[F^t((1-\delta)\mu^t + \delta\zeta^t)\boldsymbol{u}^t\right]\,,\end{aligned}$$

where the last equality uses that $\zeta^t = \zeta^{\boldsymbol{u}^t,\widehat{p}^t}$ and that both $\mu^t$ and $\widehat{p}^t$ are independent with respect to $\boldsymbol{u}^t$. And since $\nabla\widehat{\mathbb{F}}^t\big(\Lambda_{\widehat{p}^t}^{-1}(\mu^t)\big) = \big(B^{\widehat{p}^t}\big)^\intercal\nabla\widehat{F}^t(\mu^t)$, we obtain that

$$\begin{aligned}\big(B^{\widehat{p}^t}\big(B^{\widehat{p}^t}\big)^+\big)^\intercal\nabla\widehat{F}^t(\mu^t) &= \frac{1-\delta}{\delta\kappa}NS(A-1)\mathbb{E}_{\boldsymbol{u}^t\in\mathbb{S}^{NS(A-1)}}\left[F^t((1-\delta)\mu^t + \delta\zeta^t)\big(\big(B^{\widehat{p}^t}\big)^+\big)^\intercal\boldsymbol{u}^t\right] \\ &= \mathbb{E}_{\boldsymbol{u}^t\in\mathbb{S}^{NS(A-1)}}\big[\big(\big(B^{\widehat{p}^t}\big)^+\big)^\intercal\widehat{g}^t\big]\,,\end{aligned} \tag{61}$$

where

$$\widehat{g}^t := \frac{1-\delta}{\delta\kappa}NS(A-1)F^t((1-\delta)\mu^t + \delta\zeta^t)\boldsymbol{u}^t = \frac{1-\delta}{\delta\kappa}NS(A-1)F^t(\widehat{\mu}^t)\boldsymbol{u}^t\,.$$

The vector $\widehat{g}^t$ differs from $g^t$ (which is employed in Alg. 3) in that it is defined using $F^t(\widehat{\mu}^t)$ instead of $F^t(\mu^{\pi^t,p})$. For round $t \in [T]$, let $\mathcal{F}_t := \sigma(\boldsymbol{u}^1, o^1, \ldots, \boldsymbol{u}^t, o^t)$ denote the $\sigma$-algebra generated by the random events up to the end of round $t$; and let $\mathbb{E}_t[\cdot] := \mathbb{E}[\cdot \mid \mathcal{F}_{t-1}]$ with $\mathcal{F}_0$ being the trivial $\sigma$-algebra. We then have that

$$\begin{aligned}
\mathbb{E}\sum_{t=1}^T \big\langle \nabla \widehat{F}^t(\mu^t), \mu^t - \mu^{\pi,\widehat{p}^t} \big\rangle &= \mathbb{E}\sum_{t=1}^T \Big\langle \big(B^{\widehat{p}^t}\big(B^{\widehat{p}^t}\big)^+\big)^\intercal \nabla \widehat{F}^t(\mu^t), \mu^t - \mu^{\pi,\widehat{p}^t} \Big\rangle \\
&= \mathbb{E}\sum_{t=1}^T \Big\langle \mathbb{E}_t\big[\big(\big(B^{\widehat{p}^t}\big)^+\big)^\intercal \widehat{g}^t\big], \mu^t - \mu^{\pi,\widehat{p}^t} \Big\rangle \\
&= \mathbb{E}\sum_{t=1}^T \Big\langle \big(\big(B^{\widehat{p}^t}\big)^+\big)^\intercal \widehat{g}^t, \mu^t - \mu^{\pi,\widehat{p}^t} \Big\rangle,
\end{aligned}$$

where the first equality holds via the fact that $B^{\widehat{p}^t}\big(B^{\widehat{p}^t}\big)^+(\mu^t - \mu^{\pi,\widehat{p}^t}) = \mu^t - \mu^{\pi,\widehat{p}^t}$ since $\mu^t - \mu^{\pi,\widehat{p}^t}$ belongs to the column space of $B^{\widehat{p}^t}$ (see App. E.1), the second equality uses (61) and the fact that conditioned on $\mathcal{F}_{t-1}$, the only source of randomness in $\big(\big(B^{\widehat{p}^t}\big)^+\big)^\intercal \widehat{g}^t$ is $\boldsymbol{u}^t$, which is sampled independently in each round; and the last equality uses the tower rule, linearity of expectation, and the fact that $\mu^t - \mu^{\pi,\widehat{p}^t}$ is measurable with respect to $\mathcal{F}_{t-1}$. Since $\mu^t - \mu^{\pi,\widehat{p}^t} = B^{\widehat{p}^t}\big(\Lambda^{-1}_{\widehat{p}^t}(\mu^t) - \Lambda^{-1}_{\widehat{p}^t}\big(\mu^{\pi,\widehat{p}^t}\big)\big)$, we have that

$$(\widehat{g}^t)^T\big(B^{\widehat{p}^t}\big)^+\big(\mu^t - \mu^{\pi,\widehat{p}^t}\big) = (\widehat{g}^t)^T\big(B^{\widehat{p}^t}\big)^+ B^{\widehat{p}^t}\big(\Lambda^{-1}_{\widehat{p}^t}(\mu^t) - \Lambda^{-1}_{\widehat{p}^t}\big(\mu^{\pi,\widehat{p}^t}\big)\big) = (\widehat{g}^t)^T\big(\Lambda^{-1}_{\widehat{p}^t}(\mu^t) - \Lambda^{-1}_{\widehat{p}^t}\big(\mu^{\pi,\widehat{p}^t}\big)\big)$$

since $\big(B^{\widehat{p}^t}\big)^+ B^{\widehat{p}^t} = \mathbf{I}_{NS(A-1)}$, see App. E.1. Therefore,

$$\begin{aligned}
\mathbb{E}\sum_{t=1}^T \big\langle \nabla \widehat{F}^t(\mu^t), \mu^t - \mu^{\pi,\widehat{p}^t} \big\rangle &= \mathbb{E}\sum_{t=1}^T \Big\langle \widehat{g}^t, \Lambda^{-1}_{\widehat{p}^t}(\mu^t) - \Lambda^{-1}_{\widehat{p}^t}\big(\mu^{\pi,\widehat{p}^t}\big) \Big\rangle \\
&= \mathbb{E}\sum_{t=1}^T \Big\langle g^t, \Lambda^{-1}_{\widehat{p}^t}(\mu^t) - \Lambda^{-1}_{\widehat{p}^t}\big(\mu^{\pi,\widehat{p}^t}\big) \Big\rangle + \mathbb{E}\sum_{t=1}^T \Big\langle \widehat{g}^t - g^t, \Lambda^{-1}_{\widehat{p}^t}(\mu^t) - \Lambda^{-1}_{\widehat{p}^t}\big(\mu^{\pi,\widehat{p}^t}\big) \Big\rangle \\
&= \mathbb{E}\sum_{t=1}^T \big\langle \mathring{g}^t, \mu^t - \mu^{\pi,\widehat{p}^t} \big\rangle + \mathbb{E}\sum_{t=1}^T \Big\langle \widehat{g}^t - g^t, \Lambda^{-1}_{\widehat{p}^t}(\mu^t) - \Lambda^{-1}_{\widehat{p}^t}\big(\mu^{\pi,\widehat{p}^t}\big) \Big\rangle,
\end{aligned}$$

where the last equality follows from the definition of $\mathring{g}^t$ (see Alg. 3) and the fact that $\mu^t$ and $\mu^{\pi,\widehat{p}^t}$ are expansions of $\Lambda^{-1}_{\widehat{p}^t}(\mu^t)$ and $\Lambda^{-1}_{\widehat{p}^t}\big(\mu^{\pi,\widehat{p}^t}\big)$ respectively, augmented with the entries corresponding to action $a^*$. Focusing on the second sum, we have that

$$\begin{aligned}
\sum_{t=1}^T \Big\langle \widehat{g}^t - g^t, \Lambda^{-1}_{\widehat{p}^t}(\mu^t) - \Lambda^{-1}_{\widehat{p}^t}\big(\mu^{\pi,\widehat{p}^t}\big) \Big\rangle &\leqslant \sum_{t=1}^T \|\widehat{g}^t - g^t\|_\infty \big\|\Lambda^{-1}_{\widehat{p}^t}(\mu^t) - \Lambda^{-1}_{\widehat{p}^t}\big(\mu^{\pi,\widehat{p}^t}\big)\big\|_1 \\
&\leqslant \sum_{t=1}^T \|\widehat{g}^t - g^t\|_\infty \big\|\mu^t - \mu^{\pi,\widehat{p}^t}\big\|_1 \\
&\leqslant 2N \sum_{t=1}^T \|\widehat{g}^t - g^t\|_\infty \\
&= 2\frac{1-\delta}{\delta\kappa} N^2 S(A-1) \sum_{t=1}^T \|\boldsymbol{u}^t\|_\infty |F^t(\widehat{\mu}^t) - F^t(\mu^{\pi^t,p})| \\
&\leqslant \frac{4}{\varepsilon\delta} N^2 S A^2 \sum_{t=1}^T |F^t(\widehat{\mu}^t) - F^t(\mu^{\pi^t,p})| \\
&= \frac{4}{\varepsilon\delta} N^2 S A^2 \sum_{t=1}^T |F^t(\mu^{\pi^t,\widehat{p}^t}) - F^t(\mu^{\pi^t,p})| \\
&\leqslant \frac{4}{\varepsilon\delta} L N^2 S A^2 \sum_{t=1}^T \big\|\mu^{\pi^t,\widehat{p}^t} - \mu^{\pi^t,p}\big\|_1,
\end{aligned}$$

where the fourth inequality uses that $\kappa \geqslant \frac{\varepsilon}{2A}$ and that $\boldsymbol{u}^t \in \mathbb{S}^{NS(A-1)}$, and the last inequality uses the Lipschitz continuity of $F^t$. As shown in (57), we have that

$$\mathbb{E}\sum_{t=1}^{T}\left\|\mu^{\pi^t,\widehat{p}^t} - \mu^{\pi^t,p}\right\|_1 \leqslant 3N^2\sqrt{SAT}C'_{1/T} + 2SN^2\sqrt{2T\log(NT)} + 4N\,.$$

Hence,

$$\mathbb{E}\sum_{t=1}^{T}\left\langle \nabla \widehat{F}^t(\mu^t), \mu^t - \mu^{\pi,\widehat{p}^t}\right\rangle \leqslant \mathbb{E}\sum_{t=1}^{T}\left\langle \mathring{g}^t, \mu^t - \mu^{\pi,\widehat{p}^t}\right\rangle + \frac{4}{\varepsilon\delta}LN^3SA^2\big(3N\sqrt{SAT}C'_{1/T} + 2SN\sqrt{2T\log(NT)} + 4\big)\,.$$

Combining this result with (60) yields that

$$\begin{aligned}
\mathbb{E}\left[R_T(\pi)\right] &\leqslant \mathbb{E}\sum_{t=1}^{T}\left\langle \mathring{g}^t - b^t, \mu^t - \mu^{\pi,\widehat{p}^t}\right\rangle + \frac{4}{\varepsilon\delta}LN^3SA^2\big(3N\sqrt{SAT}C'_{1/T} + 2SN\sqrt{2T\log(NT)} + 4\big) \\
&\quad + 2\delta L(2 + C'_{1/T}N)NT + 3LC'_{1/T}N^3\big[3C'_{1/T}\sqrt{SAT} + 2S\sqrt{2T\log(NT)}\big] + 4LN(3 + 2C'_{1/T}N) \\
&\leqslant \mathbb{E}\sum_{t=1}^{T}\left\langle \mathring{g}^t - b^t, \mu^t - \mu^{\pi,\widehat{p}^t}\right\rangle + \delta\,\underbrace{2L(2 + C'_{1/T}N)NT}_{=:\Xi_1} + 4LN(3 + 2C'_{1/T}N) \\
&\quad + \frac{1}{\delta}\underbrace{\frac{7}{\varepsilon}LC'_{1/T}N^3SA^2\big(3N\sqrt{AT}C'_{1/T} + 2N\sqrt{2ST\log(NT)} + 4\big)}_{=:\Xi_2}\,, \qquad (62)
\end{aligned}$$

where the last inequality uses that $C'_{1/T} \geqslant \sqrt{S}$. Note that

$$\begin{aligned}
\|\mathring{g}^t\|_{1,\infty} &= \sum_{n=1}^{N}\|g_n^t\|_\infty \\
&= \frac{1-\delta}{\varepsilon\delta}NS(A-1)(A-1+\sqrt{A-1})F^t(\mu^{\pi^t,p})\sum_{n=1}^{N}\|\boldsymbol{u}_n^t\|_\infty \\
&\leqslant \frac{2}{\varepsilon\delta}N^2SA^2\sum_{n=1}^{N}\|\boldsymbol{u}_n^t\|_\infty \\
&\leqslant \frac{2}{\varepsilon\delta}N^2SA^2\sqrt{N}\sqrt{\sum_{n=1}^{N}\|\boldsymbol{u}_n^t\|_\infty^2} \leqslant \frac{2}{\varepsilon\delta}N^{5/2}SA^2\sqrt{\sum_{n=1}^{N}\sum_{x,a}|\boldsymbol{u}_n^t(x,a)|^2} \leqslant \frac{2}{\varepsilon\delta}N^{5/2}SA^2\,,
\end{aligned}$$

where the second inequality uses Cauchy-Schwarz and the last inequality uses that $\boldsymbol{u}^t \in \mathbb{S}^{NS(A-1)}$. Moreover, we have that $\|b^t\|_{1,\infty} = \sum_{n=1}^{N}\|b_n^t\|_\infty \leqslant \sum_{n=1}^{N}L(N-n)C'_{1/T} \leqslant LN^2C'_{1/T}$. Hence, using that $C'_{1/T} \geqslant \sqrt{S}$,

$$\|\mathring{g}^t - b^t\|_{1,\infty} \leqslant \frac{2}{\varepsilon\delta}N^{5/2}SA^2 + LN^2C'_{1/T} \leqslant \frac{2}{\varepsilon\delta}(L+1)C'_{1/T}\sqrt{S}A^2N^{5/2}\,.$$

Via Lems. A.4 and E.11,[5] we can invoke Lem. 2.1 with $c = 5e$, $\zeta = \frac{2}{\varepsilon\delta}(L+1)C'_{1/T}\sqrt{S}A^2N^{5/2}$, and $\alpha_t = 1/(t+1)$ to get that (from the proof of Lem. 2.1)

$$\begin{aligned}
\sum_{t=1}^{T}\left\langle \mathring{g}^t - b^t, \mu^t - \mu^{\pi,\widehat{p}^t}\right\rangle &\leqslant \tau\Big(\frac{2}{\varepsilon\delta}(L+1)C'_{1/T}\sqrt{S}A^2N^{5/2}\Big)^2T + \frac{20e^2SN\log(AT)^2(N+A)}{\tau} \\
&\qquad + \frac{10e^2}{\varepsilon\delta}(L+1)C'_{1/T}S^{3/2}A^2N^{7/2}\log(T)\,.
\end{aligned}$$

[5]To invoke Lem. E.11, we assume without loss of generality that the constant $\varepsilon$ specified in Asm. 4.2 satisfies $\varepsilon \leqslant \frac{1}{2S}$.

Tuning $\tau$ optimally yields that

$$\sum_{t=1}^{T}\left\langle \mathring{g}^t - b^t, \mu^t - \mu^{\pi,\widehat{p}^t}\right\rangle \leqslant \frac{4}{\varepsilon\delta}(L+1)C'_{1/T}SA^2N^{5/2}\sqrt{20e^2N(N+A)T\log(AT)^2} + \frac{10e^2}{\varepsilon\delta}(L+1)C'_{1/T}S^{3/2}A^2N^{7/2}\log(T)$$
$$\leqslant \frac{1}{\delta}\underbrace{\frac{10e^2}{\varepsilon}(L+1)C'_{1/T}SA^2N^3\log(AT)\big(\sqrt{(N+A)T}+\sqrt{SN}\big)}_{=:\Xi_3}.$$

Hence, plugging back into (62) yields that

$$\mathbb{E}\left[R_T(\pi)\right] \leqslant \delta\Xi_1 + \frac{1}{\delta}(\Xi_2+\Xi_3) + 4LN(3+2C'_{1/T}N).$$

Setting $\delta := \min\left\{1, \sqrt{\frac{\Xi_2+\Xi_3}{\Xi_1}}\right\}$, we get that

$$\mathbb{E}\left[R_T(\pi)\right] \leqslant \max\left\{2\sqrt{\Xi_1(\Xi_2+\Xi_3)}, 2(\Xi_2+\Xi_3)\right\} + 4LN(3+2C'_{1/T}N).$$

Consequently, the theorem follows after using the definition of $C'_{1/T}$ from Eq. (55) and ignoring log factors. □

### E.3. Self-Concordant Regularization Approach

We have used the set $(\mathcal{M}^p_{\mu_0})^-$, the preimage of $\mathcal{M}^p_{\mu_0}$ under the map $\Xi_p$ (or $\Lambda_p$), to represent in $\mathbb{R}^{NS(A-1)}$ the set of valid occupancy measures. A more concise characterization, given by Lem. E.1, is that

$$(\mathcal{M}^p_{\mu_0})^- = \left\{\xi \in \mathbb{R}^{NS(A-1)} \colon B^p\xi \geqslant -\boldsymbol{\beta}^p\right\};$$

in other words, $(\mathcal{M}^p_{\mu_0})^-$ is a convex polytope formed by the constraints $B^p(n,x,a,\cdot,\cdot,\cdot)^\intercal\xi + \boldsymbol{\beta}^p_n(x,a) \geqslant 0$ for $n,x,a \in [N]\times\mathcal{X}\times\mathcal{A}$. Moreover, Lem. E.2 asserts that int $(\mathcal{M}^p_{\mu_0})^-$, the interior of $(\mathcal{M}^p_{\mu_0})^-$, is not empty under Asm. 4.4.

We consider then the function $\psi_{\text{lb}} \colon \text{int}\,(\mathcal{M}^p_{\mu_0})^- \to \mathbb{R}$ defined as

$$\psi_{\text{lb}}(\xi) := -\sum_{n,x,a}\log\big(B(n,x,a,\cdot,\cdot,\cdot)^\intercal\xi + \boldsymbol{\beta}_n(x,a)\big).$$

As mentioned in Sec. 4.2.2, Corollary 3.1.1 in (Nemirovski, 2004) yields that $\psi_{\text{lb}}$ is a $\vartheta$-self-concordant barrier (see Definition 3.1.1 in Nemirovski, 2004) for $(\mathcal{M}^p_{\mu_0})^-$ with $\vartheta = N\cdot S\cdot A$. The approach we analyze here is to perform OMD directly on the set $(\mathcal{M}^p_{\mu_0})^-$ with $\psi_{\text{lb}}$ as the regularizer.

For $\xi \in \text{int}\,(\mathcal{M}^p_{\mu_0})^-$ and $y \in \mathbb{R}^{NS(A-1)}$, define the local norm $\|y\|_\xi := \sqrt{y^\intercal\nabla^2\psi_{\text{lb}}(\xi)y}$. This is indeed a norm since the fact that $(\mathcal{M}^p_{\mu_0})^-$ is bounded implies via Property II in (Nemirovski, 2004, Section 2.2) that the Hessian of $\psi_{\text{lb}}$ is non-singular everywhere. Its dual norm is denoted as $\|y\|_{\xi,*} := \sqrt{y^\intercal(\nabla^2\psi_{\text{lb}}(\xi))^{-1}y}$. The Dikin ellipsoid of radius $r$ at $\xi \in \text{int}\,(\mathcal{M}^p_{\mu_0})^-$ is given by

$$\mathcal{E}_r(\xi) := \{y \in \mathbb{R}^{NS(A-1)} \colon \|y-\xi\|_\xi \leqslant r\} = \xi + r(\nabla^2\psi_{\text{lb}}(\xi))^{-1/2}\mathbb{B}^{NS(A-1)}.$$

Via Property I in (Nemirovski, 2004, Section 2.2), $\mathcal{E}_1(\xi) \subseteq (\mathcal{M}^p_{\mu_0})^-$ for any $\xi \in \text{int}\,(\mathcal{M}^p_{\mu_0})^-$.

For $\xi, y \in \text{int}\,(\mathcal{M}^p_{\mu_0})^-$, we denote by $D_{\psi_{\text{lb}}}(y,\xi) := \psi_{\text{lb}}(y) - \psi_{\text{lb}}(\xi) - \langle y-\xi, \nabla\psi_{\text{lb}}(\xi)\rangle$ the Bregman divergence between $y$ and $\xi$ with respect to $\psi_{\text{lb}}$. From the proof of Thm. E.16, we recall the definition $\mathbb{F}^t := F^t \circ \Lambda_p$. As alluded to above, our OMD updates will take the form

$$\xi^{t+1} \leftarrow \operatorname*{arg\,min}_{\xi\in(\mathcal{M}^p_{\mu_0})^-} \tau\left\langle g^t, \xi\right\rangle + D_{\psi_{\text{lb}}}(\xi,\xi^t),$$

where $g^t$ will be chosen as a surrogate for $\nabla\mathbb{F}^t(\xi^t)$. Differently from the proof of Thm. E.16, we redefine the smoothed approximation $\widehat{\mathbb{F}}^t \colon (\mathcal{M}^p_{\mu_0})^- \to \mathbb{R}$ such that

$$\widehat{\mathbb{F}}^t(\xi) := \mathbb{E}_{\boldsymbol{v}\in\mathbb{B}^{NS(A-1)}}\left[\mathbb{F}^t\big((1-\delta)\xi + \delta\big(\xi^t + (\nabla^2\psi_{\text{lb}}(\xi^t))^{-1/2}\boldsymbol{v}\big)\big)\right].$$

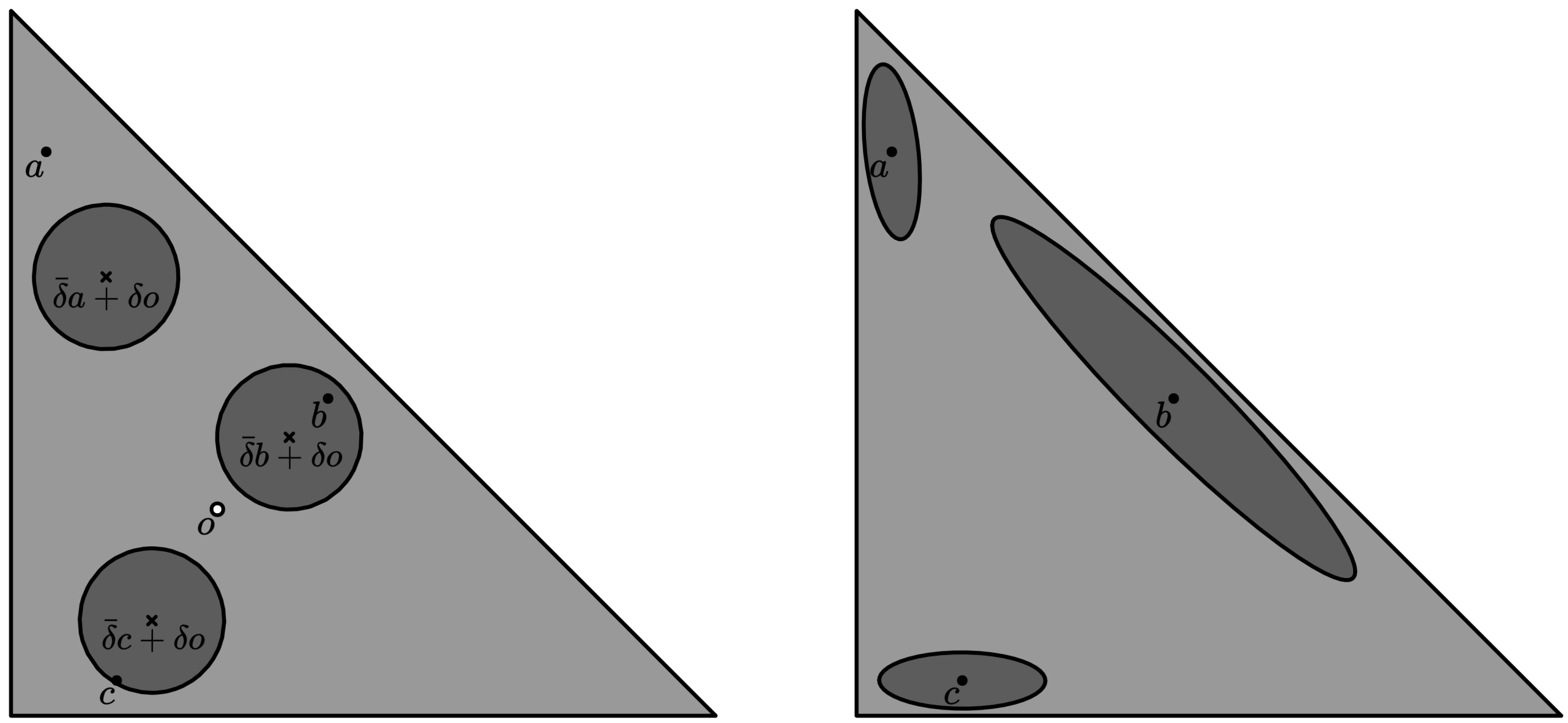

*Figure 4.* This figure provides a graphical comparison between the sampling approach used in Alg. 3, represented on the left, and that used in Alg. 4, represented on the right. The simplified domain here is $\{x \in [0,1]^2 \colon \|x\|_1 \leqslant 1\}$. Both approaches are illustrated at three points: $a$, $b$, and $c$. In the first approach, with some $\delta \in (0,1)$ and $\bar{\delta} := 1-\delta$, we sample from a circle of radius $\delta/(2+\sqrt{2})$ centered at a convex combination between the point of interest and $o := \big(1/(2+\sqrt{2}), 1/(2+\sqrt{2})\big)$. In the second approach, we consider the barrier $-\log(1-x_1-x_2) - \sum_{i=1,2}\log(x_i)$ and sample from the Dikin ellipsoid (of a certain common radius) induced by this function at each point.

This is well-defined since we are evaluating $\mathbb{F}^t$ on a convex combination of the argument $\xi$ and a point inside the ellipsoid $\mathcal{E}_1(\xi^t)$, which is a subset of $(\mathcal{M}^p_{\mu_0})^-$ as cited before. Via Corollary 6.8 in (Hazan et al., 2016) and the chain rule, we have that

$$\nabla\widehat{\mathbb{F}}^t(\xi) = \frac{(1-\delta)}{\delta} NS(A-1)\mathbb{E}_{\boldsymbol{u}\in\mathbb{S}^{NS(A-1)}}\big[\mathbb{F}^t\big((1-\delta)\xi + \delta\big(\xi^t + (\nabla^2\psi_{\text{lb}}(\xi^t))^{-1/2}\boldsymbol{u}\big)\big)(\nabla^2\psi_{\text{lb}}(\xi^t))^{1/2}\boldsymbol{u}\big]. \tag{63}$$

Hence, with $\boldsymbol{u}^t$ sampled uniformly from $\mathbb{S}^{NS(A-1)}$, we pick (as mentioned in Sec. 4.2.2)

$$g^t := \frac{(1-\delta)}{\delta} NS(A-1)\mathbb{F}^t\big(\xi^t + \delta(\nabla^2\psi_{\text{lb}}(\xi^t))^{-1/2}\boldsymbol{u}^t\big)(\nabla^2\psi_{\text{lb}}(\xi^t))^{1/2}\boldsymbol{u}^t \tag{64}$$

such that $\mathbb{E}_{\boldsymbol{u}^t}\big[g^t\big] = \nabla\widehat{\mathbb{F}}^t(\xi^t)$, see also (Saha & Tewari, 2011) for a similar estimator in another BCO setting. We summarize this approach in Alg. 4, and provide in Fig. 4 a graphical comparison with the sampling approach of Alg. 3 on a simple decision set. Before proving the regret bound of Thm. 4.5, we collect a few standard properties and auxiliary results concerning self-concordant barriers and their use as regularizers.

### E.3.1. Auxiliary Lemmas

For $x, y \in \text{int}\,(\mathcal{M}^p_{\mu_0})^-$, it holds via Property I in (Nemirovski, 2004, Section 2.2) that

$$(1-\|y-x\|_x)^2\nabla^2\psi_{\text{lb}}(x) \preccurlyeq \nabla^2\psi_{\text{lb}}(y) \preccurlyeq \frac{1}{(1-\|y-x\|_x)^2}\nabla^2\psi_{\text{lb}}(x) \tag{65}$$

whenever $\|y-x\|_x < 1$. We state the following auxiliary lemma, which will be used to assert the proximity between $\xi^t$ and $\xi^{t+1}$ for our algorithm. Establishing this 'stability' is a crucial step in the local norm analysis.

**Lemma E.17.** *Let $x \in$ int $(\mathcal{M}^p_{\mu_0})^-$ and $\ell \in \mathbb{R}^{NS(A-1)}$ be such that $\|\ell\|_{x,*} \leqslant \frac{1}{16}$, and define*

$$y := \mathop{\arg\min}_{\xi \in \textit{int}\,(\mathcal{M}^p_{\mu_0})^-} \langle \ell, \xi\rangle + D_{\psi_{lb}}(\xi, x)\,.$$

*Then, $y \in \mathcal{E}_{1/2}(x)$.*

**Algorithm 4** Bandit O-MD-CURL with logarithmic barrier regularization

**input:** domain $(\mathcal{M}^p_{\mu_0})^-$ with non-empty interior, learning rate $\tau > 0$, exploration parameter $\delta \in (0, 1]$
**initialization:** $\xi^1 \leftarrow \arg\min_{\xi \in \text{int}\,(\mathcal{M}^p_{\mu_0})^-} \psi_{\text{lb}}(\xi)$
**for** $t = 1, \dots, T$ **do**
  draw $\boldsymbol{u}^t \in \mathbb{S}^{NS(A-1)}$ uniformly at random
  $\widehat{\xi}^t \leftarrow \xi^t + \delta(\nabla^2\psi_{\text{lb}}(\xi^t))^{-1/2}\boldsymbol{u}^t$
  $\widehat{\mu}^t \leftarrow \Lambda_p(\widehat{\xi}^t)$
  $\pi^t_n(a|x) \leftarrow \widehat{\mu}^t(x, a)/\sum_{a\in\mathcal{A}} \widehat{\mu}^t(x, a)$
  output $\pi^t$ and observe $F^t(\widehat{\mu}^t)$
  $g^t \leftarrow \frac{(1-\delta)}{\delta} NS(A-1)F^t(\widehat{\mu}^t)(\nabla^2\psi_{\text{lb}}(\xi^t))^{1/2}\boldsymbol{u}^t$
  $\xi^{t+1} \leftarrow \arg\min_{\xi \in \text{int}\,(\mathcal{M}^p_{\mu_0})^-} \tau\,\langle g^t, \xi\rangle + D_{\psi_{\text{lb}}}(\xi, \xi^t)$
**end for**

*Proof.* For $\xi \in \text{int}\,(\mathcal{M}^p_{\mu_0})^-$, let

$$g(\xi) := \langle \ell, \xi\rangle + D_{\psi_{\text{lb}}}(\xi, x) = \langle \ell, \xi\rangle + \psi_{\text{lb}}(\xi) - \psi_{\text{lb}}(x) - \langle \xi - x, \nabla\psi_{\text{lb}}(x)\rangle\,.$$

Note that $g$ is a self-concordant function on int $(\mathcal{M}^p_{\mu_0})^-$ (Item (ii) in Nemirovski, 2004, Proposition 2.1.1), whose Hessian (hence, local norms and Dikin ellipsoids) coincides with that of $\psi_{\text{lb}}$ everywhere. Moreover, $g$ is below bounded thanks to $(\mathcal{M}^p_{\mu_0})^-$ being a bounded set, which implies that $g$ attains its minimum on int $(\mathcal{M}^p_{\mu_0})^-$ (Property VI in Nemirovski, 2004, Section 2.2). This minimum is also unique via strict convexity. Hence, $y$ is well-defined.

The rest of the proof is similar to the proof of Lem. 13 in (Wei & Luo, 2018) and Lem. 9 in (Van der Hoeven et al., 2023). Thanks to the strict convexity of $g$, to show that $y \in \mathcal{E}_{1/2}(x)$ it suffices to show that for any $\xi$ on the boundary of $\mathcal{E}_{1/2}(x)$, $g(x) \leqslant g(\xi)$; this is because $x \in \mathcal{E}_{1/2}(x)$ and $y = \arg\min_{\xi\in(\mathcal{M}^p_{\mu_0})^-} g(\xi)$. For any such $\xi$ on the boundary of $\mathcal{E}_{1/2}(x)$, Taylor's theorem implies that there exists some $z$ on the line segment between $x$ and $\xi$ such that

$$\begin{aligned}
g(\xi) - g(x) &= \langle \xi - x, \nabla g(x)\rangle + \frac{1}{2}(\xi - x)^\intercal\nabla^2 g(z)(\xi - x)\\
&= \langle \xi - x, \ell\rangle + \frac{1}{2}(\xi - x)^\intercal\nabla^2\psi_{\text{lb}}(z)(\xi - x)\\
&\geqslant \langle \xi - x, \ell\rangle + \frac{1}{8}(\xi - x)^\intercal\nabla^2\psi_{\text{lb}}(x)(\xi - x)\\
&= \langle \xi - x, \ell\rangle + \frac{1}{8}\|\xi - x\|_x^2\\
&\geqslant -\|\xi - x\|_x\|\ell\|_{x,*} + \frac{1}{8}\|\xi - x\|_x^2\\
&= -\frac{1}{2}\|\ell\|_{x,*} + \frac{1}{32} \geqslant 0\,,
\end{aligned}$$

where the second equality holds since $\nabla^2 g = \nabla^2\psi_{\text{lb}}$ and $\nabla g(x) = \ell + \nabla\psi_{\text{lb}}(x) - \nabla\psi_{\text{lb}}(x) = \ell$, the first inequality holds via (65) and the fact that $z \in \mathcal{E}_{1/2}(x)$, the second inequality holds via the definition of a dual norm, the last equality holds since $\xi$ is on the boundary of $\mathcal{E}_{1/2}(x)$, and the last inequality holds via the assumption that $\|\ell\|_{x,*} \leqslant \frac{1}{16}$. ◻

For $x \in \text{int}\,(\mathcal{M}^p_{\mu_0})^-$, the Minkowski function of $(\mathcal{M}^p_{\mu_0})^-$ with the pole at $x$ is defined as (Nemirovski, 2004, Section 3.2)

$$\pi_x(y) := \inf\{t > 0 \colon x + t^{-1}(y - x) \in (\mathcal{M}^p_{\mu_0})^-\}$$

for $y \in (\mathcal{M}^p_{\mu_0})^-$. The following lemma readily follows from the properties of the Minkowski function. It is used in the analysis to handle the bias term of the standard OMD regret guarantee, which is slightly more involved in this case considering that the comparator need not belong to the interior of $(\mathcal{M}^p_{\mu_0})^-$, where $\psi_{\text{lb}}$ is defined (and finite).

**Lemma E.18.** *Let $x \in$ int $(\mathcal{M}^p_{\mu_0})^-$, $y \in (\mathcal{M}^p_{\mu_0})^-$, $\delta \in (0,1)$, and $z := (1-\delta)y + \delta x$. Then,*

$$\psi_{lb}(z) \leqslant \psi_{lb}(x) + NSA \log \delta^{-1} .$$

*Further, let $\dot{x} := \arg\min_{x \in \text{int}\,(\mathcal{M}^p_{\mu_0})^-} \psi_{lb}(x)$ and $\dot{z} := (1-\delta)y + \delta \dot{x}$. Then,*

$$D_{\psi_{lb}}(\dot{z}, \dot{x}) \leqslant NSA \log \delta^{-1} .$$

*Proof.* Since $\psi_{\text{lb}}$ is an $NSA$-self-concordant barrier for $\mathcal{M}^p_{\mu_0}$, Property II in (Nemirovski, 2004, Section 3.2) implies that

$$\psi_{\text{lb}}(z) \leqslant \psi_{\text{lb}}(x) + NSA \log \frac{1}{1 - \pi_x(z)} .$$

On the other hand,

$$x + (1-\delta)^{-1}(z - x) = x + (1-\delta)^{-1}((1-\delta)y + \delta x - x) = x + y - x = y \in (\mathcal{M}^p_{\mu_0})^- ,$$

implying that $\pi_x(z) \leqslant 1 - \delta$. Hence, $\psi_{\text{lb}}(z) \leqslant \psi_{\text{lb}}(x) + NSA \log \delta^{-1}$.

Next, we note that the optimality of $\dot{x}$ implies that

$$D_{\psi_{\text{lb}}}(\dot{z}, \dot{x}) = \psi_{\text{lb}}(\dot{z}) - \psi_{\text{lb}}(\dot{x}) - \langle \dot{z} - \dot{x}, \nabla \psi_{\text{lb}}(\dot{x}) \rangle \leqslant \psi_{\text{lb}}(\dot{z}) - \psi_{\text{lb}}(\dot{x}) ,$$

which concludes the proof when combined with the first part. □

### E.3.2. Regret Analysis

We are now ready to prove the regret bound of Thm. 4.5, which is stated more explicitly in the following theorem.

**Theorem E.19.** *Under Asm. 4.4, Alg. 4 with $\tau = \frac{\delta}{16}\sqrt{\frac{\log T}{N^3 SAT}}$ and $\delta = \min\left\{\sqrt{\frac{17}{4L}}\frac{N^{3/4}S^{3/4}A^{3/4}(\log T)^{1/4}}{T^{1/4}}, 1\right\}$ satisfies for any policy $\pi \in \Pi$ that*

$$\mathbb{E}\left[R_T(\pi)\right] \leqslant \max\left\{4\sqrt{17L}N^{7/4}\left(SAT\right)^{3/4}(\log T)^{1/4}, 34\sqrt{N^5 S^3 A^3 T \log T}\right\} + 2LN .$$

*Proof.* Firstly, we assert that the iterates $\xi^t$ are well defined; similar to what was argued in the proof of Lem. E.17, the functions $\psi_{\text{lb}}(\cdot)$ and $\tau\langle g^t, \cdot\rangle + D_{\psi_{\text{lb}}}(\cdot, \xi^t)$ are self-concordant on int $(\mathcal{M}^p_{\mu_0})^-$ (Item (ii) in Nemirovski, 2004, Proposition 2.1.1) and bounded from below thanks to $(\mathcal{M}^p_{\mu_0})^-$ being a bounded set, implying via Property VI in (Nemirovski, 2004, Section 2.2) that each of these functions attains its minimum on int $(\mathcal{M}^p_{\mu_0})^-$, which is also unique via strict convexity. Also note that indeed $\widehat{\mu}^t \in \mathcal{M}^p_{\mu_0}$ since $\widehat{\xi}^t \in (\mathcal{M}^p_{\mu_0})^-$ as we argued before presenting the algorithm.

Let $\mu^* \in \arg\min_{\mu \in \mathcal{M}^p_{\mu_0}} \sum_{t=1}^T F^t(\mu)$ and $\bar{R}_T := \mathbb{E}\sum_{t=1}^T\left(F^t(\mu^{\pi^t,p}) - F^t(\mu^*)\right)$, which satisfies $\bar{R}_T = \max_{\pi \in \Pi}\mathbb{E}\left[R_T(\pi)\right]$. Define $\xi^* := (1-\dot{\delta})\Lambda_p^{-1}(\mu^*) + \dot{\delta}\xi^1$, where $\dot{\delta} \in (0,1)$ is a constant to be specified later. To start with, we have that

$$\bar{R}_T = \mathbb{E}\sum_{t=1}^T\left(F^t(\mu^{\pi^t,p}) - F^t(\mu^*)\right) = \mathbb{E}\sum_{t=1}^T\left(F^t(\widehat{\mu}^t) - F^t(\mu^*)\right) = \mathbb{E}\sum_{t=1}^T\left(\mathbb{F}^t(\widehat{\xi}^t) - \mathbb{F}^t\left(\Lambda_p^{-1}(\mu^*)\right)\right) .$$

Next, we derive that

$$\begin{aligned}
&\mathbb{F}^t(\widehat{\xi}^t) - \widehat{\mathbb{F}}^t(\xi^t) \\
&\quad = \mathbb{F}^t\left(\xi^t + \delta(\nabla^2\psi_{\text{lb}}(\xi^t))^{-1/2}\boldsymbol{u}^t\right) - \mathbb{E}_{\boldsymbol{v}\in\mathbb{B}^{NS(A-1)}}\left[\mathbb{F}^t\left(\xi^t + \delta(\nabla^2\psi_{\text{lb}}(\xi^t))^{-1/2}\boldsymbol{v}\right)\right] \\
&\quad \leqslant L\mathbb{E}_{\boldsymbol{v}\in\mathbb{B}^{NS(A-1)}}\left\|\Lambda_p\left(\xi^t + \delta(\nabla^2\psi_{\text{lb}}(\xi^t))^{-1/2}\boldsymbol{u}^t\right) - \Lambda_p\left(\xi^t + \delta(\nabla^2\psi_{\text{lb}}(\xi^t))^{-1/2}\boldsymbol{v}\right)\right\|_1 \\
&\quad = \delta L\mathbb{E}_{\boldsymbol{v}\in\mathbb{B}^{NS(A-1)}}\left\|\Lambda_p\left((\nabla^2\psi_{\text{lb}}(\xi^t))^{-1/2}\boldsymbol{u}^t\right) - \Lambda_p\left((\nabla^2\psi_{\text{lb}}(\xi^t))^{-1/2}\boldsymbol{v}\right)\right\|_1 \\
&\quad = \delta L\mathbb{E}_{\boldsymbol{v}\in\mathbb{B}^{NS(A-1)}}\left\|\Lambda_p\left(\xi^t + (\nabla^2\psi_{\text{lb}}(\xi^t))^{-1/2}\boldsymbol{u}^t\right) - \Lambda_p\left(\xi^t + (\nabla^2\psi_{\text{lb}}(\xi^t))^{-1/2}\boldsymbol{v}\right)\right\|_1
\end{aligned}$$

$$\leqslant \delta L \mathbb{E}_{\boldsymbol{v} \in \mathbb{B}^{NS(A-1)}} \left[ \left\| \Lambda_p\big(\xi^t + (\nabla^2 \psi_{\text{lb}}(\xi^t))^{-1/2} \boldsymbol{u}^t\big) \right\|_1 + \left\| \Lambda_p\big(\xi^t + (\nabla^2 \psi_{\text{lb}}(\xi^t))^{-1/2} \boldsymbol{v}\big) \right\|_1 \right]$$
$$\leqslant 2\delta L N \,,$$

where the first inequality uses the Lipschitz smoothness of $F^t$ and the fact that $\mathbb{F}^t = F^t \circ \Lambda_p$; the second and third equalities use the fact that $\Lambda_p$ is an affine function; and the last inequality holds since both $\xi^t + (\nabla^2 \psi_{\text{lb}}(\xi^t))^{-1/2} \boldsymbol{u}^t$, $\xi^t + (\nabla^2 \psi_{\text{lb}}(\xi^t))^{-1/2} \boldsymbol{v} \in \mathcal{E}_1(\xi^t) \subset (\mathcal{M}^p_{\mu_0})^-$, and that for any $\xi \in (\mathcal{M}^p_{\mu_0})^-$, $\Lambda_p(\xi) \in \mathcal{M}^p_{\mu_0}$ and therefore satisfies $\|\Lambda_p(\xi)\|_1 \leqslant N$. We similarly derive that

$$\begin{aligned}
&\widehat{\mathbb{F}}^t(\xi^*) - \mathbb{F}^t\big(\Lambda_p^{-1}(\mu^*)\big) \\
&= \mathbb{E}_{\boldsymbol{v} \in \mathbb{B}^{NS(A-1)}} \big[\mathbb{F}^t\big((1-\delta)\xi^* + \delta\big(\xi^t + (\nabla^2 \psi_{\text{lb}}(\xi^t))^{-1/2} \boldsymbol{v}\big)\big)\big] - \mathbb{F}^t\big(\Lambda_p^{-1}(\mu^*)\big) \\
&= \mathbb{E}_{\boldsymbol{v} \in \mathbb{B}^{NS(A-1)}} \big[\mathbb{F}^t\big((1-\delta)(1-\dot{\delta})\Lambda_p^{-1}(\mu^*) + (1-\delta)\dot{\delta}\xi^1 + \delta\big(\xi^t + (\nabla^2 \psi_{\text{lb}}(\xi^t))^{-1/2} \boldsymbol{v}\big)\big)\big] \\
&\qquad\qquad\qquad\qquad\qquad\qquad\qquad\qquad\qquad\qquad\qquad\qquad - \mathbb{F}^t\big(\Lambda_p^{-1}(\mu^*)\big) \\
&\leqslant L \mathbb{E}_{\boldsymbol{v} \in \mathbb{B}^{NS(A-1)}} \big\|(1-\delta)(1-\dot{\delta})\mu^* + (1-\delta)\dot{\delta}\Lambda_p(\xi^1) + \delta\Lambda_p\big(\xi^t + (\nabla^2 \psi_{\text{lb}}(\xi^t))^{-1/2} \boldsymbol{v}\big) - \mu^*\big\|_1 \\
&\leqslant L \mathbb{E}_{\boldsymbol{v} \in \mathbb{B}^{NS(A-1)}} \Big[|\delta\dot{\delta} - \delta - \dot{\delta}|\|\mu^*\|_1 + (1-\delta)\dot{\delta}\big\|\Lambda_p(\xi^1)\big\|_1 + \delta\big\|\Lambda_p\big(\xi^t + (\nabla^2 \psi_{\text{lb}}(\xi^t))^{-1/2} \boldsymbol{v}\big)\big\|_1\Big] \\
&\leqslant L \mathbb{E}_{\boldsymbol{v} \in \mathbb{B}^{NS(A-1)}} \Big[(\delta + \dot{\delta})\|\mu^*\|_1 + \dot{\delta}\big\|\Lambda_p(\xi^1)\big\|_1 + \delta\big\|\Lambda_p\big(\xi^t + (\nabla^2 \psi_{\text{lb}}(\xi^t))^{-1/2} \boldsymbol{v}\big)\big\|_1\Big] \\
&\leqslant 2\delta L N + 2\dot{\delta} L N \,.
\end{aligned}$$

Hence, using also the convexity of $\widehat{\mathbb{F}}^t$, we obtain that

$$\begin{aligned}
\bar{R}_T &\leqslant \mathbb{E} \sum_{t=1}^T \big(\widehat{\mathbb{F}}^t(\xi^t) - \widehat{\mathbb{F}}^t(\xi^*)\big) + 4\delta LNT + 2\dot{\delta} LNT \\
&\leqslant \mathbb{E} \sum_{t=1}^T \big\langle \nabla \widehat{\mathbb{F}}^t(\xi^t), \xi^t - \xi^* \big\rangle + 4\delta LNT + 2\dot{\delta} LNT \,. \qquad (66)
\end{aligned}$$

In this proof, let $\mathcal{F}_t \coloneqq \sigma(u^1, \ldots, u^t)$ denote the $\sigma$-algebra generated by $u^1, \ldots, u^t$; and let $\mathbb{E}_t[\cdot] \coloneqq \mathbb{E}[\cdot \mid \mathcal{F}_{t-1}]$ with $\mathcal{F}_0$ being the trivial $\sigma$-algebra. We then have that

$$\widehat{\mathbb{F}}^t(\xi^t) = \mathbb{E}_{\boldsymbol{u}^t}\big[g^t\big] = \mathbb{E}_t\big[g^t\big] \,,$$

where the first equality follows from (63) and the the second equality holds since conditioned on $\mathcal{F}_{t-1}$, $\boldsymbol{u}^t$ is the only source of randomness in $g^t$ and is sampled identically and independently in every round. Using that $\xi^t - \xi^*$ is measurable with respect to $\mathcal{F}_{t-1}$, we then obtain that

$$\bar{R}_T \leqslant \mathbb{E} \sum_{t=1}^T \big\langle \mathbb{E}_t g^t, \xi^t - \xi^* \big\rangle + 4\delta LNT + 2\dot{\delta} LNT = \mathbb{E} \sum_{t=1}^T \big\langle g^t, \xi^t - \xi^* \big\rangle + 4\delta LNT + 2\dot{\delta} LNT \,.$$

Via the definition of $\xi^t$ and the fact that $\xi^* \in \text{int}\,(\mathcal{M}^p_{\mu_0})^-$, Lem. 6.16 in (Orabona, 2019) implies that

$$\sum_{t=1}^T \big\langle g^t, \xi^t - \xi^* \big\rangle \leqslant \frac{D_{\psi_{\text{lb}}}(\xi^*, \xi^1)}{\tau} + \sum_{t=1}^T \frac{\tau}{2} \|g_t\|^2_{\zeta^t,*} \,,$$

where $\zeta^t$ lies on the line segment between $\xi^t$ and $\xi^{t+1}$. We firstly observe that

$$\begin{aligned}
\|g_t\|^2_{\xi^t,*} &= \left(\frac{(1-\delta)}{\delta} NS(A-1) F^t(\widehat{\mu}^t)\right)^2 \cdot (\boldsymbol{u}^t)^\intercal (\nabla^2 \psi_{\text{lb}}(\xi^t))^{1/2} \big(\nabla^2 \psi_{\text{lb}}(\xi^t)\big)^{-1} (\nabla^2 \psi_{\text{lb}}(\xi^t))^{1/2} \boldsymbol{u}^t \\
&= \left(\frac{(1-\delta)}{\delta} NS(A-1) F^t(\widehat{\mu}^t)\right)^2 \leqslant \frac{1}{\delta^2} N^4 S^2 A^2 \,,
\end{aligned}$$

where we have used that $F^t(\widehat{\mu}^t) \leqslant N$. Hence, if

$$\tau \leqslant \frac{\delta}{16 N^2 SA} \,, \qquad (67)$$

then $\tau\|g_t\|_{\xi^t,*} \leqslant 1/16$. Consequently, Lem. E.17 (with $x = \xi^t$, $y = \xi^{t+1}$, and $\ell = \tau g_t$) would assert that $\xi^{t+1} \in \mathcal{E}_{1/2}(\xi^t)$; and hence, $\zeta^t \in \mathcal{E}_{1/2}(\xi^t)$. It would then hold via (65) that

$$\|g_t\|^2_{\zeta^t,*} \leqslant \frac{1}{(1 - \|\zeta^t - \xi^t\|_{\xi_t})^2}\|g_t\|^2_{\xi^t,*} \leqslant 4\|g_t\|^2_{\xi^t,*}\,,$$

On the other hand, via Lem. E.18 and the definitions of $\xi^1$ and $\xi^*$, we have that

$$D_{\psi_{\text{lb}}}(\xi^*, \xi^1) \leqslant NSA\log\dot{\delta}^{-1}\,.$$

Hence, conditioned on (67), we obtain the following regret bound

$$\bar{R}_T \leqslant \frac{NSA\log\dot{\delta}^{-1}}{\tau} + \frac{2\tau}{\delta^2}N^4S^2A^2T + 4\delta LNT + 2\dot{\delta}LNT\,. \tag{68}$$

Setting

$$\dot{\delta} = \frac{1}{T}\,, \quad \tau = \frac{\delta}{16}\sqrt{\frac{\log T}{N^3SAT}}\,, \quad \text{and} \quad \delta = \min\left\{\sqrt{\frac{17}{4L}}\frac{N^{3/4}S^{3/4}A^{3/4}(\log T)^{1/4}}{T^{1/4}}, 1\right\};$$

we obtain that

$$\begin{aligned}
\bar{R}_T &\leqslant \frac{NSA\log T}{\tau} + \frac{2\tau}{\delta^2}N^4S^2A^2T + 4\delta LNT + 2LN \\
&\leqslant \frac{17}{\delta}\sqrt{N^5S^3A^3T\log T} + 4\delta LNT + 2LN \\
&\leqslant \max\left\{4\sqrt{17L}N^{7/4}\,(SAT)^{3/4}\,(\log T)^{1/4}, 34\sqrt{N^5S^3A^3T\log T}\right\} + 2LN\,.
\end{aligned} \tag{69}$$

If $T \geqslant NSA\log(T)$, then our choice of $\tau$ indeed satisfies (67):

$$\tau = \frac{\delta}{16}\sqrt{\frac{\log T}{N^3SAT}} \leqslant \frac{\delta}{16N^2SA}\,.$$

Otherwise, we can fall back to the trivial regret bound

$$\bar{R}_T \leqslant NT \leqslant N^2SA\log(T)\,,$$

which is dominated by the bound in (69); hence, the theorem follows. □