# OpenReview forum: "Online Episodic Convex Reinforcement Learning"
_ICML.cc/2025/Conference — ICML 2025 poster_

### Official Review · Reviewer_1pSC · 2025-03-04

**Overall Recommendation:** 3

**Summary:**

This paper studies online learning in episodic finite-horizon Markov Decision Processes (MDPs) with convex objective functions, known as the concave utility reinforcement learning (CURL) problem. CURL generalizes RL by applying convex losses to state-action distributions induced by policies, rather than just linear losses. The paper's primary contributions are:

  (1) introducing the first algorithm achieving near-optimal O(√T) regret bounds for online CURL without prior knowledge of transition dynamics, using online mirror descent with varying constraint sets and exploration bonuses

  (2) addressing a bandit version of CURL where feedback is only the value of the objective function on the state-action distribution, achieving sublinear regret by adapting techniques from bandit convex optimization.

The authors develop theoretical guarantees for different feedback settings and provide empirical validation on multi-objective and constrained MDP tasks, demonstrating that the proposed approach outperforms previous methods on tasks requiring exploration.

**Claims And Evidence:**

The claims in this paper are generally well-supported by theoretical analysis and empirical evidence:
* The claim that the proposed algorithm achieves O(√T) regret for online CURL with full-information feedback and unknown transition dynamics is well-supported by Theorem 3.2, with detailed proof involving exploration bonuses and regret decomposition.
* The claims regarding sublinear regret for bandit feedback settings are supported by Theorems 4.1, 4.3, and 4.5, each with comprehensive theoretical analyses for different scenarios.
* The claim that the proposed approach outperforms previous methods (specifically Greedy MD-CURL) on tasks requiring exploration is supported by experimental results in Section 5, showing clear performance improvements on multi-objective and constrained MDP tasks.

The assumptions made for the bandit feedback setting (particularly Assumption 4.2) might be restrictive in some real-world scenarios, though the authors acknowledge this limitation and provide an alternative approach with a less restrictive assumption (4.4) for known MDPs.

However, I find the paper's novelty claims overstated (see "Essential References Not Discussed" section for more details):
* The claim of being the "first method achieving sub-linear regret for online CURL with adversarial losses and unknown transition kernels" appears to be incorrect given Rosenberg & Mansour's prior work.
* The technical approach of using exploration bonuses with mirror descent builds more incrementally on prior work than suggested, particularly on Jin et al. (2020).
* The adaptation of bandit convex optimization techniques to the MDP setting has precedents in earlier work by Neu et al. (2012/2013).

**Essential References Not Discussed:**

Although all papers below are referenced, there are a few crucial points that I believe significantly impact this work's novelty claims:

* Rosenberg & Mansour (2019) - This is potentially an important oversight. This paper pioneered online convex MDP methods under adversarial losses with unknown dynamics using UC-O-REPS (an OMD approach over occupancy measures) achieving Õ(√T) regret for convex performance criteria. This directly contradicts the paper's claim of being "the first algorithm achieving near-optimal regret bounds for online CURL with adversarial losses and unknown transition kernels."
* Neu et al. (2012/2013) - These earlier works had already adapted online convex optimization tools to MDPs with unknown transitions. Their Follow-the-Perturbed-Optimistic-Policy algorithm and O-REPS (entropy-regularized mirror descent) approaches achieved Õ(√T) regret in adversarial MDPs, establishing that bandit convex optimization methods can be successfully applied to reinforcement learning.
* Jin et al. (2020) - While cited in the paper, this contribution seems also not fully acknowledged. This paper already incorporated exploration bonuses into mirror-descent policy updates to achieve Õ(√T) regret for adversarial MDPs with bandit losses, meaning the idea of adding exploration bonuses to mirror-descent RL methods was already established.

**Experimental Designs Or Analyses:**

The experimental evaluation supports the theoretical claims but is somewhat limited. The authors evaluate Bonus O-MD-CURL against Greedy MD-CURL on multi-objective optimization and constrained MDPs.

Strengths of the experimental design:
* As far as I understand, the tasks are meaningful and good examples of CURL problems
* The visual representations of state distributions clearly demonstrate the benefit of exploration bonuses
* The regret/loss curves show consistent improvement over the baseline method

Limitations I identified:
* Experiments use fixed objective functions and probability kernels, not adversarial settings
* Only one environment (11×11 grid world) is used
* Only one baseline method (Greedy MD-CURL) is compared against
* Limited number of iterations (1000) and repetitions (5)

Given the primarily theoretical nature of the paper, the experiments provide reasonable evidence for the benefit of the proposed approach. However, as mentioned above, more extensive empirical validation would strengthen the paper more.

**Methods And Evaluation Criteria:**

The proposed methods are appropriate for the problem. The authors use online mirror descent with exploration bonuses to handle the exploration-exploitation tradeoff in the unknown dynamics setting. The regret analysis is thorough and establishes strong theoretical guarantees.

The evaluation consists of:
* Theoretical bounds on regret for different feedback settings
* Empirical comparison of the proposed algorithm (Bonus O-MD-CURL) with Greedy MD-CURL on multi-objective and constrained MDP tasks

The benchmark tasks chosen for evaluation are reasonable examples of CURL problems. However, the empirical evaluation has limitations:
* The experiments focus only on fixed objective functions and probability kernels, not adversarial settings
* The authors acknowledge challenges in implementing adversarial and bandit MDPs
* The state space used (11×11 grid world) is relatively small

Despite these limitations, I think the evaluation criteria are reasonable given the primarily theoretical focus of the paper, and the results do support the key benefit of this approach: better exploration leading to improved performance. However, it’s worth emphasizing that more robust, larger-scale experiments would strengthen practical confidence.

**Other Comments Or Suggestions:**

Some final comments/suggestions:
* Ablation studies to isolate the impact of different components (particularly exploration bonuses) would provide more insight.
* Comparing with additional baselines beyond Greedy MD-CURL would also be beneficial in evaluation.
* A more detailed analysis of computational complexity would help assess practical applicability.
* Discussion of potential extensions to function approximation or continuous state/action spaces would enhance impact.
* Examples of practical applications in areas like multi-objective optimization or risk-averse RL would also be nice to demonstrate relevance.

**Other Strengths And Weaknesses:**

The main strengths I identified are as follows:
1. The paper introduces a novel approach to handle exploration-exploitation in online CURL with unknown dynamics using exploration bonuses in the gradient.
2. The theoretical analysis is comprehensive, covering multiple feedback settings with carefully derived regret bounds.
3. The proposed algorithm has a closed-form solution, making it practical for implementation.
4. The approach addresses a general class of problems with potential applications in pure exploration, imitation learning, mean-field control, and risk-averse RL.
5. The paper is well-written with clear explanations of complex concepts, which is impressive for such a topic in my opinion.

That being said, I also identified a few weaknesses, some of them important:
1. Insufficient acknowledgment of prior work that directly addresses similar problems using similar techniques.
2. Overstated novelty claims that could mislead readers about the paper's contributions relative to the existing literature.
3. Limited discussion of real-world impact and potential applications. Despite CURL representing a significant generalization of RL that could potentially address numerous practical challenges, the paper provides very little context about:
   * Why readers should care about convex objectives versus linear ones
   * What real-world problems become tractable with these algorithms
   * How the algorithms compare with existing approaches for specific applications like risk-averse RL, imitation learning, or exploration
   * The trade-offs involved in using convex objectives.
   I understand that the quality of this work is already high and there is a lot to say so the authors likely chose to save space from a discussion, but I think that this is a missed opportunity to connect the very interesting theoretical results to practical impact. The paper would be strengthened by having a short discussion paragraph including perhaps examples of how CURL can better capture real-world requirements than standard RL, or case studies demonstrating problems where the convex formulation enables solutions that weren't previously possible. Without this context, readers may struggle to appreciate the full significance of the contribution beyond its theoretical merits.
4. Limited empirical validation restricted to specific tasks with fixed objective functions and probability kernels.
5. Lack of discussion on scalability to large state and action spaces.
6. The bandit CURL setting only achieves O(T^(3/4)) regret rather than the optimal O(√T).
7. The paper acknowledges implementation challenges for adversarial and bandit MDPs, limiting practical applicability.


All things considered, in its current form, I'm a bit torn on how to rate this paper. On the one hand, it is a technically sound work with thorough mathematical analysis, it provides closed-form solutions that are practically implementable and successfully extends techniques to the CURL setting. However, on the other hand, I think as I mentioned that novelty claims are overstated and it is positioned as revolutionary rather than incremental.

Hence, I am currently rating it as 3 (weak accept) and I'm willing to move this rating either way depending on the author's response. If the authors addressed my citation and novelty concerns, either by explaining why I might be wrong or by providing proper context for the proposed work and accurately positioning it within the literature, I would be very happy to change my rating to accept.

**Questions For Authors:**

- How does the performance of your algorithm scale with the size of the state and action spaces? The theoretical bounds include factors of |X| and |A|, but do you have empirical insights into performance on larger environments?
- In Section 4.2.2, you restrict the self-concordant regularization method to known MDPs. Do you see a path toward extending this approach to unknown MDPs, and what are the key technical challenges?
- The regret bound for bandit CURL is O(T^(3/4)) rather than the optimal O(√T). Do you believe this is a fundamental limitation of the problem setting, or is it possible to achieve O(√T) regret for bandit CURL with unknown dynamics?
- How sensitive is the performance of your algorithm to the choice of exploration bonuses? Have you experimented with alternative forms of bonuses, and if so, how do they compare?
- Have you applied your algorithm to any specific problem settings mentioned as potential applications (pure exploration, imitation learning, etc.) beyond the grid world examples? If so, what insights did you gain?
- How does your work specifically advance beyond Rosenberg & Mansour (2019), which already achieved Õ(√T) regret for convex performance criteria in MDPs with unknown dynamics?
- Your exploration bonus approach bears similarities to Jin et al. (2020)'s method of incorporating confidence-bound bonuses in mirror descent. Could you clarify the technical differences and innovations in your approach compared to theirs?
- The paper claims to be the first to address bandit feedback in CURL, but how does your approach technically differ from earlier works by Neu et al. (2012/2013) that adapted bandit convex optimization to reinforcement learning?

**Relation To Broader Scientific Literature:**

The paper is well-positioned within the literature on RL, convex optimization, and online learning. The authors provide a thorough discussion of related work:

* For offline CURL, they discuss work by Zhang et al. (2020, 2021), Barakat et al. (2023), Zahavy et al. (2021), Geist et al. (2022), Moreno et al. (2024), and Mutti et al. (2023a, 2023b).
* For online CURL, they identify Greedy MD-CURL from Moreno et al. (2024) as the only existing regret minimization algorithm for online CURL, explaining its limitations which the proposed work addresses.
* For RL approaches, they discuss model-optimistic methods (UCRL), value-optimistic methods (UCB-VI), and policy-optimization methods, explaining how the proposed approach differs.

Table 1 comprehensively compares  the proposed method to SOTA approaches, highlighting the contributions in achieving optimal regret, supporting CURL, providing closed-form solutions, incorporating exploration, avoiding model assumptions, handling adversarial losses, and supporting bandit feedback.

**Theoretical Claims:**

The theoretical claims and proofs appear to be sound. The key theoretical innovations include:
* Carefully designed exploration bonuses added to the sub-gradient of the objective function
* Decomposition of regret terms to handle the exploration-exploitation dilemma
* Adaptation of bandit convex optimization techniques to the MDP setting

The proofs follow a logical structure and build upon established results in online learning, reinforcement learning, and convex optimization. The authors extend Lemma 2.1 from previous work to handle sequences of bounded vectors and smoothly varying transitions.

For bandit feedback settings, the analysis becomes more complex due to the constraint set structure and transition kernel uncertainty. The authors introduce two approaches (entropic regularization and self-concordant regularization) with different assumptions and guarantees.

Assumption 4.2 (requiring minimum probability for all state transitions) may be restrictive in practical scenarios, which the authors acknowledge and provide an alternative approach with Assumption 4.4 for known MDPs.

---

> ### Author Rebuttal · Authors · 2025-03-30
>
> We thank the reviewer for their long and detailed review. We address below the raised concerns.
> - **References:**
>     - **Rosenberg and Mansour (2019):** Their setting differs from ours; in fact, our setting generalizes theirs. In our notation, they consider a sequence of adversarial losses $(\ell^t)_t$ with $\ell^t : \mathcal{X} \times \mathcal{A} \to \mathbb{R}$, where the learner's loss is given by $F(\langle \mu, \ell^t \rangle)$. Here, $F$ is a *fixed* and *known beforehand* convex function, $\mu$ is the state-action distribution, and $\langle \mu, \ell^t \rangle$ is the expected loss. In contrast, our online convex RL setting is a generalization, as it does not assume linearity in the distribution within $F$, and the convex function itself is adversarial, meaning that the learner's loss in episode $t$ is given by $F^t(\mu)$. Thus, we introduce a method that achieves the same $\sqrt{T}$ regret bound as Rosenberg and Mansour, but in a more general setting.
>
>     - **Jin et al. (2020):** This work *does not* use exploration bonuses. Their approach is model-optimistic: in each episode, they solve an OMD iteration over the set of *all MDPs* induced by a probability kernel within the confidence set around the estimate $\hat{p}^t$. This is their exploration mechanism.
>        In contrast, our method solves an instance of OMD over a *single MDP* in each episode, the one induced by the estimate $\hat{p}^t$. This is crucial for obtaining a closed-form solution, but eliminates the exploration mechanism in Jin et al (2020). As shown in Section 3, without an exploration mechanism, this approach would fail to achieve low regret. To address this while maintaining a closed-form solution, we add an exploration bonus to the sub-gradient in each OMD iteration, a step not taken by Jin et al. (2020), requiring a new type of analysis.
>
>     - **Neu et al (2012/2013):** These two works do *not* use bandit convex optimization (BCO) approaches.
>         They adopt online convex optimization methods to solve RL problems with a standard *linear* objective (sum of rewards). The same applies for many subsequent works such as Jin et al. (2020). Some of these works address bandit feedback, but it is not appropriate to place them in the BCO category as they only consider linear losses. Instead, BCO refers to online problems with bandit feedback and more general families of convex loss functions (such as the family of convex Lipschitz functions we consider).
> - **Motivation:** We list some motivation applications below that we will add to the final version of the paper.
>     - **Energy grid optimization:** To balance the energy production with the consumption, an energy provider may want to control the average consumption of electrical appliances (electric vehicles, water heaters, etc) to better match a target consumption. The task involves daily control, with the target consumption varying daily due to fluctuations in energy production. To protect user privacy, the energy provider has limited access to individual trajectories, but receives the average consumption of the whole population at the end of each day. The loss is usually quadratic on the state-action distribution. This problem can be framed as our CURL formulation. See Coffman et al. 2023, or Moreno et al. 2024.
>
>     - **Mean-field games (MFG) with potential reward:** As shown by Geist et al. (2022), a MFG with potential reward can be framed as a CURL problem. Therefore, any sequential decision problem with a large population of anonymous agents with symmetric interests and potential rewards, such as epidemic spreading, crowd motion control, etc, can be cast as CURL.
> - **Questions:**
>     - Computational complexity: See question 1 of reviewer vsjZ.
>
>     - Self-concordant regularization with unknown MDP: A main challenge in adapting the approach of Alg. 4 to the unknown MDP case is that the adoption of the log barrier regularizer introduces some technical difficulties in the analysis of OMD with changing decision sets. Hence, at the moment, the only workaround we see is the adoption of a (generally less efficient) model-optimistic approach to handle the uncertainty regarding the transition kernel.
>
>     - Improved regret bound for bandit CURL: Please see our response to Reviewer 5Sqt regarding bandit feedback.
>
>     - In our practical experiments, we observed that the key factor in the bonus vector is its inverse proportionality to $ N_n^t(x, a) $. However, we did not compare the performance of different decay rates other than the one discussed in Section 3.
>
>     - Other experiments: We also applied our algorithm to the pure exploration task within the same grid world environment, whose goal is to maximize the entropy. Since this objective inherently drives exploration, we found that adding a bonus had no impact on performance in this case.
>
> *Coffman et al. 2023, A unified framework for coordination of thermostatically controlled loads, Automatica*

---

> > ### Comment · Reviewer_1pSC · 2025-04-06
> >
> > I appreciate the authors' responses - particularly their clarifications about the technical distinctions from prior work. I think their explanation of how their setting genuinely generalizes Rosenberg & Mansour (2019) through arbitrary adversarial convex functions (rather than just a fixed convex function applied to linear losses) helps position the contribution more accurately. The energy grid optimization and examples also effectively demonstrate practical relevance. While I still believe the original presentation could have been more precise about the relationship to existing literature, the technical contributions appear valid and meaningful. I'm satisfied with their responses and support acceptance.

---

> > > ### Author Response · Authors · 2025-04-07
> > >
> > > We thank the reviewer for their response. We are glad that you are satisfied with our responses. We appreciate that you support the acceptance of the paper, and we would also be grateful if you update your score to reflect this.

---

### Official Review · Reviewer_vsjZ · 2025-03-08

**Overall Recommendation:** 3

**Summary:**

This paper addresses online convex RL, a generalization of RL in which the loss is a convex/concave function of the state-action occupancy, with adversarial losses and unknown transitions. In the setting with full feedback, i.e., the loss function is revealed to the agent at the end of the episode, the paper provides an optimistic bonus-based online mirror descent algorithm achieving $O (\sqrt{T})$ regret. Then, the paper studies variants of the algorithm for the bandit feedback setting, in which the loss function is only revealed for the visited state-action pairs. In this (more challenging) setting, the paper shows nearly optimal regret for linear loss (akin to adversarial MDP) and $O (T^{2/3})$ for convex/concave loss.

**Claims And Evidence:**

Mostly yes, but the wording of a couple of claims could be improved:
- While the paper has a point on providing the first near-optimal regret for this particular online CURL setting, it could do more to acknowledge that previous work studied online CURL (e.g., in a pure exploration PAC setting in Zahavy et al. 2021 or with assumptions on transitions in Moreno et al. 2024) and also regret minimization in a "trajectory version" of CURL (Chatterji et al. 2022, Mutti et al. 2023). Perhaps the tile/abstract could mention more clearly that this is the first study of CURL regret with adversarial losses and general transitions;
- The near-optimality of the regret is also partially misleading. An algorithm with additional $\sqrt{|X|}$ factor would not be considered nearly matching in standard RL and the claim that the latter is necessary for convex utilities (line 260) is not supported.

(Chatterji et al. 2022) "On the theory of reinforcement learning with once-per-episode feedback"

**Essential References Not Discussed:**

A stream of work that is closely related but (almost) not discussed is the one on trajectory-based CURL, such as (Chatterji et al. 2022, Mutti et al. 2022 and 2023) and submodular RL (Prajapat et al. 2023 "Submodular reinforcement learning" and De Santi et al. 2024 "Global reinforcement learning: Beyond linear and convex rewards via submodular semi-gradient methods"). I think the paper could do more to relate their findings to these works.

**Experimental Designs Or Analyses:**

The paper reports a brief empirical validation, although the validated results do not fully match the theoretical setting (e.g., adversarial losses).

**Methods And Evaluation Criteria:**

Regret minimization is widely accepted as an evaluation metric for online RL settings.

**Other Comments Or Suggestions:**

The paper is interesting overall, and while the technical novelty is limited, it still covers some important aspects of CURL that seem to be missing from previous works. I think the paper gives a net positive contribution and shall be accepted ideally, but with the given bandwidth constraints there might be more deserving papers for the limited spots. I report below some comments, suggestions on how the paper could further be improved, and questions.

FORMULATION

The formulation of the problem is somewhat odd. From my understanding, the agent receives a feedback at each step that is a convex/concave function of the state-action occupancy induced by the policy at that step. Thus, the feedback seems to be independent from the realization (given the policy). While this formulation might be easier to motivate in the offline CURL setting, I am wondering when it is justified in online CURL. This has been touched upon in Mutti et al. 2023 with the trajectory-based CURL formulation. While the latter is mentioned briefly in the paper, the justification for adopting the state-action occupancy version ("To align with prior work, we adopt the classic CURL formulation") is rather weak. Perhaps a stronger motivation would be to show that the trajectory version cannot be solved efficiently, and to see the occupancy version as a relaxation? I owuld be happy to hear more from the authors on this point of the motivation of CURL.

PRESENTATION

This is a nice technical work. However, I think the presentation would benefit from a more focused discussion of the results that focuses on fundamental questions (e.g., is CURL harder than RL? Does online CURL require different approaches w.r.t. online RL?) and leave technical aspects for the second part of the paper.

**Other Strengths And Weaknesses:**

Strengths
- Interesting results that seem to place CURL in the same ballpark of RL statistically, in terms of regret and methodologies, also with adversarial losses;
- The paper reads well and significant space is devoted to an overview of technical problems and how to overcome them.

Weaknesses
- While CURL counts several recent studies, the paper could do more to motivate the specific setting addressed, perhaps with potential applications. The formulation of the objective looks somewhat odd (see below).
- Techniques are mostly incremental, especially w.r.t. to Moreno et al. 2024;
- The presentation could be streamlined to give more breath between implications of the reported results and technical challenges to obtain them.

**Questions For Authors:**

1) Can the authors comment on the computational tractability of the presented algorithm?

2) Any algorithmic idea to take home for practice? This does not look dissimilar from Hazan et al. FW algorithm in nature, with the addition of count-based optimistic bonuses. However, count-based methods may not be widely employed in practice, e.g., for the same reasons they are not widely adopted in deep RL.

3) What does it mean stochastic/adversarial feedback in the CURL setting? The loss function appears to be always deterministic...

4) What is the point of showing bandit feedback in RL? This result shall not be new in the literature. Perhaps the idea is to see whether the general algorithm is nearly optimal also for the important RL sub-case?

5) The statements of the regret theorems report "For any policy $\pi \in \Pi$" and then they show the regret. What does this mean exactly? Is not the policy chosen by the algorithm?

6) The paper analyzes CURL with a generic concave/convex loss (or sometimes linear). Another interesting aspect is to study whether specific form of the loss (e.g. entropy of the occupancy) can lead to better specialized results.

**Relation To Broader Scientific Literature:**

This paper fits into a stream of works in convex utility reinforcement learning (Hazan et al. 2019, Zhang et al. 2020, Zahavy et al. 2021, Geist et al. 2022...) that generalize the standard RL setting to convex loss functions.
Within this area, it mostly builds upon Moreno et al. 2024, which also studies an online mirror descent algorithm in a similar setting but deals with transitions forced to a (less general) structure and only covers full-feedback setting. On a technical level, the extension require the design of optimistic bonuses for exploration (as it is common in online RL with unknown transitions) and borrowing techniques from bandit convex optimization for the bandit-feedback setting.

**Theoretical Claims:**

I did not check any proof in the appendix, so the derivations have not been closely inspected other than the high-level considerations on the techniques reported in the main text. While a further inspection of those may be useful, the techniques seem to be generally standard and the results reasonable for the considered settings.

---

> ### Author Rebuttal · Authors · 2025-03-30
>
> We thank the reviewer for their time and comments. We address the raised concerns below.
> - **Claims (papers):** We discuss Zahavy et al. (2021) as an offline approach, as they consider a fixed and known loss function aiming to minimize the optimization error (see their Eq. 5). In the final version, we will expand the discussion over works on trajectory-based CURL, see also **Formulation** below. The work of Moreno et al. (2024) is extensively discussed in the paper.
>
> - **Claims (near-optimality):** We use the term *near-optimal* to refer to optimality with respect to $T$. In the final version, we will add a clarification. For the $\sqrt{|\mathcal{X}|}$ term, our intention was to point that it is a consequence of our CURL approach, this too will be clarified.
>
> - **References:** Thank you for pointing out the works on submodular RL. We will add them to our related work, as well as expand on the discussion regarding trajectory-based CURL.
>
> - **Formulation:** The feedback related to the loss of the learner is indeed independent of the trajectory of the agent. The trajectory-based formulation of Mutti et al. (2023) is an interesting alternative. However, they show that non-Markovian policies can be necessary to optimize this objective, which entails an increased computational burden. On the other hand, the occupancy-measure-based formulation that we study can be solved efficiently, and allows direct comparison with methods from the CURL literature such as Moreno et al. (2024). Moreover, in application scenarios with many homogeneous agents, a mean-field approach can justify this choice. We give a motivating example in our response to reviewer 1pSC.
>
> - **New techniques:** We emphasize that, although our algorithm is based on the OMD framework proposed by Moreno et al. (2024), we introduce new technical tools that may be useful to the CURL and RL community.
> To address unknown probability kernels while preserving their closed-form solution, we introduce the exploration bonuses. Although the use of bonuses for exploration in RL is not new, algorithms with closed-form solutions and adversarial losses with bonuses initially achieved only sub-optimal regret in $T$ (Efroni et al. 2020). Luo et al. (2021) addressed this limitation by constructing a dilated bonus equation satisfying a dilated Bellman equation. However, this technique is not applicable to CURL, as its objective is convex, not linear over the distribution.
> To overcome this, we develop a novel additive exploration bonus which offers a new solution to this challenge. As demonstrated, this also achieves optimal regret in $T$ for bandit RL through a mechanism and analysis distinct from prior methods. In addition, we address, for the first time, bandit feedback in CURL for general convex Lipschitz functions.
>
> - **Questions:**
>   - 1. In the full-feedback setting, the bandit RL setting, and the first approach in the bandit CURL setting, the algorithm has a closed-form solution for the policy. In each episode, it performs $O(N \times |\mathcal{X}| \times |\mathcal{A}|)$ operations to compute the policy. Whereas for the second approach in the last setting, computing the policy at every step requires solving a convex program.
>   - 2. The algorithm of Hazan et al. is designed for a different objective than ours. Firstly, they focus on entropy maximization, while we deal with the CURL problem for any convex function. Secondly, our algorithm is made for an online episodic environment with adversarial losses, while their method works only with a fixed reward function, and they do not prove regret bounds. Finally, they assume access to an approximate planning oracle and a state distribution estimation oracle, which eliminates the need for an explicit exploration tool, unlike our approach, where the bonus plays an essential role in guaranteeing sufficient exploration.
>
>          An idea to take home for practice is that OMD in the space of occupancy-measures handles adversarial losses and, with a well-designed bonus, ensures sufficient exploration while still enjoying a closed-form solution.
>   - 3. One example of stochastic losses is when $F_t(\mu) = (\mu - \gamma_t)^2$ where $\gamma_t$ is sampled i.i.d. from some distribution. In our case, we consider adversarial losses, meaning we make no statistical assumptions about the loss sequence, which can follow any pattern.
>   - 4. Yes. Since our bandit CURL approaches achieve a regret of $T^{3/4}$, we demonstrate that when $F$ is linear, the problem is simpler, allowing us to achieve a regret of $\sqrt{T}$. Another point is to show a novel approach to adversarial online RL that matches the existing bounds, while introducing a new algorithm based on different tools and analysis that could be of interest to the community.
>   - 5. The policy $\pi $ is the comparator strategy in the regret (see Eq.(5)).
>   - 6. We agree with the reviewer and leave this question for future work (see also answer to Question 1 of reviewer 5Sqt).

---

> > ### Comment · Reviewer_vsjZ · 2025-04-03
> >
> > Dear authors,
> >
> > Thanks for the various clarifications and for addressing my comments. A few notes below to clarify some of my previous points (they do not need to be addressed).
> >
> > **Formulation.** "Therefore, any sequential decision problem with a large population of anonymous agents with symmetric interests and potential rewards, such as epidemic spreading, crowd motion control, etc, can be cast as CURL." Those are interesting setting, but perhaps the area of application can be better highlighted in the manuscript.
> >
> > 2) I think the algorithm of Hazan et al. is general (or it can be generalized easily to any convex/concave utility). I did not meant to say that Hazan et al. have the same results of this paper. What I meant, is that the algorithm presented here have some similarities with their FW + exploration bonuses.

---

> > > ### Author Response · Authors · 2025-04-07
> > >
> > > We thank the reviewer for their response. We are glad to have addressed your comments.

---

### Official Review · Reviewer_5Sqt · 2025-03-11

**Overall Recommendation:** 3

**Summary:**

This paper studies the setting of online RL with concave utility function. This paper analyze two settings:
1. When the learner receive the full information of the utility function at each step.
2. When the learner only receive bandit feedback at each step.
At the first setting, the learner propose an FTRL-type algorithm, and proved that the regret is bounded by \sqrt{T} for tabular MDP case. For the second setting, the authors consider the bandit feedback, where they obtained regret upper bound for unknown MDP but with a lower bound on the transition probability, or known MDP without the assumption. Both the regret bounds are $T^{3/4}$.

**Claims And Evidence:**

Yes.

**Essential References Not Discussed:**

No, as far as I known.

**Experimental Designs Or Analyses:**

Yes, the experiment setting seems to be sound to me.

**Methods And Evaluation Criteria:**

Yes.

**Other Comments Or Suggestions:**

I do not have further comments or suggestions.

**Other Strengths And Weaknesses:**

This paper has the following strengths and weaknesses:

Strengths:
1. The setting of concave utility function for RL is interesting to me.
2. This paper is well-written.

Weaknesses:
1. This paper only considers the tabular RL case, which is restrictive. It will be better if the results can be generalized into more general setting, e.g. linear MDP or MDP with function approximation.
2. The regret bound for bandit feedback to suboptimal. This setting is equivalent to the setting of adversarial MDP, where we know the optimal regret is $O(\sqrt{T})$, without any additional assumptions.
3. The algorithms and analysis in this paper are very similar to the FTRL and its analysis, except in the MDP setting. The novelty of the results in this paper is limited.

**Questions For Authors:**

I have the following questions for the author:
1. A common belief in convex optimization is that the curvature of the loss function could help reducing the regret. Can you improve the results from $O(\sqrt{T}$ to sharper regret bound, if we assume some curvature of the loss function, e.g. if the loss function is quadratic?

**Relation To Broader Scientific Literature:**

This paper improves from previous literatures, as they come up with an algorithm which works for concave utility function.

**Theoretical Claims:**

Yes, I checked the proof of some of the theorems, e.g. Theorem 3.2 and Theorem 4.1.

---

> ### Author Rebuttal · Authors · 2025-03-30
>
> We thank the reviewer for their time and comments. We address the raised concerns below.
>
> - **Tabular RL:** We agree with the reviewer that both the linear MDP case and the case with function approximation are interesting directions for future work. However, since the adversarial convex reinforcement learning setting with an unknown probability kernel had not been addressed even in the tabular case, we believe that developing a foundational algorithm like ours is a necessary step towards adapting to these more general scenarios in the future.
>
> - **Bandit feedback:** The adversarial convex RL setting is not equivalent to adversarial MDPs with the standard (linear) RL objective.
>     Under bandit feedback, we indeed show that in the latter setting (standard linear RL), our method achieves $\sqrt{T}$ regret (see Thm. 4.1).
>     On the other hand, addressing bandit feedback in convex RL is a more challenging problem, please refer to the second paragraph in Section 4.2 (page 6, line 285, column 2).
>     In short, the adoption of general convex objectives places this problem in the more challenging Bandit Convex Optimization (BCO) field, as opposed to bandit (or semi-bandit) linear optimization, where one can generally categorize standard RL problems with bandit feedback.
>
>     As we discuss in the beginning of Subsection 4.2.1, achieving $\sqrt{T}$ regret (still an active research area) has been shown to be possible in plain BCO problems with Lipschitz objectives via significantly involved algorithmic techniques and analyses (Hazan and Li, 2016; Bubeck et al., 2021; Fokkema et al., 2024).
>     On the contrary, we adopt more common and foundational techniques (building upon works such as Flaxman et al., 2005 and Saha and Tewari, 2011)
>     that though only capable of achieving $T^{3/4}$ regret in our setting (Hu et al., 2016),
>     are arguably more practical, less complicated, and enjoy better dimension dependence.
>     This allows us to better isolate the difficulties posed by the specific structure of this new problem (BCO in MDPs) as we discuss in much detail throughout Section 4.2.
>     Finally, we note that when the dynamics of the MDP are not known (as is the case in Subsection 4.2.1), a new and significant difficulty is added to the BCO formulation of the problem (namely, uncertainty over the decision set), hence it is not clear in that case whether $\sqrt{T}$ is still achievable.
>
> - **MDP setting:** Applying online learning algorithms to the MDP setting is challenging and demands the development of new tools. Numerous studies in the literature have focused on adapting online learning methods (such as FTRL, or OMD) to MDPs within the reinforcement learning framework, ranging from early works addressing the basic case with known transition dynamics (Even-Dar et al., 2009; Neu et al., 2010) to more recent approaches handling unknown transitions (see references in Section 2).
>     In our paper, dealing with adversarial convex RL *is not a straightforward application of OMD* because of the need for exploration (which we handle with our bonuses technique), the changing decision sets (due to uncertainty over the transition kernel), and the bandit feedback.
>
> - **Question:** As outlined in Section 3, Eq.(8), the analysis of the regret can be divided into two main components: one concerning the quality of the MDP estimation under the executed policy, $ R_T^{\text{MDP}} $, and the other related to the online algorithm that computes the executed policy, $ R_T^{\text{policy}} $. Indeed, the term related to the online algorithm could be improved by assuming curvature in the loss function, but it is unclear if the term related to estimating the MDP can be improved
> because it is not strongly tied to the structure of the loss function.
>
> *Neu et al., 2010, Online Markov decision processes under bandit feedback, NeurIPS*

---

> > ### Comment · Reviewer_5Sqt · 2025-04-03
> >
> > I think my concerns are addressed. Hence I increase my score accordingly.

---

> > > ### Author Response · Authors · 2025-04-07
> > >
> > > We thank the reviewer for their response. We are glad that we have addressed your concerns, and that you have increased your evaluation of the paper.

---

### Official Review · Reviewer_mGhi · 2025-03-11

**Overall Recommendation:** 3

**Summary:**

The paper proposes a mirror descent algorithm (with exploration bonuses) for achieving sub-linear regret in concave utility RL in an online episodic and adversarial setting. The authors first propose an algorithm designed for full-feedback over adversarial losses, achieving sub-linear regret. Then, they propose two algorithms for the bandit feedback and adversarial losses, achieving sub-linear regret under some assumptions of the convex MDP, but less restrictive ones related to previous works ( model of the dynamics assumed to be known, out of additive noise [Moreno et al 2024]). Finally, the authors test the most scalable algorithm (in terms of assumptions required for sub-linear regret) against one by Moreno, showing better performances in terms of losses and regret.

**Claims And Evidence:**

Yes. The claims are supported by proofs and empirical corroboration.

**Essential References Not Discussed:**

Not that I am aware of.

**Experimental Designs Or Analyses:**

Yes, all of the experiments in section 5. Unfortunately, I was not able to find a comment on how the regret was actually computed (with respect to which policy). Also, I was not able to find a discussion on why Moreno et al. 2024's algorithm fails miserably in Fig.3 even though I would say it is not expected to. Finally, I am curious about some performances  in Fig.2 (see Questions).

**Methods And Evaluation Criteria:**

Yes, the authors selected a relevant empirical scenario already considered in the most related work in the literature.

**Other Comments Or Suggestions:**

- I would suggest to find some space for some concluding comments.
- I would suggest to comment on the failure modes of Moreno's Algorithm in Fig.3.
- I would suggest to make what assumptions are make throughout the paper more explicit in the introduction.

**Other Strengths And Weaknesses:**

Strengths:
- Generally extremely well written and rigorous.
- Relevant contribution for the field.

Weaknesses:
- Some text re-formatting might help the reader in following the proof outlined in the main paper.
- Original contributions in proof techniques and/or algorithmic tools (in terms of bonuses) might be made more explicit.
- Assumptions needed for sub-linearity could be made way more explicit, even in the introduction.

**Questions For Authors:**

- Does Alg. 4 require to know the MDP? It is stated but not written particularly explicitly. I would definitely suggest to make this fact explicit even in the introduction in case this Is the case.
- What is $\xi$ in Eq. 9? Did I miss it in the previous parts?
- How is the regret computed?
- Why in Fig. 2 Moreno's algorithm show loss minimisation similar to the most recent algorithm but regret far worse?

**Relation To Broader Scientific Literature:**

This is the first paper addressing online episodic RL with concave (adversarial) utilities without strong assumptions on the model of the cMDP but guaranteeing sub-linear regret.

**Theoretical Claims:**

Yes, I checked the soundness of the steps in the main paper.

---

> ### Author Rebuttal · Authors · 2025-03-30
>
> We thank the reviewer for their time and comments. We address the raised concerns below.
>
> - **Regret:** In our experiments, we compare our approach with the oracle optimal policy, which can be well approximated when the dynamics are fully known. We will specify it in the final version of the paper.
>
> - **Failure of Moreno et al. (2024):** To succeed in the constrained MDP task of Figure 3, the agent must explore sufficiently to arrive at the rewarding final state. However, without an explicit exploration incentive, the agent remains static to avoid the constraints that induce negative rewards. Since Greedy-MD-CURL does not have an explicit exploration mechanism, it fails to converge within 1000 iterations. Could the reviewer clarify why they believe Greedy-MD-CURL would not fail in this scenario?
>
> - **Assumption for sub-linearity:** In the full-information case, and the bandit RL case, the two assumptions needed for sub-linear regret are convexity and Lipschitzness of the objective function, which are both stated in the introduction and the setting. For bandit feedback in the CURL setting, in addition to these two hypothesis, we state in the introduction (page 2, line 84-86, column 1) that one of the presented algorithms requires the MDP to be known and that the other requires an assumption on the structure of the MDP, which is then precisely detailed and motivated in Section 4.2.1. In the final version, we will add more details in the introduction.
>
> - **Questions:**
>     - **Assumptions Alg. 4:** Yes, we assume the MDP is known for this algorithm. We mention this assumption in the introduction (page 2, lines 84-85, column 1) and before the first reference to the algorithm (page 7, lines 370-371, column 2). Additionally, the first input argument in the definition of the algorithm is the set of occupancy measures under the true MDP.
>     - **Definition of $\xi$:** $\xi_n^t(x,a) := \|p_n(\cdot|x,a) - \hat{p}^t_n(\cdot|x,a) \|_1$. We define it in the paragraph before Eq.(9) (see page 4, line 218, column 2).
>     - **Fig.2:** The regret plot represents the cumulative loss over time against the loss of the optimal policy. The loss plot indicates that Greedy-MD-CURL (Moreno et al. 2024) does not converge to the minimum value, justifying why the regret increases linearly. Note that the loss plot is in loglog scale while the regret plot is not.
>
> - **Comments:** Could the reviewer specify which parts of the paper could benefit from reformatting? We would appreciate more detailed feedback to improve the paper's presentation.
> Regarding the exploration bonuses, we would like to emphasize that the entire analysis in Section 3, which introduces an additive bonus in mirror descent, is an original contribution, see also **New techniques** in our response to Reviewer vsjZ. Additionally, we will include a conclusion section and comment more on the experimental results.

---

> > ### Comment · Reviewer_mGhi · 2025-04-04
> >
> > I thank the authors for the adeguate response and for addressing my doubts (on the reasons for failures of Greedy-MD-CURL), I think addressing reviewer 1pSC's concerns will be enough for motivating an accept.
> >
> > In general, I would say I am more comfortable when proofs are provided by points that align with the main logical steps, while blocks of text which much verbosity do not help me follow them. Yet, this is far from being a blocking weaknesses.

---

> > > ### Author Response · Authors · 2025-04-07
> > >
> > > We thank the reviewer for their response. We are glad to have adequately addressed your doubts. We appreciate that you support the acceptance of the paper, and we would also be grateful (considering also our discussion with reviewer 1pSC) if you update your score to reflect this.

---

### Decision · Program_Chairs · 2025-05-01

**Decision:**

Accept (poster)

**Comment:**

This paper studies the episodic setting for online MDPs with concave reward functions (generalized from linear rewards) and provides near-optimal regret bounds in the tabular setting through a novel mirror descent and exploration bonus. They also study the bandit-feedback version of this problem and leverage tools from bandit convex optimization. All reviewers were fairly positive about this submission, in particular mentioning that it makes a solid contribution to the literature. However, some reviewers did have concerns about the restriction to the tabular setting, limited technical novelty over the prior work of Moreno et al (2024), which addressed a similar setting, and the formulation not being sufficiently connected to practice. Based on this feedback, I recommend that the paper be accepted if there is room.